# Optimal deep learning of holomorphic operators between Banach spaces

**Ben Adcock**
Department of Mathematics
Simon Fraser University
Canada

**Nick Dexter**
Department of Scientific Computing
Florida State University
USA

**Sebastian Moraga**
Department of Mathematics
Simon Fraser University
Canada

## Abstract

Operator learning problems arise in many key areas of scientific computing where Partial Differential Equations (PDEs) are used to model physical systems. In such scenarios, the operators map between Banach or Hilbert spaces. In this work, we tackle the problem of learning operators between Banach spaces, in contrast to the vast majority of past works considering only Hilbert spaces. We focus on learning holomorphic operators – an important class of problems with many applications. We combine arbitrary approximate encoders and decoders with standard feedforward Deep Neural Network (DNN) architectures – specifically, those with constant width exceeding the depth – under standard $\ell^2$-loss minimization. We first identify a family of DNNs such that the resulting Deep Learning (DL) procedure achieves optimal generalization bounds for such operators. For standard fully-connected architectures, we then show that there are uncountably many minimizers of the training problem that yield equivalent optimal performance. The DNN architectures we consider are 'problem agnostic', with width and depth only depending on the amount of training data $m$ and not on regularity assumptions of the target operator. Next, we show that DL is optimal for this problem: no recovery procedure can surpass these generalization bounds up to log terms. Finally, we present numerical results demonstrating the practical performance on challenging problems including the parametric diffusion, Navier-Stokes-Brinkman and Boussinesq PDEs.

## 1 Introduction

Operator learning is increasingly being investigated for problems arising in computational science and engineering. These problems are often posed in terms of Partial Differential Equations (PDEs), which can be viewed as operators mapping function spaces to function spaces. Depending on the requirements for well-posedness of the PDE, both the input and solution spaces are often Hilbert, or, more generally, Banach spaces. The aim of operator learning is to efficiently capture the dynamic behavior of these operators using surrogate models, typically based on Deep Neural Networks (DNNs). Specifically, we want to learn

$$F : \mathcal{X} \to \mathcal{Y}, \qquad X \in \mathcal{X} \mapsto F(X) \in \mathcal{Y}, \tag{1.1}$$

where $\mathcal{Y}$ is the PDE solution space, $X$ represents the data supplied to the PDE, i.e., possibly multiple functions describing initial and boundary conditions or forcing terms or, equivalently, a vector of parameters defining such functions.

38th Conference on Neural Information Processing Systems (NeurIPS 2024).

Let $\mu$ be a probability measure on $\mathcal{X}$. Given noisy training data

$$\{(X_i, F(X_i) + E_i)\}_{i=1}^m, \tag{1.2}$$

where $X_1, \ldots, X_m \sim_{\text{i.i.d.}} \mu$ and $E_i$ is noise, a typical operator learning methodology consists of three objects: an approximate encoder $\mathcal{E}_{\mathcal{X}} : \mathcal{X} \to \mathbb{R}^{d_{\mathcal{X}}}$, an approximate decoder $\mathcal{D}_{\mathcal{Y}} : \mathbb{R}^{d_{\mathcal{Y}}} \to \mathcal{Y}$ and a DNN $\widehat{N} : \mathbb{R}^{d_{\mathcal{X}}} \to \mathbb{R}^{d_{\mathcal{Y}}}$. It then approximates $F$ as

$$F \approx \widehat{F} := \mathcal{D}_{\mathcal{Y}} \circ \widehat{N} \circ \mathcal{E}_{\mathcal{X}}. \tag{1.3}$$

The encoder and decoder are either specified by the problem, learned separately from data, or learned concurrently with $\widehat{N}$. The goal, as in all supervised learning problems, is to ensure good generalization via the learned operator $\widehat{F}$ from as little training data $m$ as possible.

## 1.1 Contributions

As noted in, e.g., [16, 66], the theory of deep operator learning is still in its infancy. We contribute to this growth in the following ways. We consider learning classes of *holomorphic* operators (Assumption 2.2), with arbitrary approximate encoders $\mathcal{E}_{\mathcal{X}}$ and decoders $\mathcal{D}_{\mathcal{Y}}$. As we explain in §2.3 (see also [52, §5.2], [53, §3.4] and [41]) these operators are relevant in many applications, notably those involving *parametric* PDEs. The main contributions of this work are as follows.

1. We consider operators taking values in general Banach spaces. As noted, the vast majority of existing work (with the notable exception of [16]) considers Hilbert spaces.
2. We consider standard feedforward DNN architectures (constant width, width exceeds depth) and training procedures ($\ell^2$-loss minimization).
3. (Theorem 3.1) We construct a family of DNNs such that any approximate minimizer of the corresponding training problem satisfies a generalization bound that is explicit in the various error sources: namely, an *approximation error*, which decays algebraically in the amount of training data $m$; *encoding-decoding errors*, which depend on the accuracy of the learned encoders and decoders; an *optimization error*, and; a *sampling error*, which depends on the noise $E_i$ in (1.2).
4. These DNN architectures are *problem agnostic*; they depend on $m$ only. In particular, the architectures are completely independent on the regularity assumptions of target operator.
5. (Theorem 3.2) We show that training problems based on *any* family of fully-connected DNNs possess uncountably many minimizers that achieve the same generalization bounds.
6. (Theorems 3.1-3.2) We provide bounds in both the $L_\mu^2$- and $L_\mu^\infty$-norms that hold in high probability, rather than just expectation.
7. (Theorems 4.1-4.2) We show that the generalization bound is optimal with respect to $m$: no learning procedure (not necessarily DL-based) can achieve better rates in $m$ up to log terms.
8. Finally, we present a series of experiments demonstrating the efficacy of DL on challenging problems such as the parametric diffusion, Navier-Stokes-Brinkman and Boussinesq PDEs, the latter two of which involve operators whose codomains are Banach, as opposed to Hilbert, spaces.

## 1.2 Relation to previous work

Approximating an operator between function spaces with training data obtained through numerical PDEs solves presents a formidable challenge. Nevertheless, in recent years, significant advances have been made through the development of DL techniques, leading to the field of *operator learning* [15, 40, 51, 53, 56, 58, 60, 62, 69, 83, 93, 103]. These approaches often leverage intricate DNN architectures to approximate the complex mappings inherent in physical modelling scenarios. Many works have also focused on the practical aspects of operator learning in real-world applications [13, 24, 35–37, 42, 45, 48, 49, 59, 61, 64, 65, 70, 73, 77, 80, 81, 84, 96, 98–101, 104, 105].

On the theoretical side, universal approximation theorems for operator learning have been developed in [50, 55, 68, 69] and elsewhere. Such bounds are typically not quantitative in the size of the DNN needed to achieve a certain error. For this, one typically either restricts to specific operators (e.g., certain PDEs) or imposes regularity conditions. One such assumption is Lipschitz regularity – see [10, 16, 55, 66, 87] and references therein. However, learning Lipschitz operators suffers from a *curse of parametric complexity* [54], meaning that algebraic rates may not be achievable. Another

common assumption is holomorphy. While stronger, it is, as noted, very relevant to operator learning problems involving parametric PDEs. Quantitative approximation results for holomorphic operators have been shown in [26, 31, 41, 55, 71] and elsewhere.

However, these works do not consider the generalization error, i.e., the error incurred when learning the approximation (1.3) from the finite training data (1.2). This is particularly important in applications of operator learning where data is obtained through expensive numerical PDE solves, since such problems are highly *data-starved*. Several works have tackled this question from the perspective of statistical learning theory and nonparametric estimation [16, 55, 66], but only for Lipschitz operators. As observed in [6, §9.5], this approach generally leads to a best $\mathcal{O}(m^{-1/2})$ decay of the $L_\mu^2$-norm error with respect to $m$. Theorem 4.1 shows that such a rate is strictly suboptimal for learning the classes of holomorphic operators we consider. Our generalization bounds in Theorems 3.1-3.2 do not use such techniques, and yield near-optimal rates in both the $L_\mu^2$- *and* $L_\mu^\infty$-norms. See also [30] for some related work in this direction for reduced-order modelling with convolutional autoencoders.

Our work is inspired by recent research on learning holomorphic, Banach-valued functions [2, 5, 6]. We extend both these works, in particular, [5], to learning holomorphic operators. We also significantly improve the error decay rates in [5] with respect to $m$ and show they can be achieved using substantially smaller DNNs with standard training (i.e., $\ell^2$-loss minimization). See Remarks C.1-C.2. Our theoretical guarantees fall into the category of *encoder-decoder-nets* [52], which includes the well-known *PCA-Net* [10] and *DeepONet* [68] frameworks. As in other recent works [16, 30, 55, 66], in Theorems 3.1-3.2 we assume the encoder-decoder pair $(\mathcal{E}_\mathcal{X}, \mathcal{D}_\mathcal{Y})$ in (1.3) have been learned, and focus on the generalization error when training the DNN $\widehat{N}$.

## 2 Notation, assumptions, setup and examples

### 2.1 Notation

Let $(\mathcal{X}, \|\cdot\|_\mathcal{X})$ and $(\mathcal{Y}, \|\cdot\|_\mathcal{Y})$ be Banach spaces and $\mu$ be a probability measure on $\mathcal{X}$. Let $(\mathcal{Y}^*, \|\cdot\|_{\mathcal{Y}^*})$ be the dual of $\mathcal{Y}$ and $B(\mathcal{Y}^*)$ be its unit ball. The *Bochner* and *Pettis* $L^p$-norms of a (strongly and weakly, respectively) measurable operator $F : \mathcal{X} \to \mathcal{Y}$ are defined as

$$\|F\|_{L_\mu^p(\mathcal{X};\mathcal{Y})} = \left( \int_\mathcal{X} \|F(X)\|_\mathcal{Y}^p \, \mathrm{d}\mu(X) \right)^{1/p}$$

$$\|F\|_{L_\mu^p(\mathcal{X};\mathcal{Y})} = \sup_{y^* \in B(\mathcal{Y}^*)} \left( \int_\mathcal{X} |y^*(F)|^p \, \mathrm{d}\mu(X) \right)^{1/p},$$

respectively, for $1 \leq p < \infty$, and analogously for $p = \infty$ (see, e.g., [8, 44]). Notice that $\|F\|_{L_\mu^p(\mathcal{X};\mathcal{Y})} \leq \|F\|_{L_\mu^p(\mathcal{X};\mathcal{Y})}$ for $1 \leq p < \infty$, while $\|F\|_{L_\mu^\infty(\mathcal{X};\mathcal{Y})} = \|F\|_{L_\mu^\infty(\mathcal{X};\mathcal{Y})}$.

Throughout this work, $\ell^p(\mathbb{N})$, $0 < p \leq \infty$ denotes the standard $\ell^p$ space with (quasi-)norm $\|\cdot\|_p$. We also define the *monotone $\ell^p$ space* $\ell_M^p(\mathbb{N})$ as the space of all sequences $\boldsymbol{z} = (z_i)_{i=1}^\infty \in \mathbb{R}^\mathbb{N}$ whose minimal monotone majorant $\tilde{\boldsymbol{z}} \in \ell^p(\mathbb{N})$. Here $\boldsymbol{z} = (\tilde{z}_i)_{i=1}^\infty$ is defined as $\tilde{z}_i = \sup_{j \geq i} |z_j|$.

Given a (componentwise) activation function $\sigma$, we consider feedforward DNNs of the form

$$N : \mathbb{R}^n \to \mathbb{R}^k, \ \boldsymbol{z} \mapsto N(\boldsymbol{z}) = \mathcal{A}_{L+1}(\sigma(\mathcal{A}_L(\sigma(\cdots \sigma(\mathcal{A}_0(\boldsymbol{z})) \cdots)))), \tag{2.1}$$

where $\mathcal{A}_l : \mathbb{R}^{N_l} \to \mathbb{R}^{N_{l+1}}$ are affine maps, and $N_0 = n$ and $N_{L+2} = k$. We define $\mathrm{width}(N) = \max\{N_1, \ldots, N_{L+1}\}$ and $\mathrm{depth}(N) = L$. We denote a class of DNNs of the form (2.1) with a fixed architecture (i.e., fixed activation function, depth and widths) as $\mathcal{N}$, and write $\mathrm{width}(\mathcal{N}) = \max\{N_1, \ldots, N_{L+1}\}$ and $\mathrm{depth}(\mathcal{N}) = L$.

### 2.2 Assumptions and setup

Let $F : \mathcal{X} \to \mathcal{Y}$ be the unknown operator we seek to learn and

$$\widetilde{\mathcal{E}}_\mathcal{X} : \mathcal{X} \to \mathbb{R}^{d_\mathcal{X}}, \ \widetilde{\mathcal{D}}_\mathcal{X} : \mathbb{R}^{d_\mathcal{X}} \to \mathcal{X}, \qquad \mathcal{E}_\mathcal{Y} : \mathcal{Y} \to \mathbb{R}^{d_\mathcal{Y}}, \ \mathcal{D}_\mathcal{Y} : \mathbb{R}^{d_\mathcal{Y}} \to \mathcal{Y}$$

be approximate encoders and decoders for $\mathcal{X}$ and $\mathcal{Y}$, respectively. As mentioned, we assume that these maps have already been learned, and focus on the training of the DNN $\widehat{N}$ in (1.3). Our main

results allow for arbitrary encoders and decoders (subject to the assumptions detailed below), and provide generalization bounds that are explicit in these terms: specifically, they depend on how well each encoder-decoder pair approximates the respective identity map on $\mathcal{X}$ or $\mathcal{Y}$.

In order to formulate the precise notion holomorphy for the operator $F$, we require the following. Let $D = [-1, 1]^{\mathbb{N}}$ and $\varrho$ be the uniform probability measure on $D$. Given $\rho > 1$, we define the *Bernstein ellipse* $\mathcal{E}(\rho) = \left\{ (x + x^{-1})/2 : x \in \mathbb{C},\ 1 \leq |x| \leq \rho \right\} \subset \mathbb{C}$, and, for convenience, we let $\mathcal{E}(1) = [-1, 1]$. Next, for $\boldsymbol{\rho} = (\rho_i)_{i \in \mathbb{N}} \geq \mathbf{1}$, we define the *Bernstein polyellipse* as the product $\mathcal{E}(\boldsymbol{\rho}) = \mathcal{E}(\rho_1) \times \mathcal{E}(\rho_2) \times \cdots \subset \mathbb{C}^{\mathbb{N}}$.

**Definition 2.1** (Holomorphic map). Let $\varepsilon > 0$, $\boldsymbol{b} \in \ell^1(\mathbb{N})$ with $\boldsymbol{b} \geq \mathbf{0}$. A Banach-valued function $f : D \to \mathcal{Y}$ is $(\boldsymbol{b}, \varepsilon)$-*holomorphic* if it is holomorphic in the region

$$\mathcal{R}(\boldsymbol{b}, \varepsilon) = \bigcup \left\{ \mathcal{E}(\boldsymbol{\rho}) : \boldsymbol{\rho} \geq \mathbf{1},\ \sum_{j=1}^{\infty} \left( (\rho_j + \rho_j^{-1})/2 - 1 \right) b_j \leq \varepsilon \right\} \subset \mathbb{C}^{\mathbb{N}}, \quad \boldsymbol{b} = (b_j)_{j \in \mathbb{N}}. \quad (2.2)$$

See, e.g., [17, 85]. As noted in [7] we can, by rescaling $\boldsymbol{b}$, assume that $\varepsilon = 1$. For convenience, we define the following unit ball, consisting of all such functions of norm at most one over $\mathcal{R}(\boldsymbol{b}, 1)$:

$$\mathcal{H}(\boldsymbol{b}) = \{ f : D \to \mathcal{V}\ (\boldsymbol{b}, 1)\text{-holomorphic} : \|f(\boldsymbol{x})\|_{\mathcal{Y}} \leq 1,\ \forall \boldsymbol{x} \in \mathcal{R}(\boldsymbol{b}, 1) \}. \quad (2.3)$$

**Assumption 2.2.** *Let $D = [-1, 1]^{\mathbb{N}}$ and $\varrho$ be the uniform probability measure on $D$.*
*(A.I) There is a measurable mapping $\iota : \mathcal{X} \to \mathbb{R}^{\mathbb{N}}$ such that pushforward $\varsigma := \iota \sharp \mu$ is a quasi-uniform measure supported on $D$ and $\iota|_{\mathrm{supp}(\mu)} : \mathcal{X} \to \ell^{\infty}(\mathbb{N})$ is Lipschitz with constant $L_{\iota} \geq 0$.*
*(A.II) The operator $F$ has the form $F = f \circ \iota$, where $f \in \mathcal{H}(\boldsymbol{b})$ for some $\boldsymbol{b} \in \ell_{\mathsf{M}}^p(\mathbb{N})$ and $0 < p < 1$.*
*(A.III) The map $\mathcal{E}_{\mathcal{X}} := \iota_{d_{\mathcal{X}}} \circ \widetilde{\mathcal{D}}_{\mathcal{X}} \circ \widetilde{\mathcal{E}}_{\mathcal{X}}$ is measurable (here $\iota_{d_{\mathcal{X}}} : \mathcal{X} \to \mathbb{R}^{d_{\mathcal{X}}}$ is the restriction of $\iota$, i.e., $\iota_{d_{\mathcal{X}}}(X) = (\iota(X)_i)_{i=1}^{d_{\mathcal{X}}})$ and the pushforward $\tilde{\varsigma} := \mathcal{E}_{\mathcal{X}} \sharp \mu$ is absolutely continuous with respect to $\varrho$.*
*(A.IV) The maps $\mathcal{D}_{\mathcal{Y}}$ and $\mathcal{E}_{\mathcal{Y}}$ are linear and bounded.*

Now let $X_1, \ldots, X_m \sim_{\mathrm{i.i.d.}} \mu$ and consider the training data

$$\{(X_i, Y_i)\}_{i=1}^m \subset (\mathcal{X} \times \mathcal{Y})^m, \quad \text{where } Y_i = F(X_i) + E_i \in \mathcal{Y} \quad (2.4)$$

and $E_i \in \mathcal{Y}$ represents noise. Let $\mathcal{N}$ be a class of DNNs $N : \mathbb{R}^{d_{\mathcal{X}}} \to \mathbb{R}^{d_{\mathcal{Y}}}$, and define

$$F \approx \widehat{F} := \mathcal{D}_{\mathcal{Y}} \circ \widehat{N} \circ \mathcal{E}_{\mathcal{X}}, \quad \text{where } \widehat{N} \in \operatorname*{argmin}_{N \in \mathcal{N}} \frac{1}{m} \sum_{i=1}^m \|Y_i - \mathcal{D}_{\mathcal{Y}} \circ N \circ \mathcal{E}_{\mathcal{X}}(X_i)\|_{\mathcal{Y}}^2. \quad (2.5)$$

## 2.3 Discussion of assumptions

We now discuss (A.I)-(A.IV). In §6 we describe future work on relaxing these assumptions.

(A.I) is a weak assumption. It asserts that there is a Lipschitz map $\iota$ under which the pushforward of $\mu$ is a quasi-uniform measure supported in $D$. As we discuss in Example 2.3, this is notably the case when $\mu$ is the law of some random field with an affine parametrization involving bounded random variables – a situation that occurs frequently in parametric and stochastic PDE problems. (A.II) describes the specific holomorphy of the operator $F$ – see Remark 2.4 for details. Note that we require $\boldsymbol{b} \in \ell_{\mathsf{M}}^p(\mathbb{N})$, not just $\boldsymbol{b} \in \ell^p(\mathbb{N})$. It is known [7] that one cannot learn holomorphic functions (and hence operators) from finite data if $\boldsymbol{b} \in \ell^p(\mathbb{N})$ only. (A.III) is a relatively weak assumption. In view of (A.I), we expect it to hold as long as the $\widetilde{\mathcal{D}}_{\mathcal{X}} \circ \widetilde{\mathcal{E}}_{\mathcal{X}} \approx \mathcal{I}_{\mathcal{X}}$ sufficiently well. Finally, (A.IV) is a standard assumption, which holds for instance in the case of PCA-Net and DeepONet. The former also enforces the learned encoder $\widetilde{\mathcal{E}}_{\mathcal{X}}$ to be linear, which is not needed in our setup. Moreover, both approaches usually only deal with the case where both $\mathcal{X}$ and $\mathcal{Y}$ are Hilbert spaces.

**Example 2.3 (Parametric PDEs)** A common operator learning problem involves learning the map

$$F : a \in \mathcal{X} \mapsto u(a) \in \mathcal{Y}, \quad \text{where } u(a) \text{ satisfies } \mathcal{F}_a u = 0 \quad (2.6)$$

and $\mathcal{F}_a$ specifies a certain PDE depending on a parameter or function $a$. A standard example is the elliptic diffusion equation over a domain $\Omega \subset \mathbb{R}^n$. Here $a = a(\boldsymbol{x}) \in \mathrm{L}^{\infty}(\Omega) =: \mathcal{X}$ is the diffusion coefficient and $u = u(\cdot; a)$ is the solution of the PDE

$$-\nabla \cdot (a \nabla u(\boldsymbol{z}; a)) = g,\ \boldsymbol{z} \in \Omega, \quad u(\boldsymbol{z}; a) = 0,\ \boldsymbol{z} \in \partial\Omega. \quad (2.7)$$

Problems such as (2.6) are ubiquitous in scientific computing, with many applications in engineering, biology, physics, finance and beyond. In many such applications, it is common to assume that the measure $\mu$ on $\mathcal{X}$ is the law of a random field

$$a(\boldsymbol{x}) = a(\cdot; \boldsymbol{x}) = a_0(\cdot) + \sum_{i=1}^{\infty} c_i x_i \phi_i(\cdot), \qquad (2.8)$$

for functions $a_0, \phi_i \in \mathcal{X}$, where the $x_i$ are random variables and $c_i \geq 0$ are scalars that ensure that $a \in \mathrm{L}^{\infty}(\Omega)$. Under some mild assumptions, (2.8) is then the Karhunen–Loève (KL) expansion of the measure $\mu$. See, e.g., [91] (see also [55, §3.5.1]). The $x_i$ are typically independent. While in some settings, they may have infinite support, it is also common in practice to assume they range between finite maxima and minima. After rescaling, one may therefore assume that $\boldsymbol{x} \in D = [-1, 1]^{\mathbb{N}}$.

Problems of this type fits into our framework. Suppose that $\boldsymbol{x} = (x_1, x_2, \ldots) \sim \varrho$. The measure $\mu$ is then given as the pushforward $\mu = a\sharp\varrho$ and $f : D \to \mathcal{Y}$ is the parametric solution map $f : \boldsymbol{x} \in D \mapsto u(\cdot; a(\boldsymbol{x})) \in \mathcal{Y}$. If needed, the map $\iota$ can be defined in a number of different ways. Suppose, for instance, that $\mathcal{X}$ is a Hilbert space, e.g., $\mathcal{X} = \mathrm{L}^2(\Omega)$, and $\{\phi_i\}_{i=1}^{\infty}$ is a Riesz system (this holds, for instance, in the case of a KL expansion, in which case $\{\phi_i\}_{i=1}^{\infty}$ is an orthonormal basis). Then $\{\phi_i\}_{i=1}^{\infty}$ has a unique biorthogonal dual Riesz system $\{\psi_i\}_{i=1}^{\infty}$. We may therefore define $\iota : a \mapsto \left(\langle a - a_0, \psi_i \rangle_{\mathrm{L}^2(\Omega)} / c_i\right)_{i=1}^{\infty}$. Notice that $\iota$ is a bounded linear map and $F(X) = f \circ \iota(X) = f(\boldsymbol{x})$ for $X = a(\boldsymbol{x}) \sim \mu$. However, evaluating $\iota$ is often not required for computations (see §A.1).

This example considers an affine parametrization (2.8) inducing the measure $\mu$. Note that other parametrizations can be considered. Common examples include the *quadratic* $a(z; \boldsymbol{x}) = a_0(z) + \left(\sum_{i=1}^{\infty} c_i x_i \phi_i(z)\right)^2$ and *log-transformed* $a(z, \boldsymbol{x}) = \exp\left(\sum_{i=1}^{\infty} c_i x_i \phi_i(z)\right)$ parametrizations [17].

**Remark 2.4 (Holomorphy assumption)** In the previous example, the operator $F$ stems from the solution map $f : D \to \mathcal{Y}$ of a parametric PDE. The regularity of solution maps of parametric PDEs has been intensively studied, and it is known that many such maps are $(\boldsymbol{b}, \varepsilon)$-holomorphic (hence the resulting operator satisfies (A.II)). Consider, for instance, the affine diffusion problem (2.7)-(2.8). Under a mild *uniform ellipticity* condition, the solution map of the standard weak form of the PDE $f : \boldsymbol{x} \in D \mapsto u(a(\cdot; \boldsymbol{x})) \in \mathrm{H}_0^1(\Omega)$ is $(\boldsymbol{b}, \varepsilon)$-holomorphic with $\boldsymbol{b} = (b_i)_{i=1}^{\infty}$ and $b_i = c_i \|\phi_i\|_{\mathrm{L}^{\infty}(\Omega)}$. See, e.g., [3, Prop. 4.9], as well as §B.3. Similar results are known for other parametric PDEs. This includes parabolic PDEs, various types of nonlinear, elliptic PDEs, PDEs over parametrized domains, parametric hyperbolic problems and parametric control problems. See [19] or [3, Chpt. 4] for reviews.

## 3  Main results I: upper bounds

We now present our first two main results. In these results, given an optimization problem $\min_t f(t)$, we say that $\hat{t}$ is a *$\tau$-approximate minimizer* for some $\tau \geq 0$ if $f(\hat{t}) \leq \min_t f(t) + \tau^2$.

**Theorem 3.1** (Existence of good DNN architectures)**.** *Let $m \geq 3$, $\delta > 0$, $0 < \epsilon < 1$ and $L = L(m, \epsilon) = \log^4(m) + \log(1/\epsilon)$. Then there exists a class $\mathcal{N}$ of hyperbolic tangent (tanh) DNNs $N : \mathbb{R}^{d_{\mathcal{X}}} \to \mathbb{R}^{d_{\mathcal{Y}}}$ depending on $m$ and $\epsilon$ only with*

$$\mathrm{width}(\mathcal{N}) \lesssim (m/L)^{1+\delta}, \quad \mathrm{depth}(\mathcal{N}) \lesssim \log(m/L), \qquad (3.1)$$

*such that following holds. Suppose that Assumption 2.2 holds and*

$$d_{\mathcal{X}} \geq \lceil m/L \rceil, \qquad L_\iota \cdot \|\mathcal{I}_{\mathcal{X}} - \widetilde{\mathcal{D}}_{\mathcal{X}} \circ \widetilde{\mathcal{E}}_{\mathcal{X}}\|_{L^2_\mu(\mathcal{X}; \mathcal{X})} \leq c \cdot (m/L)^{-1/2}, \qquad (3.2)$$

*where $\mathcal{I}_{\mathcal{X}} : \mathcal{X} \to \mathcal{X}$ is the identity map and $c > 0$ is a universal constant. Let $X_1, \ldots, X_m \sim_{\text{i.i.d.}} \mu$ and consider the noisy training data (2.4) with arbitrary noise $E_i \in \mathcal{Y}$. Then, with probability at least $1 - \epsilon$, every $\tau$-minimizer $\widehat{N}$ of (2.5), where $\tau \geq 0$ is arbitrary, yields an approximation $\widehat{F}$ that satisfies*

$$\|F - \widehat{F}\|_{L^2_\mu(\mathcal{X}; \mathcal{Y})} \lesssim E_{\text{app},2} + E_{\mathcal{X},2} + E_{\mathcal{Y},2} + E_{\text{opt},2} + E_{\text{samp},2}, \qquad (3.3)$$

$$\|F - \widehat{F}\|_{L^{\infty}_\mu(\mathcal{X}; \mathcal{Y})} \lesssim E_{\text{app},\infty} + E_{\mathcal{X},\infty} + E_{\mathcal{Y},\infty} + E_{\text{opt},\infty} + E_{\text{samp},\infty}, \qquad (3.4)$$

*and, if $\mathcal{Y}$ is a Hilbert space,*

$$\|F - \widehat{F}\|_{L^2_\mu(\mathcal{X}; \mathcal{Y})} \lesssim E_{\text{app},2} + E_{\mathcal{X},2} + E_{\mathcal{Y},2} + E_{\text{opt},2} + E_{\text{samp},2}. \qquad (3.5)$$

*Here, the approximation error terms $E_{\text{app},q}$, $q = 2, \infty$, are given by*

$$E_{\text{app},q} = a_{\mathcal{Y}} \cdot C(\boldsymbol{b}, p, \xi) \cdot (m/L)^{\theta + 1 - 1/q - 1/p}, \tag{3.6}$$

*where $a_{\mathcal{Y}} = \|\mathcal{D}_{\mathcal{Y}} \circ \mathcal{E}_{\mathcal{Y}}\|_{\mathcal{Y} \to \mathcal{Y}}$, $C(\boldsymbol{b}, p, \xi) > 0$ depends on $\boldsymbol{b}$, $p$ and $\xi$ only and $\theta = 0$ if $\mathcal{Y}$ is a Hilbert space (as in (3.5)) or $\theta = 1/2$ otherwise (as in (3.3)-(3.4)). The other terms are given by*

$$E_{\mathcal{X},2} = a_{\mathcal{Y}} \cdot L_\iota \cdot \sqrt{m/(L\epsilon)} \cdot \|\mathcal{I}_{\mathcal{X}} - \widetilde{\mathcal{D}}_{\mathcal{X}} \circ \widetilde{\mathcal{E}}_{\mathcal{X}}\|_{L^2_\mu(\mathcal{X};\mathcal{X})}$$

$$E_{\mathcal{X},\infty} = a_{\mathcal{Y}} \cdot L_\iota \cdot \sqrt{m/L} \cdot \left( \sqrt{m/L} \cdot \|\mathcal{I}_{\mathcal{X}} - \widetilde{\mathcal{D}}_{\mathcal{X}} \circ \widetilde{\mathcal{E}}_{\mathcal{X}}\|_{L^2_\mu(\mathcal{X};\mathcal{X})} + \|\mathcal{I}_{\mathcal{X}} - \widetilde{\mathcal{D}}_{\mathcal{X}} \circ \widetilde{\mathcal{E}}_{\mathcal{X}}\|_{L^\infty_\mu(\mathcal{X};\mathcal{X})} \right)$$

$$E_{\mathcal{Y},2} = \|\mathcal{I}_{\mathcal{Y}} - \mathcal{D}_{\mathcal{Y}} \circ \mathcal{E}_{\mathcal{Y}}\|_{L^2_{F \sharp \mu}(\mathcal{Y};\mathcal{Y})} / \sqrt{\epsilon}$$

$$E_{\mathcal{Y},\infty} = \|\mathcal{I}_{\mathcal{Y}} - \mathcal{D}_{\mathcal{Y}} \circ \mathcal{E}_{\mathcal{Y}}\|_{L^\infty_{F \sharp \mu}(\mathcal{Y};\mathcal{Y})} + \sqrt{m/L} \cdot \|\mathcal{I}_{\mathcal{Y}} - \mathcal{D}_{\mathcal{Y}} \circ \mathcal{E}_{\mathcal{Y}}\|_{L^2_{F \sharp \mu}(\mathcal{Y};\mathcal{Y})},$$

$$\tag{3.7}$$

*where $\mathcal{I}_{\mathcal{Y}} : \mathcal{Y} \to \mathcal{Y}$ is the identity map and, if $\|\boldsymbol{E}\|^2_{2;\mathcal{Y}} = \sum_{i=1}^m \|E_i\|^2_{\mathcal{Y}}$,*

$$E_{\text{opt}} = \begin{cases} \tau + 2^{-m} & q = 2 \\ \sqrt{m/L}\tau + 2^{-m} & q = \infty \end{cases}, \quad E_{\text{samp},q} = \begin{cases} \|\boldsymbol{E}\|_{2;\mathcal{Y}}/\sqrt{m} & q = 2 \\ \|\boldsymbol{E}\|_{2;\mathcal{Y}}/\sqrt{L} & q = \infty \end{cases}. \tag{3.8}$$

(Proofs of this and all other theorems are in §C-G of the supplemental material.) This theorem shows that there is a family of tanh DNNs that yield provable bounds for learning holomorphic operators. The error (3.3)-(3.5) decomposes into an *approximation error* (3.6), which decays algebraically in the amount of training data $m$. Later, in Theorems 4.1-4.2, we show that these rates are optimal when $\mathcal{Y}$ is a Hilbert space, up to log factors. Next, are the *encoding-decoding errors* (3.7), which depend on how well the approximate encoder-decoder pairs $(\widetilde{\mathcal{E}}_{\mathcal{X}}, \widetilde{\mathcal{D}}_{\mathcal{X}})$ and $(\mathcal{E}_{\mathcal{Y}}, \mathcal{D}_{\mathcal{Y}})$ approximate the identity maps on $\mathcal{X}$ and $\mathcal{Y}$, respectively. Observe that these terms are increasing in $m$ for fixed encoders and decoders. Therefore, as one expects, the accuracy of the encoder-decoder approximations $\widetilde{\mathcal{D}}_{\mathcal{X}} \circ \widetilde{\mathcal{E}}_{\mathcal{X}} \approx \mathcal{I}_{\mathcal{X}}$ and $\mathcal{D}_{\mathcal{Y}} \circ \mathcal{E}_{\mathcal{Y}} \approx \mathcal{I}_{\mathcal{Y}}$ should increase with increasing $m$ to ensure decay to zero of the generalization error as $m \to \infty$. The specific terms in (3.7) (for $q = 2$) are quite standard in operator learning. See, e.g., [53, 55]. When the encoders and decoders are computed via PCA, as in PCA-Net, standard bounds can be derived for these terms [53]. For similar analysis in the case of DeepONets, see [55]. Finally, the error (3.3)-(3.5) involves an *optimization error* $E_{\text{opt}}$, which primarily depends on how accurately the optimization problem (2.5) is solved (i.e., the term $\tau$), and a *sampling error* $E_{\text{samp}}$, which depends on the error in the training data (2.4).

Theorem 3.1 allows $\mathcal{Y}$ to be a Banach or a Hilbert space. Overall, when $\mathcal{Y}$ is only a Banach space, we obtain a weaker $L^2_\mu$-norm bound involving the Pettis norm (3.3) and, moreover, the approximation error $E_{\text{app},q}$ is worse by a factor of $1/2$ than when $\mathcal{Y}$ is a Hilbert space. (Note that one can establish a bound for the Bochner $L^2_\mu$-norm error when $\mathcal{Y}$ is a Banach space via (3.4) and the inequality $\|\cdot\|_{L^2_\mu(\mathcal{X};\mathcal{Y})} \leq \|\cdot\|_{L^\infty_\mu(\mathcal{X};\mathcal{Y})}$. However, we do not believe the resulting bound is sharp). As we discuss in Remark D.18, the discrepancies between the two cases stem from the lack of an inner product structure and, in particular, the absence of Parseval's identity when $\mathcal{Y}$ is a Banach space.

Observe that the DNN architecture in Theorem 3.1 is independent of the smoothness of the operator being learned. We term such an architecture *problem agnostic*. This theorem considers tanh activations only. However, as we discuss in Remark D.11, other activations can be readily used instead. Other key facets of Theorem 3.1 are the width and depth bounds (3.1). Qualitatively, these agree with empirical practice: namely, better performing DNNs tend to be wider than they are deep, and relatively shallow DNNs perform well in practice (see [24, 25] and references therein). We also see this later in §5.

On the other hand, the family $\mathcal{N}$ is not fully connected. As we describe in §C.2.1, while the weights on the final layer can be arbitrary real numbers, the weights and biases in the hidden layers come from a finite (but large) set: they are *handcrafted* to approximately emulate certain multivariate orthogonal polynomials. Since fully-connected DNNs are typically used in practice, Theorem 3.1 is essentially a theoretical contribution. In our next result, we consider the more practical scenario of fully-connected DNNs.

**Theorem 3.2** (Fully-connected DNN architectures are good)**.** *There are universal constants $c_1, c_2, c_3, c_4 \geq 1$ such that the following holds. Let $m$, $\delta$, $\epsilon$ and $L$ be as in Theorem 3.1,*

$$d_{\mathcal{X}} \geq c_1(m + \log(1/\epsilon)), \quad L_\iota \cdot \|\mathcal{I}_{\mathcal{X}} - \widetilde{\mathcal{D}}_{\mathcal{X}} \circ \widetilde{\mathcal{E}}_{\mathcal{X}}\|_{L^2_\mu(\mathcal{X};\mathcal{X})} \leq c(\delta) \cdot (m + \log(1/\epsilon))^{-1/2}, \tag{3.9}$$

*where $c(\delta) > 0$ depends on $\delta$ only, consider any class $\mathcal{N}$ of fully-connected DNNs satisfying*

$$(n_0, n_{L+2}) = (d_{\mathcal{X}}, d_{\mathcal{Y}}), \quad N_1, \ldots, N_{L+1} \geq c_2 \cdot (m + \log(1/\epsilon)) \cdot (m/L)^{\delta}, \quad L \geq c_3 \cdot \log(m/L). \tag{3.10}$$

*Suppose that Assumption 2.2 holds and that the pushforward $\varsigma$ in (A.I) is the tensor-product of a univariate probability distribution with mean zero and variance $\omega \gtrsim 1$. Let $X_1, \ldots, X_m \sim_{\text{i.i.d.}} \mu$ and consider (2.4) with arbitrary $E_i \in \mathcal{Y}$. Then the following hold with probability at least $1 - \epsilon$.*

*(A)* ***Uncountably many 'good' minimizers.*** *The problem (2.5) has uncountably many minimizers that satisfy (3.3) with $\tau = 0$ or (3.5) with $\tau = 0$ if $\mathcal{Y}$ is a Hilbert space. They also satisfy (3.4) with $\tau = 0$ and the modified right-hand side $\sqrt{L}E_{\text{app},\infty} + LE_{\mathcal{X},\infty} + \sqrt{L}E_{\mathcal{Y},\infty} + E_{\text{opt},\infty} + \sqrt{L}E_{\text{samp},\infty}$.*

*(B)* ***Good minimizers are stable.*** *Suppose that $\mathcal{E}_{\mathcal{X}} \in L_{\mu}^{\infty}(\mathcal{X}; \mathbb{R}^{d_{\mathcal{X}}})$ and let $\tau_o > 0$ be arbitrary. Then there is a neighbourhood of DNN parameters around the parameters of each minimizer in (A) for which the approximation corresponding to any parameters in this neigbourhood also satisfies the same bounds as in (A) with $\tau = \tau_o$.*

*(C)* ***Good minimizers can be far apart in parameter space.*** *For sufficiently large $m$, there are at least $(m/(c_4 L))^{2\delta m}$ minimizers satisfying the bounds in (A) such that, for any two such minimizers, their parameters satisfy $\|\boldsymbol{\theta}'\| = \|\boldsymbol{\theta}\|$ and $\|\boldsymbol{\theta}' - \boldsymbol{\theta}\| \gtrsim 1$.*

This theorem states that DL with fully-connected DNN architectures of sufficient width and depth (3.10) can succeed, since there are minimizers that yield the optimal bounds of Theorem 3.1. Such minimizers are uncountably many in number (A), stable to perturbations (B) and many of them (exponentially in $m$) have sufficiently distinct and nonvanishing/nonexploding parameters (C). This theorem does not imply that *all* minimizers are 'good' – an issue we discuss further in §6 – but our numerical results in §5 suggest that (approximate) minimizers obtained through training do, at least for the experiments considered, achieve the rates specified in Theorem 3.1.

## 4 Main results II: lower bounds

We now show that the various approximation errors are nearly optimal. For this, we ignore the encoding-decoding, optimization and sampling errors and proceed as follows. Let $C(\mathcal{X}; \mathcal{Y})$ be the Banach space of continuous operators. We term an *(adaptive) sampling map* as any map

$$\mathcal{L} : C(\mathcal{X}, \mathcal{Y}) \to \mathcal{Y}^m, \quad F \mapsto \mathcal{L}(F) = (F(X_i))_{i=1}^m, \tag{4.1}$$

where $X_1 \in \mathcal{X}$, $X_2 = X_2(F(X_1)) \in \mathcal{X}$ potentially depends on the previous evaluation $F(X_1)$, $X_3 = X_3(F(X_1), F(X_2)) \in \mathcal{X}$, and so forth. Next, we term a *reconstruction map* as any map $\mathcal{R} : \mathcal{Y}^m \to L_{\mu}^2(\mathcal{X}; \mathcal{Y})$. Given this, we let $\mathcal{H}(\boldsymbol{b}, \iota) = \{F = f \circ \iota : f \in \mathcal{H}(\boldsymbol{b})\}$ and define

$$\theta_m(\boldsymbol{b}) = \inf_{\mathcal{L}, \mathcal{R}} \sup_{F \in \mathcal{H}(\boldsymbol{b}, \iota)} \||F - \mathcal{R} \circ \mathcal{L}(F)\||_{L_{\mu}^2(\mathcal{X}; \mathcal{Y})}, \tag{4.2}$$

where the infimum is taken over all such $\mathcal{L}$ and $\mathcal{R}$. In other words, $\theta_m(\boldsymbol{b})$ measures how well one can learn holomorphic operators using *arbitrary* training data and an *arbitrary* reconstruction procedure.

**Theorem 4.1** (Optimal $L^2$ error rates). *Suppose that (A.I) holds. Then, for any $0 < p < 1$ there is a constant $c(p) > 0$ such that the following hold.*

*(i) For each $m \in \mathbb{N}$, there is a $\boldsymbol{b} \in \ell_{\text{M}}^p(\mathbb{N})$, $\boldsymbol{b} \geq \boldsymbol{0}$, $\|\boldsymbol{b}\|_{p,\text{M}} = 1$ such that $\theta_m(\boldsymbol{b}) \geq c(p) \cdot m^{1/2 - 1/p}$.*

*(ii) There is a $\boldsymbol{b} \in \ell_{\text{M}}^p(\mathbb{N})$, $\boldsymbol{b} \geq \boldsymbol{0}$, $\|\boldsymbol{b}\|_{p,\text{M}} = 1$ such that $\theta_m(\boldsymbol{b}) \geq c(p) \cdot \frac{m^{1/2 - 1/p}}{\log^{2/p}(2m)}, \forall m \in \mathbb{N}$.*

This theorem shows that the error $E_{\text{app},2}$ in Theorems 3.1-3.2 is optimal, up to log terms, whenever $\mathcal{Y}$ is a Hilbert space: there does not exist a reconstruction map surpasses the rate $m^{1/2 - 1/p}$ for learning holomorphic operators. Note that this result applies not only to DL-based procedures, but *any* procedure that learns such operators from $m$ samples. Another consequence of this result is that adaptive sampling, i.e., *active learning*, is of no benefit. As shown by Theorems 3.1-3.2, the optimal rate $m^{1/2 - 1/p}$ can, up to log terms, be achieved through inactive learning, i.e., i.i.d. sampling from $\mu$.

Theorem 4.1 considers $L^2$-norm. For the $L^{\infty}$-norm, we present a somewhat weaker result. Let

$$\tilde{\theta}_m(\boldsymbol{b}) = \inf_{\mathcal{R}} \{\mathbb{E}_{X_1, \ldots, X_m \sim \mu} \sup_{F \in \mathcal{H}(\boldsymbol{b}, \iota)} \|F - \mathcal{R}(\{X_i, F(X_i)\})\|_{L_{\mu}^{\infty}(\mathcal{X}; \mathcal{Y})}\}, \tag{4.3}$$

where the infimum is taken over all reconstruction maps $\mathcal{R} : (\mathcal{X} \times \mathcal{Y})^m \to L^\infty_\mu(\mathcal{X}; \mathcal{Y})$ only.

**Theorem 4.2** (Optimal $L^\infty$ error rates). *Suppose that (A.I) holds and that the pushforward $\varsigma$ is the tensor-product of a univariate probability distribution with mean zero and variance $\omega \gtrsim 1$. Then, for any $0 < p < 1$ there is a constant $c(p) > 0$ such that the following hold.*

*(i) For each $m \in \mathbb{N}$, there is a $\boldsymbol{b} \in \ell^p_\mathsf{M}(\mathbb{N})$, $\boldsymbol{b} \geq \boldsymbol{0}$, $\|\boldsymbol{b}\|_{p,\mathsf{M}} = 1$ such that $\tilde{\theta}_m(\boldsymbol{b}) \geq c(p) \cdot \frac{m^{1-1/p}}{\log(m)}$.*

*(ii) There is a $\boldsymbol{b} \in \ell^p_\mathsf{M}(\mathbb{N})$, $\boldsymbol{b} \geq \boldsymbol{0}$, $\|\boldsymbol{b}\|_{p,\mathsf{M}} = 1$ such that $\tilde{\theta}_m(\boldsymbol{b}) \geq c(p) \cdot \frac{m^{1-1/p}}{\log^{2/p+1}(2m)}$, $\forall m \in \mathbb{N}$.*

As with the previous theorem, this result asserts that the rate $m^{1-1/p}$ is optimal in the $L^\infty$-norm when $\mathcal{Y}$ is a Hilbert space. However, it is strictly weaker than Theorem 4.1 as it only considers i.i.d. random sampling from $\mu$, as opposed to arbitrary (adaptive) samples. Note that Theorem 4.1 is an extension of [7, Thm. 4.4]. Theorem 4.2 is new, and is of independent interest since it partially addresses an open problem of [7] about deriving lower bounds in the $L^\infty_\mu$-norm, as opposed to just the $L^2_\mu$-norm. See §C.2.3-C.2.4 for more discussion.

## 5 Numerical experiments

We now present numerical results for DL applied to various different parametric PDE problems, as in Example 2.3. For a full description of our experimental setup, see §A-B.

Since the main objective of this work is to examine the approximation error, we follow a standard setup and fix the encoder and decoders for each experiment, so that $\mathcal{E}_\mathcal{X}$ and $\mathcal{D}_\mathcal{Y}$ in (1.3) do not change for different choices of $\widehat{N}$. We also set up our experiments so that encoding-decoding (3.7) and sampling (3.8) errors are zero. We do this in a standard way. To ensure that $E_{\mathcal{X},q} = 0$, we truncate the parametric expansions (2.8) after $d$ terms (henceforth termed the *parametric dimension*) and define the encoder $\mathcal{E}_\mathcal{X}$ accordingly. This means we effectively consider a parametric PDE depending on finitely-many parameters. We use Finite Element Methods (FEMs) to both solve the PDE (for generating training and testing data) and define the decoder $\mathcal{D}_\mathcal{Y}$ (see (A.2)). To ensure that $E_{\mathcal{Y},q} = 0$, we compute errors with respect to the Bochner $L^2_\mu(\mathcal{X}; \widetilde{\mathcal{Y}})$-norm, where $\widetilde{\mathcal{Y}} = \mathcal{D}_\mathcal{Y}(\mathbb{R}^{d_\mathcal{Y}})$ is the FEM discretization of $\mathcal{Y}$. In other words, we use the same FEM code to generate test data and compute the errors as we do to construct the operator approximation $\widehat{F}$. See §A.1 for further details.

The DNNs in our experiments are fully-connected and of the form (2.1). We denote by $\sigma$ $L \times N$ DNN a DNN $\widehat{N}$ with activation function $\sigma$, width $N$ and depth $L$. To solve (2.5) we use Adam [47] with early stopping and an exponentially decaying learning rate. We train our DNN architectures for 60,000 epochs and results are averaged over a number of trials. See §A.2 for further details.

**Parametric elliptic diffusion equation.** Our first example is the parametric elliptic diffusion equation (2.7). This PDE arises in many scientific computing applications, such as groundwater flow modelling, see, e.g., [95]. We describe the full PDE and its FE discretization in §B.3. In our experiments, we consider both affine (B.1) and log-transformed (B.2) diffusion coefficients. The latter is particularly useful in the groundwater flow problem as the permeability of various layers of sediment can vary on logarithmic scales. Differing from most prior work, we consider a novel *mixed variational formulation* [32] of (2.7), which has a number of key practical benefits (see §B.3.1). In this case, $\mathcal{Y} = \mathrm{L}^2(\Omega)$ is a Hilbert space. Fig. 1 compares the error versus the amount of training data $m$ for various DNN architectures for learning the solution map of this PDE in $d = 4$ and $d = 8$ parametric dimensions with these two diffusion coefficients. We observe that architectures with the Exponential Linear Unit (ELU) or hyperbolic tangent (tanh) activation generally outperform similar architectures with the Rectified Linear Unit (ReLU) activation (as we discuss in Remark D.11, this difference is in agreement with our theoretical analysis). Overall, the best performing DNNs appear to roughly match the plotted rate $m^{-1}$. As we explain further in §B.3.2, this rate is precisely that predicted by our theory. In particular, the parametric solution map (recall Remark 2.4) is $(\boldsymbol{b}, \varepsilon)$-holomorphic with $\boldsymbol{b} \in \ell^p_\mathsf{M}(\mathbb{N})$ for any $p < 2/3$, giving an effective convergent rate $m^{1/2-1/p}$ that is arbitrarily close to $m^{-1}$. Another important fact that we observe is that despite the parametric dimension doubling from 4 to 8, there is little change in the error behaviour.

**Parametric Navier-Stokes-Brinkman equations.** We next consider the parametric Navier-Stokes-Brinkman (NSB) equations. See §B.4 and (B.14) for the full definition. Here the solution is a pair

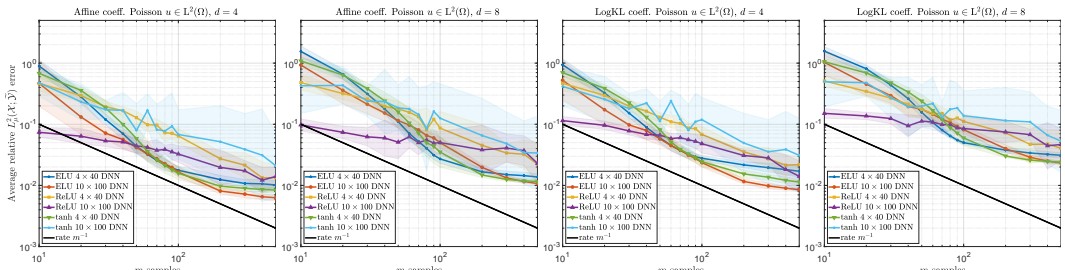

Figure 1: **Elliptic diffusion equation.** Average relative $L^2_\mu(\mathcal{X}; \widetilde{\mathcal{Y}})$-norm error versus $m$ for different DNNs approximating the solution operator for the elliptic diffusion equation (B.9). The first two plots use the affine coefficient $a_{1,d}$ (B.1) with $d = 4, 8$, respectively. The rest use the log-transformed coefficient $a_{2,d}$ (B.2).

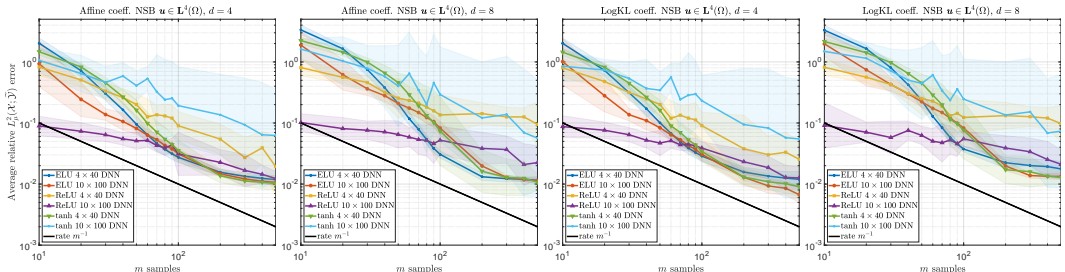

Figure 2: **NSB equations.** Average relative $L^2_\mu(\mathcal{X}; \widetilde{\mathcal{Y}})$-norm error versus $m$ for different DNNs approximating the velocity field $\boldsymbol{u}$ of the NSB problem in (B.14). See Fig. 7 for results for the pressure component $p$. The diffusion coefficients $a_{1,d}, a_{2,d}$ and $d = 4, 8$ are as in Fig. 1.

$(\boldsymbol{u}, p)$, where $\boldsymbol{u}$ is the velocity field and $p$ is the pressure. These equations describe the dynamics of a viscous fluid flowing through porous media with random viscosity. See, e.g., [28, 43, 46, 94]. We use a mixed variational formulation [34] to discretize the PDE. This formulation is more sophisticated that standard variational formulations, but conveys various practical advantages. Unlike the previous example, it leads to $\mathcal{Y}$ being either $\mathcal{Y} = \mathbf{L}^4(\Omega)$ for $\boldsymbol{u}$ or $\mathcal{Y} = L^2(\Omega)$ for $p$. See §B.4.1 for details. Fig. 2 compares a variety of DNN architectures for approximating the velocity field component in $d = 4$ and $d = 8$ parametric dimensions. Here again we observe the ELU and tanh DNN architectures outperform similar sized ReLU architectures. We also observe a rate close to $m^{-1}$. Note that it is currently unknown whether this or the next example possess the same $(\boldsymbol{b}, \varepsilon)$-holomorphy guarantee as that of the previous example. Yet we observe the same rate, and therefore conjecture that such a property does indeed hold in these cases. Similar to the previous example, there is also no deterioration of the rate when moving from $d = 4$ to $d = 8$.

**Parametric stationary Boussinesq equation.** Our final example is a parametric stationary Boussinesq PDE. See §B.5 and (B.16) for the full definition. Here the solution is a triplet $(\boldsymbol{u}, \varphi, p)$, where $\boldsymbol{u}$ is the velocity field, $\varphi$ is the temperature and $p$ is the pressure of the solution. The Boussinesq model arises in a variety of engineering, fluid dynamics and natural convection problems where changes in temperature affect the velocity of a fluid [14, 22, 39]. Similar to the previous example, we consider a fully mixed variational formulation (see §B.5.1), which leads to $\mathcal{Y} = \mathbf{L}^4(\Omega)$ (for $\boldsymbol{u}$), $\mathcal{Y} = L^4(\Omega)$ (for $\varphi$) or $\mathcal{Y} = L^2_0(\Omega)$ (for $p$). Fig. 3 provides numerical results. Our observations are in line with the previous two examples, with the ELU and the smaller tanh networks being most often the best performers in this problem. Once more, the errors roughly correspond to the rate $m^{-1}$ and there is no deterioration with increasing $d$.

## 6 Conclusions and limitations

The purpose of this work was to derive near-optimal generalization bounds for learning certain classes of holomorphic operators that arise frequently in operator learning tasks involving PDEs. Complementing and extending previous works [26, 31, 41, 55, 71] on the approximation of such operators via DNNs, we showing sharp algebraic rates of convergence in $m$, thus confirming that

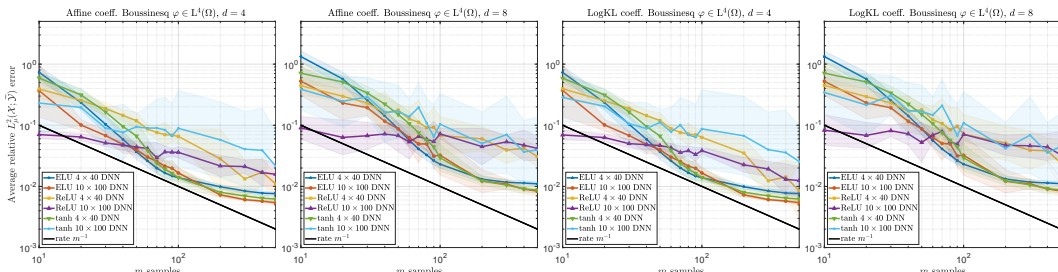

Figure 3: **Boussinesq equation.** Average relative $L_\mu^2(\mathcal{X}; \widetilde{\mathcal{Y}})$-norm error versus $m$ for different DNNs approximating the temperature $\varphi$ of the Boussinesq problem in (B.16) (see Fig. 9 for $\boldsymbol{u}$ and $p$). The diffusion coefficients $a_{1,d}, a_{2,d}$ and $d = 4, 8$ are as in Fig. 1. In this example, we also consider an additional parametric dependence in the tensor $\mathbb{K} = \mathbb{K}_d$ describing the thermal conductivity of the fluid. See §B.5 and (B.17).

such operators can be learned efficiently and without the *curse of dimensionality*. It is notable that the sizes of the various DNNs in Theorems 3.1-3.2 also do not succumb to the so-called *curse of parametric complexity* [54], since the width and depth bounds are at most algebraic in $m$.

We end by discussing a number of limitations. First, assumption (A.I) may not hold in some applications. The domain $D$ can easily be replaced by bounded hyperrectangle through rescaling and the condition that $\varsigma$ be quasi-uniform relaxed to quasi-ultraspherical (by considering ultraspherical polynomials). However, it is currently an open problem whether our results can be extended to the case where $\mu$ is Gaussian, in which case $\varsigma$ would typically be a tensor-product Gaussian measure on $\mathbb{R}^{\mathbb{N}}$ and the relevant polynomials would be the Hermite polynomials. Second, the reader may have noticed that the encoder $\mathcal{E}_\mathcal{X}$ defined in (A.III) and used to construct the approximation (2.5) involves the pair $(\widetilde{\mathcal{E}}_\mathcal{X}, \widetilde{\mathcal{D}}_\mathcal{X})$ and the map $\iota_{d_\mathcal{X}}$. This is a technical requirement – also found in other theoretical works on operator learning – needed to obtain encoding-decoding errors of the form $E_{\mathcal{X},q}$, $q = 2, \infty$. It is unknown whether it can be relaxed. It is also unknown whether the assumption on $\tilde{\varsigma}$ in (A.III) can be relaxed. We believe this can be done, at least if the $L_\mu^2$-norm in (3.2) is replaced by the $L_\mu^\infty$-norm. Whether this is possible without modifying (3.2) is currently unknown.

Third, a limitation of Theorem 3.2 is that it only asserts that some minimizers are 'good', not all. Techniques from statistical learning theory can provide stronger bounds that hold for all minimizers. Yet, as noted in §1.2, these tools typically produce slower rates of decay in $m$. Overcoming this limitation – e.g., by refining these tools for the holomorphic setting or showing that the 'good' minimizers can indeed be obtained via standard training – is a topic of future work.

Finally, as noted, our theorems provided worse generalization bounds when $\mathcal{Y}$ is a Banach space than when $\mathcal{Y}$ is a Hilbert space. Our numerical results in Figs. 2-3 suggest that this factor is an artefact of the proofs. Whether it can be removed is an interesting open problem.

## Acknowledgments and Disclosure of Funding

BA acknowledges the support of the Natural Sciences and Engineering Research Council of Canada of Canada (NSERC) through grant RGPIN-2021-611675. ND acknowledges the support of Florida State University through the CRC 2022-2023 FYAP grant program. The authors would like to thank Gregor Maier for helpful comments and feedback.

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

## A Experimental setup

In this section, we describe our experimental setup.

### A.1 Formulation of the learning problems

We first explain how all our experiments are formulated as operator learning problems. We do this in a standard way, by first truncating the random field which generates the measure $\mu$, and then using an FEM to discretize the output space.

All our examples follow Example (2.3) and involve operators of the form (2.6), where $\mathcal{F}_a$ represents a different type of PDE in each case (specifically, either an elliptic diffusion, Navier-Stokes-Brinkman or Boussinesq PDE). We consider both affine (2.8) and log-transformed parametrizations of the random field $a(\boldsymbol{x})$. Thus, in general we write

$$a(\boldsymbol{x}) = a(\cdot; \boldsymbol{x}) = g\left(a_0(\cdot) + \sum_{i=1}^{\infty} c_i x_i \phi_i(\cdot)\right), \tag{A.1}$$

where $g : \mathbb{R} \to \mathbb{R}$ is a (measurable) map. In the affine case $g(t) = 1$. In the log-transformed case $g(t) = \exp(t)$.

As discussed, since the main objective of this work is to examine the approximation error, we set up our experiments so that encoding-decoding errors are zero. We do this as follows. First, we fix a parametric dimension $d$ and truncate the expansion in (A.1) after $d$ terms, giving a map $a_d : [-1,1]^d \to \mathcal{X}$ and measure $\mu = a_d \sharp \varrho_d$, where $\varrho_d$ is the uniform probability measure on $[-1,1]^d$. We then define the operator $F$ as $F(a_d(\boldsymbol{x})) = f(\boldsymbol{x}) = u(\cdot; a_d(\boldsymbol{x}))$, where $u(\cdot; a)$ is the solution of the PDE $\mathcal{F}_a u = 0$.

In alignment with our theorems, we focus on the *in-distribution* performance of the learned approximation $\widehat{F}$. This means we define the encoder only on $\operatorname{supp}(\mu)$, as $\mathcal{E}_{\mathcal{X}}(X) = \boldsymbol{x}$ when $X = a_d(\boldsymbol{x})$ with $\boldsymbol{x} \in [-1,1]^d$. As a result, the encoding-decoding error $E_{\mathcal{X},q}$ in (3.7) satisfies $E_{\mathcal{X},q} = 0$.

To perform our simulations, we use FEMs to solve the PDE and discretize the output space $\mathcal{Y}$. Let $\{\varphi_i\}_{i=1}^K \subset \mathcal{Y}$ be a FEM basis and set $d_{\mathcal{Y}} = K$. Then we define the decoder as

$$\mathcal{D}_{\mathcal{Y}} : \mathbb{R}^K \to \mathcal{Y}, \quad \mathcal{D}_{\mathcal{Y}}(\boldsymbol{c}) = \sum_{i=1}^K c_i \varphi_i \tag{A.2}$$

and set $\widetilde{\mathcal{Y}} = \mathcal{D}_{\mathcal{Y}}(\mathbb{R}^K)$ as the discretization of $\mathcal{Y}$.

With this in hand, we now describe the simulation of training data in general terms. First, we draw $\boldsymbol{x}_1, \ldots, \boldsymbol{x}_m \sim_{\text{i.i.d.}} \varrho_d$. Then, for each training sample $\boldsymbol{x}_i$, we compute $Y_i$ by using the FEM to solve the PDE with parameter $X_i = a_d(\boldsymbol{x}_i)$. Notice that $Y_i \in \widetilde{\mathcal{Y}}$ in this setup.

We consider a DNN architecture with input dimension $n_0 = d$ and output dimension $n_{L+2} = K$. After training, we evaluate the learned approximation $\widehat{F} = \mathcal{D}_{\mathcal{Y}} \circ \widehat{N} \circ \mathcal{E}_{\mathcal{X}}$ over $\operatorname{supp}(\mu)$ as $\widehat{F}(X) = \mathcal{D}_{\mathcal{Y}} \circ \widehat{N} \circ \mathcal{E}_{\mathcal{X}}(X) = \mathcal{D}_{\mathcal{Y}} \circ \widehat{N}(\boldsymbol{x})$ for $X = a_d(\boldsymbol{x})$ with $\boldsymbol{x} \in [-1,1]^d$. Finally, as noted, we us the same FEM discretization to generate testing data, which allows us to measure the error with respect to the $L^2_\mu(\mathcal{X}; \widetilde{\mathcal{Y}})$-norm. This effectively means that the encoding-decoding error $E_{\mathcal{Y},q}$ in (3.7) satisfies $E_{\mathcal{Y},q} = 0$ as well.

### A.2 Computational setup for the numerical experiments

In this work, we investigate the trade-off between the accuracy of the learned operator and the number of samples $m$ used in training. Our methodology is summarized as follows.

(i) **Implementation**. We use the open-source finite element library `FEniCS`, specifically version 2019.1.0 [9], and Google's `TensorFlow` version 2.12.0. More information about `TensorFlow` can be found at `https://www.tensorflow.org/`.

(ii) **Hardware**. We train the DNN models in single precision on the Digital Research Alliance of Canada's Cedar compute cluster (see `https://docs.alliancecan.ca/wiki/Cedar`), using

Intel Xenon Processor E5-2683 v4 CPUs with either 125GB or 250GB per node. The setup for each of our PDEs is as follows. For each experiment we consider training with 14 sets of points of size $m \in \{10, 20, 30, 40, 50, 60, 70, 80, 90, 100, 200, 300, 400, 500\}$ and for 6 different architectures (4 x 40 and 10 x 100 with ReLU, ELU, and tanh activations) over two parametric dimensions ($d = 4$ and $d = 8$) and two coefficients ($a_{1,d}$ from (B.1) and $a_{2,d}$ from (B.2)), giving 336 DNNs to be trained for each trial. For the Poisson PDE and Navier-Stokes-Brinkman PDEs we run 12 trials. For the Boussinesq PDE we run 8 trials due to the larger problem size. For the Poisson PDE, we allocate 336 nodes with $1 \times 32$ core CPUs running 4 threads (totalling 1344 threads, 6.8 GB RAM per node) each running 3 trials, taking approximately 4 hours and 15 minutes to complete. For the Navier-Stokes-Brinkman PDE, we use the same setup allocating 9.88 GB RAM per node and the runs take approximately 9 hours and 13 minutes for each of the two components of the solution to complete. For the Boussinesq PDE, we allocate 336 nodes with $1 \times 32$ core CPUs running 4 threads per node (totaling 1344 threads, 10 GB RAM per node) each running 2 trials, taking approximately 12 hours and 32 minutes for each of the 3 components to complete. Given this, the total time required to reproduce the results in parallel with the above setup is approximately 60 hours or 2.5 days. Results were stored locally on the cluster and the estimated total space used to store the data for testing and training and results from computation is approximately 50 GB. Trained models were not retained due to space limitations on the cluster.

(iii) **Choice of architectures and initialization**. Based on the strategies in [1], we fix the number of nodes per layer $N$ and depth $L$ such that the ratio $\beta := L/N$ is $\beta = 0.5$. In addition, we initialize the weights and biases using the `HeUniform` initializer from `keras` setting the seed to the trial number. We consider the Rectified Linear Unit (ReLU)

$$\sigma_1(z) := \max\{0, z\},$$

hyperbolic tangent (tanh)

$$\sigma_2(z) := \frac{e^z - e^{-z}}{e^z + e^{-z}},$$

or Exponential Linear Unit (ELU)

$$\sigma_3(z) = \begin{cases} z & z > 0, \\ e^z - 1 & z \leq 0 \end{cases}$$

activation functions in our experiments.

(iv) **Optimizers for training and parametrization**. To train the DNNs, we use the `Adam` optimizer [47], incorporating an exponentially-decaying learning rate. We train our models for 60,000 epochs or until converging to a tolerance level of $\epsilon_{\text{tol}} = 5 \cdot 10^{-7}$ in single precision. In light of the nonmonotonic convergence behavior observed during the minimization of the nonconvex loss (see, e.g., [1, 2]), we implement early stopping. More precisely, we save the weights and biases of the partially trained network once the ratio between the current loss and the last checkpoint loss is reduced below $1/8$, or if the current weights and biases produce the best loss value observed in training. We then restore these weights after training only if the loss value of the current weights is larger than that of the saved checkpoint.

(v) **Training data and design of experiments**. First, we define a 'trial' as a complete training run for a DNN approximating a specific function, initialized as mentioned above.

Following the setup of §A.1, we run several trials solving the problem:

Given training data $\{(X_i, Y_i)\}_{i=1}^m \subset (\mathcal{X} \times \mathcal{Y})^m$, $X_i \sim_{\text{i.i.d.}} \mu$, $Y_i = F(X_i) + E_i \in \mathcal{Y}$, approximate $F \in L_\mu^2(\mathcal{X}; \mathcal{Y})$.

We generate the measurements $Y_i$ using mixed variational formulations of the parametric elliptic, Navier-Stokes-Brinkman and Boussinesq PDEs discretized using `FEniCS` with input data $X_i$. The noise $E_i \in \mathcal{Y}$ encompasses the discretization errors from numerical solution. Further details of the discretization can be found in §B. Each of our architectures is trained across a range of datasets with increasing sizes. This involves using a set of training data consisting of values $\{(X_i, Y_i))\}_{i=1}^m$, where $m$ denotes the size of the training data and belongs to the set $\{10, 20, 30, 40, 50, 60, 70, 80, 90, 100, 200, 300, 400, 500\}$. After training we calculate the testing error for each trial and run statistics across all trials for each dataset.

(vi) **Testing data and error metric**. The testing data is generated similarly to the training data, obtaining solutions at different points $X_i \in \mathcal{X}$ for $i = 1, \ldots, m_{\text{test}}$. However, the testing data

$\{(X_i, Y_i = F(X_i) + E_i)\}_{i=1}^{m_{\text{test}}}$ is generated using a deterministic high-order sparse grid collocation method [76]. In particular, we use sparse grid quadrature rules to compute approximations to the Bochner norms

$$\|F\|_{L^2_\mu(\mathcal{X}; \widetilde{\mathcal{Y}})} = \left( \int_{\mathcal{X}} \|F(X)\|^2_{\widetilde{\mathcal{Y}}} \, d\mu(X) \right)^{1/2} \approx \left( \sum_{i=1}^{m_{\text{test}}} \|F(X_i)\|^2_{\widetilde{\mathcal{Y}}} w_i \right)^{1/2},$$

where $\mu = a_d \sharp \varrho_d$ is the pushforward measure defined in (A.1) and $w_i, i = 1, \ldots, m_{\text{test}}$ are the quadrature weights. We use these approximations to compute the relative $L^2_\mu(\mathcal{X}; \widetilde{\mathcal{Y}})$ error

$$e_F^{\text{test}} = \frac{\left( \sum_{i=1}^{m_{\text{test}}} \|F(X_i) - \widehat{F}(X_i)\|^2_{\widetilde{\mathcal{Y}}} w_i \right)^{1/2}}{\left( \sum_{i=1}^{m_{\text{test}}} \|F(X_i)\|^2_{\widetilde{\mathcal{Y}}} w_i \right)^{1/2}}.$$

We use a high order isotropic Clenshaw Curtis sparse grid quadrature rule to evaluate $e_F^{\text{test}}$, as described in [2]. This method shows superior convergence over Monte Carlo integration to evaluate the global Bochner error. The sparse grid rule gives $m_{\text{test}}$ points at a level $\ell$ for $d$ dimensions. We rely on the `TASMANIAN` sparse grid toolkit [88–90] for the generation of the isotropic rule to study the generalization performance of the DNN.

(vii) **Visualization**. The graphs in Figs. 1-3 show the geometric mean (the main curve) and plus/minus one (geometric) standard deviation (the shaded region). We use the geometric mean because our errors are plotted in logarithmic scale on the $y$-axis. See [3, Sec. A.1] for further discussion about this choice.

## B  Description of the parametric PDEs used in the numerical experiments and their discretization

In this section, we provide full details of the parametric PDEs considered in our numerical experiments. We also describe their variational formulations and numerical solution using FEM.

### B.1  Parametric coefficients

We consider two parametric coefficients of the form (A.1) in our numerical experiments. Our first coefficient is

$$a_1(\boldsymbol{z}, \boldsymbol{x}) = 2.62 + \sum_{j=1}^{\infty} x_j \frac{\sin(\pi z_1 j)}{j^{3/2}}, \quad \forall \boldsymbol{z} \in \Omega, \tag{B.1}$$

where $x_j \in [-1, 1], \forall j$. Our second example involves a log-transformed coefficient, which is a rescaling of an example from [76] of a diffusion coefficient with one-dimensional (layered) spatial dependence given by

$$a_2(\boldsymbol{z}, \boldsymbol{x}) = \exp\left( 1 + x_1 \left( \frac{\sqrt{\pi}\beta}{2} \right)^{1/2} + \sum_{j=2}^{\infty} \zeta_j \vartheta_j(\boldsymbol{z}) x_j \right), \quad \forall \boldsymbol{z} \in \Omega,$$

$$\zeta_j := (\sqrt{\pi}\beta)^{1/2} \exp\left( \frac{-(\lfloor j/2 \rfloor \pi \beta)^2}{8} \right), \tag{B.2}$$

$$\vartheta_j(\boldsymbol{z}) := \begin{cases} \sin(\lfloor j/2 \rfloor \pi z_1 / \beta_p) & \text{if } j \text{ is even} \\ \cos(\lfloor j/2 \rfloor \pi z_1 / \beta_p) & \text{if } j \text{ is odd} \end{cases},$$

where $x_j \in [-1, 1], \forall j$. Here we let $\beta_c = 1/8$, and $\beta_p = \max\{1, 2\beta_c\}$, $\beta = \beta_c / \beta_p$.

In both cases we consider truncation of the expansion after $d$ terms, giving the map $a_{j,d} : [-1, 1]^d \to \mathcal{X}$, with $j = 1$ corresponding to (B.1) and $j = 2$ corresponding to (B.2). Our input samples are then $\{X_i = a_{j,d}(\boldsymbol{x}_i)\}_{i=1}^m \subset \mathcal{X}$ with $\boldsymbol{x}_i \in [-1, 1]^d$ drawn identically and independently from $\varrho_d$ and $j \in \{1, 2\}$. Note for the Boussinesq problem we also consider an additional parametric dependence in the tensor $\mathbb{K}$ describing the thermal conductivity of the fluid. See §B.5 and (B.17).

## B.2 Relevant spaces

We require several function space definitions for the development of the mixed variation formulations of the Poisson, Navier-Stokes-Brinkman and Boussinesq PDEs. Let $\Omega \subset \mathbb{R}^n$, $n \in \{2,3\}$, be the physical domain of a PDE. We write $\mathrm{L}^p(\Omega)$, $1 \le p \le \infty$, for the $L^p$-space of scalar-valued functions with respect to the Lebesgue measure (to avoid confusion with the Bochner space $L^2_\mu(\mathcal{X};\mathcal{Y})$). We denote the standard Sobolev spaces as $\mathrm{W}^{s,p}(\Omega)$ for $s \in \mathbb{R}$ and $p > 1$, and write $\mathrm{H}^k(\Omega)$ when $p = 2$ and $s = k$. Additionally, we consider the space of traces of functions in $\mathrm{H}^1(\Omega)$, denoted by $\mathrm{H}^{1/2}(\partial\Omega)$, and its dual, $\mathrm{H}^{-1/2}(\partial\Omega)$ (see, e.g., [11, Sec. 1.2] for further details). We also define the following closed subspace of $\mathrm{H}^1(\Omega)$:

$$\mathrm{H}^1_0(\Omega) := \overline{C_0^\infty(\Omega)}^{\|\cdot\|_{1,\Omega}}.$$

Here $\overline{C_0^\infty(\Omega)}^{\|\cdot\|_{1,\Omega}}$ denotes the closure of $C_0^\infty(\Omega)$ (i.e., the space of $C^\infty(\Omega)$ functions with compact support) with respect to the norm $\|\cdot\|_{1,\Omega}$, which, for any $v \in \mathrm{H}^1(\Omega)$, is given by

$$\|v\|_{1,\Omega} := \left\{ |v|^2_{1,\Omega} + \|v\|^2_{\mathrm{L}^2(\Omega)} \right\}^{1/2}, \quad \text{where } |v|_{1,\Omega} := \|\nabla v\|_{\mathrm{L}^2(\Omega)}.$$

For scalar functions $u$ and vector fields $\boldsymbol{v}$, we use $\nabla u$ and $\mathrm{div}(\boldsymbol{v})$ to denote their gradient and divergence, respectively. For tensor fields $\boldsymbol{\sigma}$ and $\boldsymbol{\tau}$, represented by $(\sigma_{i,j})^n_{i,j=1}$ and $(\tau_{i,j})^n_{i,j=1}$, respectively, we define $\mathbf{div}(\boldsymbol{\sigma})$ as the divergence operator $\mathrm{div}$ acting along the rows of $\boldsymbol{\sigma}$, and we define the trace and the tensor inner-product as

$$\mathrm{tr}(\boldsymbol{\sigma}) = \sum_{i=1}^n \sigma_{i,i}, \text{ and } \boldsymbol{\tau} : \boldsymbol{\sigma} = \sum_{i,j=1}^n \tau_{i,j}\sigma_{i,j},$$

respectively. Furthermore, we introduce the notation $\mathbf{L}^p(\Omega)$ and $\mathbb{L}^p(\Omega)$ to represent the vectorial and tensorial counterparts of $\mathrm{L}^p(\Omega)$, respectively, and $\mathbf{H}^1(\Omega)$ and $\mathbb{H}^1(\Omega)$ for the vectorial and tensorial counterparts of $\mathrm{H}^1(\Omega)$, respectively. Keeping this in mind, we introduce the Banach spaces

$$\mathbf{H}(\mathrm{div}_q; \Omega) := \left\{ \boldsymbol{v} \in \mathbf{L}^2(\Omega) : \mathrm{div}(\boldsymbol{v}) \in \mathrm{L}^q(\Omega) \right\},$$
$$\mathbb{H}(\mathbf{div}_q; \Omega) := \left\{ \boldsymbol{\tau} \in \mathbb{L}^2(\Omega) : \mathbf{div}(\boldsymbol{\tau}) \in \mathbf{L}^q(\Omega) \right\} \tag{B.3}$$

with norms

$$\|\boldsymbol{v}\|_{\mathbf{H}(\mathrm{div}_q;\Omega)} := \|\boldsymbol{v}\|_{\mathbf{L}^2(\Omega)} + \|\mathrm{div}(\boldsymbol{v})\|_{\mathrm{L}^q(\Omega)},$$
$$\|\boldsymbol{\tau}\|_{\mathbb{H}(\mathbf{div}_q;\Omega)} := \|\boldsymbol{\tau}\|_{\mathbb{L}^2(\Omega)} + \|\mathbf{div}(\boldsymbol{\tau})\|_{\mathbf{L}^q(\Omega)}.$$

The cases of $q = 4/3$ and $q = 2$ appear in the mixed variational formulations of the considered PDEs, and for the latter we simply write $\mathbf{H}(\mathrm{div};\Omega)$.

Often, under certain conditions, such as incompressibility conditions [33, eq.(2.4)], it is convenient to define variants of these spaces. For example, we define

$$\mathbb{L}^2_{\mathrm{tr}}(\Omega) := \left\{ \boldsymbol{\tau} \in \mathbb{L}^2(\Omega) : \mathrm{tr}(\boldsymbol{\tau}) = 0 \right\}, \tag{B.4}$$

which represents the space of integrable functions with zero trace over $\Omega$. Furthermore, given the decomposition (see, e.g., [32])

$$\mathbb{H}(\mathbf{div}_{4/3}; \Omega) = \mathbb{H}_0(\mathbf{div}_{4/3}; \Omega) \oplus \mathbb{R}\,\mathbb{I}, \tag{B.5}$$

we may also consider

$$\mathbb{H}_0(\mathbf{div}_{4/3}; \Omega) := \left\{ \boldsymbol{\tau} \in \mathbb{H}(\mathbf{div}_{4/3}; \Omega) : \int_\Omega \mathrm{tr}(\boldsymbol{\tau}) = 0 \right\}, \tag{B.6}$$

as the space of elements in $\mathbb{H}(\mathbf{div}_{4/3}; \Omega)$ with zero mean trace. Finally, we define

$$\mathbb{L}^2_{\mathrm{skew}}(\Omega) = \{ \boldsymbol{\eta} \in \mathbb{L}^2(\Omega) : \boldsymbol{\eta} + \boldsymbol{\eta}^t = 0 \}, \tag{B.7}$$

and the space of $\mathrm{L}^2(\Omega)$ functions with zero integral over $\Omega$ as

$$\mathrm{L}^2_0(\Omega) = \left\{ \nu \in \mathrm{L}^2(\Omega) : \int_\Omega \nu = 0 \right\}. \tag{B.8}$$

## B.3 The parametric diffusion equation

We now describe our first example, which is the parametric elliptic diffusion equation. Let $\Omega \subset \mathbb{R}^2$ be a bounded Lipschitz domain, $\partial\Omega$ be the boundary of $\Omega$, $f \in \mathrm{L}^2(\Omega)$ and $g \in \mathrm{H}^{1/2}(\partial\Omega)$. Given $\boldsymbol{x} \in [-1,1]^d$, we consider the PDE

$$-\mathrm{div}(a(\boldsymbol{x})\nabla u(\boldsymbol{x})) = f, \quad \text{in } \Omega \tag{B.9}$$
$$u(\boldsymbol{x}) = g, \quad \text{on } \partial\Omega.$$

Here $a(\boldsymbol{x}) = a(\cdot, \boldsymbol{x}) \in \mathrm{L}^\infty(\Omega) =: \mathcal{X}$ is the parametric diffusion coefficient. The terms $f$ and $g$ are not parametric.

### B.3.1 Mixed variational formulation

Our first step in precisely defining the problem is to identify sufficient conditions for the solution map $\boldsymbol{x} \mapsto u(\boldsymbol{x})$ to be well-defined. To do this, we diverge from the standard variational formulation involving the space $\mathcal{Y} = \mathrm{H}_0^1(\Omega)$ (see Remark B.3) and instead consider a *mixed* variational formulation of (B.9). Using a mixed formulation to study the solution of PDEs offers several benefits over the standard variational formulation. One key advantage is that it allows us to introduce additional variables that can be of physical interest. Additionally, mixed formulations can naturally accommodate different types of boundary conditions and introduce Dirichlet boundary conditions directly into the formulation rather than imposing them on the search space. For further details on mixed formulations, we refer to [32] and references within.

Assume that there exists $r, M > 0$ such that, for all $\boldsymbol{x} \in [-1,1]^d$,

$$0 < r \le \mathrm{essinf}_{\boldsymbol{z}\in\Omega} a(\boldsymbol{z}, \boldsymbol{x}) =: a_{\min}(\boldsymbol{x}) \text{ and } a_{\max}(\boldsymbol{x}) := \mathrm{esssup}_{\boldsymbol{z}\in\Omega} a(\boldsymbol{z}, \boldsymbol{x}) \le M. \tag{B.10}$$

Then the problem can be recast as a first-order system: given $\boldsymbol{x} \in [-1,1]^d$, find $(\boldsymbol{\sigma}, u)(\boldsymbol{x}) \in \mathbf{H}(\mathrm{div}; \Omega) \times \mathrm{L}^2(\Omega)$ such that

$$d_{a(\boldsymbol{x})}(\boldsymbol{\sigma}(\boldsymbol{x}), \boldsymbol{\tau}) + b(\boldsymbol{\tau}, u(\boldsymbol{x})) = G(\boldsymbol{\tau}), \quad \forall \boldsymbol{\tau} \in \mathbf{H}(\mathrm{div}; \Omega),$$
$$b(\boldsymbol{\sigma}, v) = F(v), \quad \forall v \in \mathrm{L}^2(\Omega). \tag{B.11}$$

Here $d$ and $b$ are the bilinear forms defined by

$$d_{a(\boldsymbol{x})}(\boldsymbol{\sigma}, \boldsymbol{\tau}) = \int_\Omega \frac{\boldsymbol{\sigma} \cdot \boldsymbol{\tau}}{a(\boldsymbol{x})}, \quad \forall (\boldsymbol{\tau}, \boldsymbol{\sigma}) \in \mathbf{H}(\mathrm{div}; \Omega) \times \mathbf{H}(\mathrm{div}; \Omega),$$

$$b(\boldsymbol{\tau}, v) = \int_\Omega \mathrm{div}(\boldsymbol{\tau}) v, \quad \forall (\boldsymbol{\tau}, v) \in \mathbf{H}(\mathrm{div}; \Omega) \times \mathrm{L}^2(\Omega)$$

and the functionals $G \in (\mathbf{H}(\mathrm{div}; \Omega))'$ and $J \in (\mathrm{L}^2(\Omega))'$ are defined by

$$J(v) = -\int_\Omega fv, \quad \forall v \in \mathrm{L}^2(\Omega) \text{ and } G(\boldsymbol{\tau}) = \langle \gamma(\boldsymbol{\tau}) \cdot \boldsymbol{n}, g \rangle_{1/2, \partial\Omega}, \quad \forall \boldsymbol{\tau} \in \mathbf{H}(\mathrm{div}; \Omega). \tag{B.12}$$

For the experiments in this work, we consider $\Omega = (0,1)^2$ and $f = 10$. For the boundary condition, we consider a constant value $u(\boldsymbol{z}, \boldsymbol{x}) = 0.5$ on the bottom of the boundary $(0,1) \times \{0\}$, and zero boundary conditions on the remainder of the boundary.

### B.3.2 Holomorphy assumption

Consider the affine parametrization (B.1). Setting $M = 2.7$ and observing that

$$\left| \sum_{j\in\mathbb{N}} x_j \frac{\sin(\pi z_1 j)}{j^{3/5}} \right| \le \sum_{j\in\mathbb{N}} \frac{1}{j^{3/5}} \approx 2.61238 = 2.62 - r, \tag{B.13}$$

for some $r < 0.00762$, we deduce that (B.10) holds, which makes the mixed variational formulation (B.11) well defined, i.e., for each $\boldsymbol{x} \in [-1,1]^\infty$, there exists a unique solution $(\boldsymbol{\sigma}, u)(\boldsymbol{x}) \in \mathbf{H}(\mathrm{div}; \Omega) \times \mathrm{L}^2(\Omega)$. Moreover, one can show that the parametric solution map $\boldsymbol{x} \mapsto (\sigma, u)(\boldsymbol{x})$ is $(\boldsymbol{b}, \varepsilon)$-holomorphic for $0 < \varepsilon < 0.00762$ and where $\boldsymbol{b} = (b_i)_{i=1}^\infty$ is given by $b_j = \| \sin(\pi j \cdot)/j^{3/2} \|_{\mathrm{L}^\infty(\Omega)} = j^{-3/2}$. See [74, Prop. A.3.2].

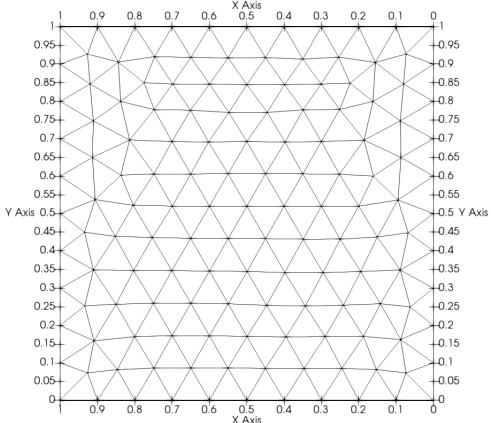

Figure 4: The domain $\Omega$ and FE mesh for the parametric diffusion equation.

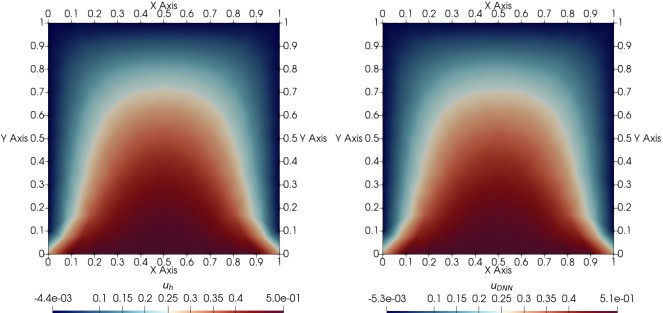

Figure 5: The solution $\boldsymbol{u}(\boldsymbol{x})$ of the parametric Poisson problem in (B.9) for a given parameter $\boldsymbol{x} = (1, 0, 0, 0)^\top$ with affine coefficient $a_{1,d}$ and $d = 4$, using a total of $K = 2622$ DoF. The left plot shows the solution given by the FEM solver. The right plot show the ELU $4 \times 40$ DNN approximation after $60,000$ epochs of training with $m = 500$ sample points for training.

In view of this property, this example falls within our theory. Note that $\boldsymbol{b} \in \ell_{\mathsf{M}}^p(\mathbb{N})$ for every $p < 2/3$. Thus, we expect a theoretical rate of convergence with respect to the amount of training data that is arbitrarily close to $m^{1/2-3/2} = m^{-1}$. This holomorphy result applies to the affine diffusion (B.1), not the log-transformed diffusion (B.2). However, we expect that it is possible to extend [74, Prop. A.3.2] to the latter case.

### B.3.3 Finite element discretization

We use so-called *conforming* Finite Element (FE) discretizations [18, Chp. 3]. Given a number of Degrees of Freedom (DoF) $K$, this results in finite-dimensional spaces $H_K \subseteq \mathbf{H}(\mathrm{div}; \Omega)$ and $Q_K \subseteq \mathrm{L}^2(\Omega)$. Specifically, we consider a regular triangulation $\mathcal{T}_K$ of $\overline{\Omega}$ made up of triangles of minimum diameter $h_{\min} = 0.0844$ and maximum diameter $h_{\max} = 0.1146$. This corresponds to a total number of DoF $K = 2622$. See Fig. 4 for an illustration of the FE mesh.

Fig. 5 shows a comparison between a reference solution computed by the `FEniCS` FEM solver and the approximation obtained by an ELU $4 \times 40$ DNN.

### B.4 Parametric Navier-Stokes-Brinkman equations

We next consider a parametric model describing the dynamics of a viscous fluid through porous media. Consider a bounded and Lipschitz physical domain $\Omega \subseteq \mathbb{R}^2$. Given $\boldsymbol{x} \in [-1, 1]^d$, we consider the incompressible nonlinear stationary Navier-Stokes-Brinkman (NSB) equations: find

$\boldsymbol{u} : [-1, 1]^d \times \Omega \to \mathbb{R}^2$ and $p : [-1, 1]^d \times \Omega \to \mathbb{R}$ such that

$$\eta \boldsymbol{u} - \lambda \mathbf{div}(a(\boldsymbol{x})e(\boldsymbol{u}(\boldsymbol{x}))) + (\boldsymbol{u}(\boldsymbol{x}) \cdot \nabla)\boldsymbol{u}(\boldsymbol{x}) + \nabla p(\boldsymbol{x}) = f \quad \text{in } \Omega$$

$$\mathrm{div}(\boldsymbol{u}(\boldsymbol{x})) = 0 \quad \text{in } \Omega \qquad \text{(B.14)}$$

$$\boldsymbol{u} = \begin{cases} \boldsymbol{u}_D & \text{on } \partial\Omega_{\text{in}} \\ 0 & \text{on } \partial\Omega_{\text{wall}} \end{cases}$$

$$(a(\boldsymbol{x})\nabla e(\boldsymbol{u}) - p\mathbb{I})\nu = 0 \quad \text{on } \partial\Omega_{\text{out}}$$

$$\int_\Omega p = 0,$$

where $\lambda = \mathrm{Re}^{-1}$ and Re is the Reynolds number, $a(\boldsymbol{x}) = a(\cdot; \boldsymbol{x}) \in \mathcal{X} := \mathrm{L}^\infty(\Omega)$ and $a : [-1, 1]^d \times \Omega \to \mathbb{R}^+$ is the random viscosity of the fluid, $\eta \in \mathbb{R}^+$ is the scaled inverse permeability of the porous media, $\boldsymbol{u}$ is the velocity of the fluid, $e(\boldsymbol{u}) = \frac{1}{2}(\nabla \boldsymbol{u} + (\nabla \boldsymbol{u})^t)$ is the symmetric part of the gradient, $p$ is the pressure of the fluid and $f : \Omega \to \mathbb{R}$ is an external force independent of the parameters. Here, the fourth condition imposes a zero normal Cauchy stress

$$(a(\boldsymbol{x})\nabla e(\boldsymbol{u}) - p\mathbb{I})\nu = 0$$

for the output boundary $\partial\Omega_{\text{out}}$. In addition, the incompressibility of the fluid imposes the following compatibility condition on $\boldsymbol{u}_D$:

$$\int_\Gamma \boldsymbol{u}_D \cdot \boldsymbol{n} = 0 \quad \text{on } \partial\Omega_{\text{in}}.$$

The third condition also imposes a no-slip condition on the walls $\Omega_{\text{wall}}$ [34, eq.(2.3)].

### B.4.1 Mixed variational formulation

The analysis of the detailed mixed formulation used for this problem in the nonparametric case can be found in [34]. Over the last decade, many works have used a mixed formulation employing a Banach space framework, allowing one to solve different PDEs in continuum mechanics in suitable Banach spaces. The advantage of this formulation is that no augmentation is required, the spaces are simpler and closer to the original model, and it allows one to obtain more direct approximations of the variables of physical interest [34, Sec. 1].

Based on the analysis in [34], the mixed variational formulation of the parametric NSB equations in (B.14) becomes: given $\boldsymbol{x} \in [-1, 1]^d$, find $(\boldsymbol{u}, \boldsymbol{t}, \boldsymbol{\sigma}, \boldsymbol{\gamma})(\boldsymbol{x}) \in \mathbf{L}^4(\Omega) \times \mathbb{L}^2_{\text{tr}}(\Omega) \times \mathbb{H}_0(\mathbf{div}_{4/3}; \Omega) \times \mathbb{L}^2_{\text{skew}}(\Omega)$ such that

$$\lambda \int_\Omega a_i(\boldsymbol{x})\boldsymbol{t}(\boldsymbol{x}) : \boldsymbol{s} - \int_\Omega \boldsymbol{s} : \boldsymbol{\sigma}(\boldsymbol{x}) - \int_\Omega (\boldsymbol{u} \otimes \boldsymbol{u})(\boldsymbol{x}) : \boldsymbol{s} = 0,$$

$$\int_\Omega \boldsymbol{t}(\boldsymbol{x}) : \boldsymbol{\tau} + \int_\Omega \boldsymbol{\gamma}(\boldsymbol{x}) : \boldsymbol{\tau} + \int_\Omega \boldsymbol{u}(\boldsymbol{x}) \cdot \mathbf{div}(\boldsymbol{\tau}) = \langle \boldsymbol{\tau}\boldsymbol{n}, \boldsymbol{u}_D \rangle_{\partial\Omega_{\text{in}}}, \qquad \text{(B.15)}$$

$$\int_\Omega \boldsymbol{\delta} : \boldsymbol{\sigma}(\boldsymbol{x}) + \int_\Omega \boldsymbol{v} \cdot \mathbf{div}(\boldsymbol{\sigma}(\boldsymbol{x})) - \int_\Omega \eta \boldsymbol{u}(\boldsymbol{x}) \cdot \boldsymbol{v} = \int_\Omega \boldsymbol{f} \cdot \boldsymbol{v},$$

for all $(\boldsymbol{v}, \boldsymbol{s}, \boldsymbol{\tau}, \boldsymbol{\delta}) \in \mathbf{L}^4(\Omega) \times \mathbb{L}^2_{\text{tr}}(\Omega) \times \mathbb{H}_0(\mathbf{div}_{4/3}; \Omega) \times \mathbb{L}^2_{\text{skew}}(\Omega)$. Numerically, the skew-symmetry of $\boldsymbol{\gamma}$ is imposed by searching for $\gamma \in \mathrm{L}^2(\Omega)$ and setting

$$\boldsymbol{\gamma} = \begin{bmatrix} 0 & \gamma \\ -\gamma & 0 \end{bmatrix}.$$

Moreover, we impose the Neumann boundary condition via a Nietsche method as in [34, Sec. 5.2]. Specifically, we add

$$\kappa \langle (\boldsymbol{\sigma} + \boldsymbol{u} \otimes \boldsymbol{u})\boldsymbol{n}), \boldsymbol{\tau}\boldsymbol{n} \rangle_{\partial\Omega_{\text{out}}} = 0$$

to the second equation where $\kappa \gg 1$ is a large constant (e.g., $\kappa = 10^4$). As usual in this formulation, the pressure $p \in \mathrm{L}^2(\Omega)$ can be computed according to the post-processing formula

$$p = -\frac{1}{2}\mathrm{tr}(\boldsymbol{\sigma} + (\boldsymbol{u} \otimes \boldsymbol{u})).$$

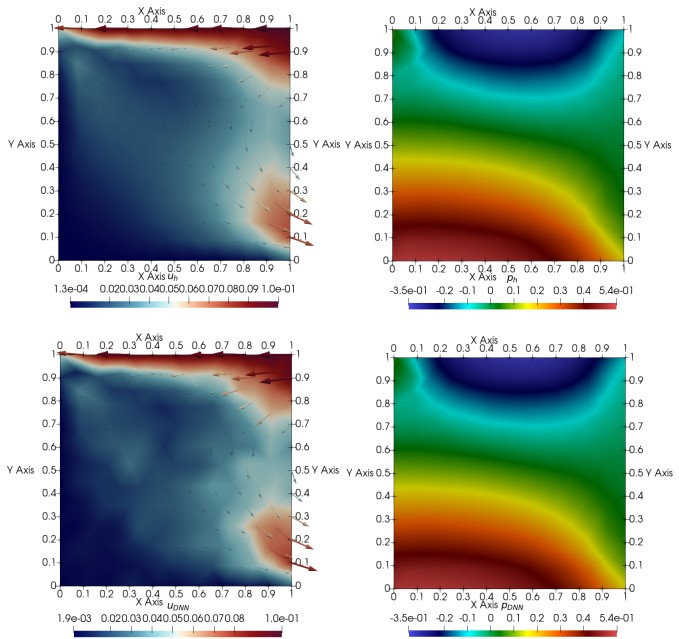

Figure 6: The solution $(\boldsymbol{u}, p)(\boldsymbol{x})$ of the parametric NSB problem (B.14) for a given parameter $\boldsymbol{x} = (1, 0, 0, 0)^\top$ with affine coefficient $a_{1,d}$ and $d = 4$, using a total of 1464 DoF for $\boldsymbol{u}$ and 244 DoF for $p$. The top row shows the solution given by the FEM solver, and the bottom row shows the ELU $4 \times 40$ DNN approximation after $60,000$ epochs of training with $m = 500$ sample points. The left plots show the vector field $\boldsymbol{u}$. The right plots show the points of highest pressure $p$.

Note that above we omitted the term $\boldsymbol{x}$ for simplicity.

In our experiments, we consider approximating solutions to the parametric NSB problem with $\lambda = 0.1$, a scaled inverse permeability of $\eta = 10 + z_1^2 + z_2^2$, an external force $\boldsymbol{f} = (0, -1)^\top$ and random viscosity $a_{j,d}$ as in (B.1)–(B.2) with $j \in \{1, 2\}$.

We consider the unit square $\Omega = (0, 1)^2$ as the domain, an inlet boundary $\partial\Omega_\text{in} = (0, 1) \times \{1\}$, an outlet boundary $\partial\Omega_\text{out} = \{1\} \times (0, 1)$ and walls $\partial\Omega_\text{wall} = \{0\} \times (0, 1) \cup (0, 1) \times \{0\}$. For simplicity, we use the same mesh as that of the previous example. See Fig. 4. On the Neumann boundary $\partial\Omega_\text{out}$ we consider a zero normal Cauchy stress, a Dirichlet condition $\boldsymbol{u}_D = (0.0625)^{-1}((z_2 - 0.5)(1 - z_2), 0)$ on $\partial\Omega_\text{in}$ and a no-slip velocity on $\partial\Omega_\text{wall}$.

Fig. 6 provides a comparison between a reference solution of the vector field $\boldsymbol{u}$ and pressure $p$ computed by the `FEniCS` FEM solver and the approximation generated by a ELU $4 \times 40$ DNN.

**Remark B.1 (Other auxiliary variables)** We report the performance of the DNNs approximating $(\boldsymbol{u}, p)(\boldsymbol{x}) \in \mathbf{L}^4(\Omega) \times \mathrm{L}^2(\Omega)$. Note that any solver based on the above formulation outputs several other variables, e.g., $(\boldsymbol{t}, \boldsymbol{\sigma}, \boldsymbol{\gamma})(\boldsymbol{x}) \in \mathbb{L}_\text{tr}^2(\Omega) \times \mathbb{H}_0(\mathbf{div}_{4/3}; \Omega) \times \mathbb{L}_\text{skew}^2(\Omega)$. One could also approximate these auxiliary variables using DNNs. However, we restrict our experiments to $(\boldsymbol{u}, p)$ as these are the primary variables of interest in the problem.

To conclude this discussion, in Fig. 7 we plot the numerical results for the approximation of the pressure $p$ in the above problem. This complements Fig. 2, which showed results for the velocity field $\boldsymbol{u}$. We once more observe similar results: ELU and tanh DNNs outperform ReLU DNNs, the rate of convergence appears to be close to $\mathcal{O}(m^{-1})$ and there is no degradation with increasing parametric dimension $d$.

## B.5 Parametric stationary Boussinesq equation

To recap, in our first example, we considered a mixed formulation of a parametric diffusion equation that provably satisfies the $(\boldsymbol{b}, \varepsilon)$-holomorphy assumption. Using this formulation, we considered

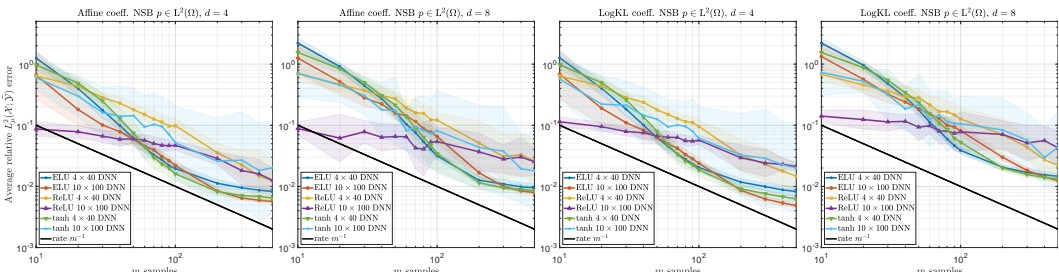

Figure 7: The same as Fig. 2, except showing results for the pressure $p$.

problems with nonzero Dirichlet boundary conditions, whereas previous works [2, 19, 27] study the more restrictive case of homogeneous Dirichlet boundary conditions, where $u \in \mathrm{H}_0^1(\Omega)$. In our next example, we studied a more complicated parametric PDE, namely, the parametric NSB equations. While this example currently lacks a holomorphy guarantee, we observe a convergence rate that aligns with what we expect. We conjecture that the $m^{-1}$ rate holds both for this and for even more complicated problems. To illustrate this claim with an example, we now consider a parametric coupled partial differential equation in three dimensions ($\Omega \subset \mathbb{R}^3$) with two random coefficients affecting different parts of the coupled problem. The nonparametric version of this problem is based on [20].

Specifically, we consider the Boussinesq formulation in [20] that combines a parametric incompressible Navier–Stokes equation with a parametric heat equation. The parametric dependence affects both equations. The Navier–Stokes equation is affected by a parametric variable multiplying the temperature-dependent viscosity, and the equation for heat flow is affected directly by the thermal conductivity of the fluid. To be more precise, given $\boldsymbol{x} \in [-1, 1]^d$, our goal is to find the velocity $\boldsymbol{u} : [-1, 1]^d \times \Omega \to \mathbb{R}^2$, pressure $p : [-1, 1]^d \times \Omega \to \mathbb{R}$ and temperature $\varphi : [-1, 1]^d \times \Omega \to \mathbb{R}$ of a fluid such that

$$
\begin{aligned}
-\mathbf{div}(2a(\boldsymbol{x})\varpi(\varphi(\boldsymbol{x}))\boldsymbol{e}(\boldsymbol{u}(\boldsymbol{x}))) + (\boldsymbol{u}(\boldsymbol{x}) \cdot \nabla)\boldsymbol{u}(\boldsymbol{x}) + \nabla p(\boldsymbol{x}) &= \varphi(\boldsymbol{x})\boldsymbol{g} \quad \text{in } \Omega, \\
\mathrm{div}(\boldsymbol{u}(\boldsymbol{x})) &= 0 \quad \text{in } \Omega, \\
-\mathrm{div}(\mathbb{K}(\boldsymbol{x})\nabla\varphi(\boldsymbol{x})) + \boldsymbol{u}(\boldsymbol{x}) \cdot \nabla\varphi(\boldsymbol{x}) &= 0 \quad \text{in } \Omega, \\
\boldsymbol{u} &= \boldsymbol{u}_D \quad \text{on } \partial\Omega, \\
\varphi &= \varphi_D \quad \text{on } \partial\Omega, \\
\int_\Omega p(\boldsymbol{x}) &= 0.
\end{aligned}
\tag{B.16}
$$

Here $\boldsymbol{g} = (0, 0, -1)^\top$ is a gravitational force and $\mathbb{K}(\boldsymbol{x}) = \mathbb{K}(\cdot; \boldsymbol{x}) \in \mathbb{L}^\infty(\Omega)$, where $\mathbb{K} : [-1, 1]^d \times \Omega \to \mathbb{R}^{3 \times 3}$ is a parametric uniformly-positive tensor describing the thermal conductivity of the fluid. It is given explicitly by

$$
\mathbb{K}(\boldsymbol{z}, \boldsymbol{x}) = \left(1.89 + \sum_{j \in \mathbb{N}} x_j \frac{\sin(\pi z_3 j)}{j^{9/5}}\right) \begin{bmatrix} \exp(-z_1) & 0 & 0 \\ 0 & \exp(-z_2) & 0 \\ 0 & 0 & \exp(-z_3) \end{bmatrix}, \ \forall \boldsymbol{z} \in \Omega, \quad \text{(B.17)}
$$

for $\boldsymbol{x} \in [-1, 1]^d$. The term $\varpi : \mathbb{R} \to \mathbb{R}^+$ is a temperature-dependent viscosity given by $\varpi(\varphi) = 0.1 + \exp(-\varphi)$ and the term $a(\boldsymbol{x}) = a(\cdot; \boldsymbol{x}) \in \mathrm{L}^\infty(\Omega)$, where $a : [-1, 1]^d \times \Omega \to \mathbb{R}$ is a parametric variable affecting the viscosity of the fluid. As in the previous example $\boldsymbol{e}(\boldsymbol{u})$ is the symmetric part of $\nabla\boldsymbol{u}$. Note that in this case, we have $(a(\boldsymbol{x}), \mathbb{K}(\boldsymbol{x})) \in \mathcal{X} := \mathrm{L}^\infty(\Omega) \times \mathbb{L}^\infty(\Omega)$.

### B.5.1 Fully mixed variational formulation

The complete derivation of a fully-mixed variational formulation for the non-parametric Boussinesq equation in Banach spaces can be found in [20, Sec. 3.1]. To make the presentation simpler we rewrite it for the parametric case. Given $\boldsymbol{x} \in [-1, 1]^d$, find $(\boldsymbol{u}, \boldsymbol{t}, \boldsymbol{\sigma}, \varphi, \tilde{\boldsymbol{t}}, \tilde{\boldsymbol{\sigma}})(\boldsymbol{x}) \in \mathbf{L}^4(\Omega) \times \mathbb{L}^2_{\mathrm{tr}}(\Omega) \times$

$\mathbb{H}_0(\mathbf{div}_{4/3}; \Omega) \times \mathrm{L}^4(\Omega) \times \mathbf{L}^2(\Omega) \times \mathbf{H}(\mathrm{div}_{4/3}; \Omega)$ such that

$$
\begin{aligned}
-\int_\Omega \boldsymbol{v} \cdot \mathbf{div}(\boldsymbol{\sigma}(\boldsymbol{x})) + \frac{1}{2}\int_\Omega t(\boldsymbol{x})\boldsymbol{u}(\boldsymbol{x}) \cdot \boldsymbol{v} - \int_\Omega \varphi(\boldsymbol{x})\boldsymbol{g} \cdot \boldsymbol{v} &= 0 \\
\int_\Omega 2a(\boldsymbol{x})\varpi(\varphi(\boldsymbol{x}))\boldsymbol{t}_{\mathrm{sym}}(\boldsymbol{x}) : \boldsymbol{s} - \frac{1}{2}\int_\Omega (\boldsymbol{u} \otimes \boldsymbol{u})(\boldsymbol{x}) : \boldsymbol{s} &= \int_\Omega \boldsymbol{\sigma}(\boldsymbol{x}) : \boldsymbol{s} \\
\int_\Omega \boldsymbol{\tau} : \boldsymbol{t}(\boldsymbol{x}) + \int_\Omega \boldsymbol{u}(\boldsymbol{x}) \cdot \mathbf{div}(\boldsymbol{\tau}) &= \langle \boldsymbol{\tau}\boldsymbol{\nu}, \boldsymbol{u}_D \rangle_\Gamma \\
-\int_\Omega \psi \, \mathrm{div}(\tilde{\boldsymbol{\sigma}}(\boldsymbol{x})) + \frac{1}{2}\int_\Omega \psi(\boldsymbol{x})\boldsymbol{u}(\boldsymbol{x}) \cdot \tilde{\boldsymbol{t}} &= 0 \\
\int_\Omega \mathbb{K}(\boldsymbol{x})\tilde{\boldsymbol{t}}(\boldsymbol{x}) \cdot \tilde{\boldsymbol{s}} - \frac{1}{2}\int_\Omega \varphi(\boldsymbol{x})\boldsymbol{u}(\boldsymbol{x}) \cdot \tilde{\boldsymbol{s}} &= \int_\Omega \tilde{\boldsymbol{\sigma}}(\boldsymbol{x}) \cdot \tilde{\boldsymbol{s}} \\
\int_\Omega \tilde{\boldsymbol{\tau}} \cdot \tilde{\boldsymbol{t}}(\boldsymbol{x}) + \int_\Omega \varphi(\boldsymbol{x})\, \mathrm{div}(\tilde{\boldsymbol{\tau}}) &= \langle \tilde{\boldsymbol{\tau}} \cdot \boldsymbol{\nu}, \varphi_D \rangle_\Gamma \\
\int_\Omega \mathrm{tr}(2\boldsymbol{\sigma} + \boldsymbol{u} \otimes \boldsymbol{u})(\boldsymbol{x}) &= 0,
\end{aligned}
\tag{B.18}
$$

for all $(\boldsymbol{v}, \boldsymbol{s}, \boldsymbol{\tau}, \psi, \tilde{\boldsymbol{\sigma}}, \tilde{\boldsymbol{\tau}}) \in \mathbf{L}^4(\Omega) \times \mathbb{L}^2_{\mathrm{tr}}(\Omega) \times \mathbb{H}_0(\mathbf{div}_{4/3}; \Omega) \times \mathrm{L}^4(\Omega) \times \mathbf{L}^2(\Omega) \times \mathbf{H}(\mathrm{div}_{4/3}; \Omega)$. Here $p \in \mathrm{L}^2_0(\Omega)$ can be recovered as

$$
p = -\frac{1}{6}\mathrm{tr}(2\sigma + 2c\mathbb{I} + \boldsymbol{u} \otimes \boldsymbol{u}), \quad \text{where } c = -\frac{1}{6|\Omega|}\int_\Omega \mathrm{tr}(\boldsymbol{u} \otimes \boldsymbol{u}).
\tag{B.19}
$$

As in the previous example, we omitted the term $\boldsymbol{x}$ from this equation for simplicity. For further details on this formulation we refer to [20] and references within.

Given $\boldsymbol{x} \in [-1, 1]^d$, we approximate the solution $(\boldsymbol{u}, p, \varphi)(\boldsymbol{x}) \in (\mathbf{L}^4(\Omega) \times \mathrm{L}^2_0(\Omega) \times \mathrm{L}^4(\Omega))$ of (B.18) by using DNNs and study the approximation capabilities as we increase the number of training samples $m$. As in the previous example (see Remark B.1), we do not aim to approximate the other variables $(\boldsymbol{t}, \boldsymbol{\sigma}, \tilde{\boldsymbol{t}}, \tilde{\boldsymbol{\sigma}})(\boldsymbol{x}) \in \mathbb{L}^2_{\mathrm{tr}}(\Omega) \times \mathbb{H}_0(\mathbf{div}_{4/3}; \Omega) \times \mathbf{L}^2(\Omega) \times \mathbf{H}(\mathrm{div}_{4/3}; \Omega)$.

In our experiments, we consider the unit cube $\Omega = (0, 1)^3$ as the domain in $\mathbb{R}^3$. We consider a nonzero boundary condition $u_D = (1, 1, 0)$ on the bottom face of the cube $\partial\Omega_{\mathrm{bottom}} = (0, 1) \times (0, 1) \times \{0\}$, and zero on the other faces. We set $\varphi_D = \exp(4(-(z_1 - 0.5)^2 - (z_2 - 0.5)^2))$ on $\partial\Omega_{\mathrm{bottom}}$ and zero otherwise. For simplicity, we consider the same parametric coefficients $a_{1,d}$ and $a_{2,d}$ given by (B.1) and (B.2), respectively. See Fig. 8 for an example of the solution $(\boldsymbol{u}, p, \varphi)(\boldsymbol{x})$ for a given $\boldsymbol{x} \in [-1, 1]^4$.

To conclude this section, we provide a comparison of the performance of the DNN architectures in approximating the velocity field $\boldsymbol{u}$ and pressure $p$ for the Boussinesq PDE. This complements Fig. 3, which showed results for the temperature $\varphi$. As with $\varphi$, the convergence rate for $p$ agrees roughly with the rate $m^{-1}$, and does not appear to deteriorate with the parametric dimension $d$. On the other hand, the convergence rate for the velocity field $\boldsymbol{u}$ is somewhat slower.

# C   Overview of the proofs

In this section, we first introduce additional notation that is needed for the proofs of the main results. We then give a brief overview of the proofs.

## C.1   Additional notation

### C.1.1   Lipschitz constants

Let $(\mathcal{X}, \|\cdot\|_\mathcal{X})$ and $(\mathcal{Y}, \|\cdot\|_\mathcal{Y})$ be Banach spaces, $G : \mathcal{X} \to \mathcal{Y}$ and $\mathcal{B} \subseteq \mathcal{X}$. We define the Lipschitz constant as $L = \mathrm{Lip}(G; \mathcal{B}, \mathcal{Y})$ as the smallest constant $L \geq 0$ such that

$$
\|G(X') - G(X)\|_\mathcal{Y} \leq L\|X' - X\|_\mathcal{X}, \quad \forall X, X' \in \mathcal{B}.
$$

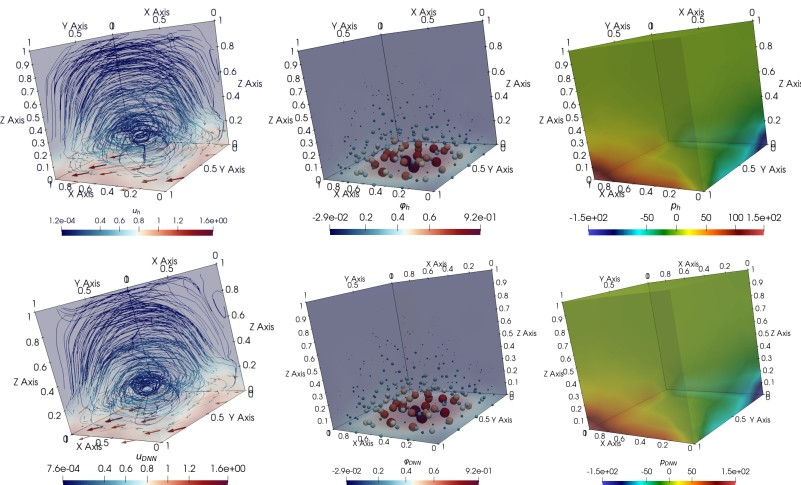

Figure 8: The solution $(\boldsymbol{u}, \varphi, p)(\boldsymbol{x})$ to the parametric Boussinesq problem in (B.18) for a given parameter $\boldsymbol{x} = (1, 0, 0, 0)^\top$ with affine coefficient $a_{1,d}$ and $d = 4$, using a total of $18, 480$ DoF for $\boldsymbol{u}$ and $528$ DoF for both $\varphi$ and $p$. The top row shows the solution given by the FEM solver and the bottom row shows the $4 \times 40$ ELU–DNN approximation after $60, 000$ epochs of training with $m = 500$ sample points. The left plots show streamlines of the vector field $\boldsymbol{u}$ and their directions indicated with coloured arrows. The middle plots visualize the temperature distribution inside the cube using coloured spheres, with the hottest region at the centre of the cube. The right plots illustrate the points of highest pressure $p$.

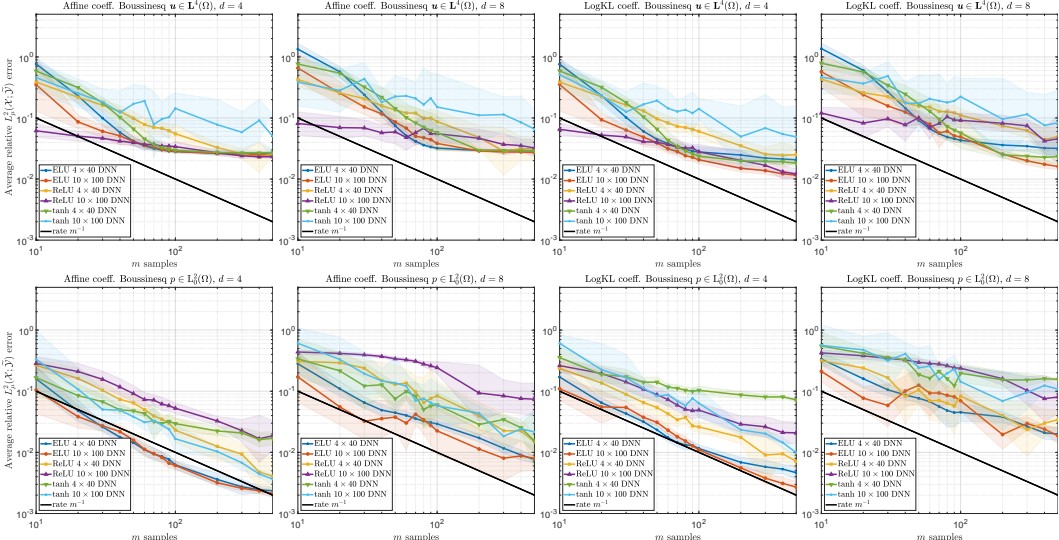

Figure 9: The same as Fig. 3, except showing results for the velocity field $\boldsymbol{u}$, where $\mathcal{Y} = \mathrm{L}^4(\Omega)$, and pressure $p$, where $\mathcal{Y} = \mathrm{L}^2_0(\Omega)$.

### C.1.2 Sequence spaces

We require some notation for sequences. Let $(\mathcal{Z}, \|\cdot\|_{\mathcal{Z}})$ be a Banach space, $d = \mathbb{N} \cup \{\infty\}$ and write $\boldsymbol{\nu} = (\nu_k)_{k=1}^d$ for an arbitrary multi-index in $\mathbb{N}_0^d$. If $\Lambda \subseteq \mathbb{N}_0^d$ is a finite or countable set of multi-indices and $0 < p \leq \infty$ we define the space $\ell^p(\Lambda; \mathcal{Z})$ as the set of all $\mathcal{Z}$-valued sequences $\boldsymbol{c} = (c_{\boldsymbol{\nu}})_{\boldsymbol{\nu} \in \Lambda}$ for which $\|\boldsymbol{c}\|_{p; \mathcal{Z}} < \infty$, where

$$\|\boldsymbol{c}\|_{p; \mathcal{Z}} = \begin{cases} \left( \sum_{\boldsymbol{\nu} \in \Lambda} \|c_{\boldsymbol{\nu}}\|_{\mathcal{Z}}^p \right)^{1/p} & 0 < p < \infty, \\ \sup_{\boldsymbol{\nu} \in \Lambda} \|c_{\boldsymbol{\nu}}\|_{\mathcal{Z}} & p = \infty. \end{cases}$$

When $(\mathcal{Z}, \|\cdot\|_{\mathcal{Z}}) = (\mathbb{R}, |\cdot|)$, we just write $\ell^p(\Lambda)$ and $\|\cdot\|_p$. We also define the following:

$$\||\boldsymbol{c}|\|_{p;\mathcal{Z}} = \sup_{z^* \in B(\mathcal{Z}^*)} \|z^*(\boldsymbol{c})\|_p, \quad \forall \boldsymbol{c} \in \ell^p(\Lambda; \mathcal{Z}),$$

where $z^*(\boldsymbol{c}) = (z^*(c_{\boldsymbol{\nu}}))_{\boldsymbol{\nu} \in \Lambda} \in \mathbb{R}^{|\Lambda|}$ for $\boldsymbol{c} = (c_{\boldsymbol{\nu}})_{\boldsymbol{\nu} \in \Lambda}$. Notice that $\||\boldsymbol{c}|\|_{p;\mathcal{Z}} \le \|\boldsymbol{c}\|_{p;\mathcal{Z}}$.

Given a sequence $\boldsymbol{c} = (c_{\boldsymbol{\nu}})_{\boldsymbol{\nu} \in \Lambda}$, we write

$$\mathrm{supp}(\boldsymbol{c}) = \{\boldsymbol{\nu} \in \Lambda : c_{\boldsymbol{\nu}} \ne 0\} \subseteq \Lambda.$$

For $d \in \mathbb{N} \cup \{\infty\}$, we write $\boldsymbol{e}_j, j = 1, \ldots, d$, for the canonical basis vectors in $\mathbb{R}^d$. We also write $\boldsymbol{0}$ for the multi-index of zeros and $\boldsymbol{1}$ for the multi-index of ones. Finally, for multi-indices $\boldsymbol{\nu} = (\nu_k)_{k=1}^d$ and $\boldsymbol{\mu} = (\mu_k)_{k=1}^d$, we write $\boldsymbol{\nu} \ge \boldsymbol{\mu}$ to mean $\nu_k \ge \mu_k, \forall k$ and likewise for $\boldsymbol{\nu} > \boldsymbol{\mu}$.

### C.1.3 Weights and weighted sequence spaces

Let $d \in \mathbb{N} \cup \{\infty\}$, $\Lambda \subseteq \mathbb{N}_0^d$ and $\boldsymbol{w} = (w_{\boldsymbol{\nu}})_{\boldsymbol{\nu} \in \Lambda} \ge \boldsymbol{0}$ be a sequence of nonnegative weights. We define the weighted cardinality of a set $S \subseteq \Lambda$ as

$$|S|_{\boldsymbol{w}} = \sum_{\boldsymbol{\nu} \in S} w_{\boldsymbol{\nu}}^2. \tag{C.1}$$

Given a sequence $\boldsymbol{c} = (c_{\boldsymbol{\nu}})_{\boldsymbol{\nu} \in \Lambda}$ we write

$$\|\boldsymbol{c}\|_{0,\boldsymbol{w}} = |\mathrm{supp}(\boldsymbol{c})|_{\boldsymbol{w}}.$$

Next, for a Banach space $(\mathcal{Z}, \|\cdot\|_{\mathcal{Z}})$ and $0 < p \le 2$, we define the weighted $\ell^p_{\boldsymbol{w}}(\Lambda; \mathcal{Z})$ space as the space of $\mathcal{Z}$-valued sequences $\boldsymbol{c} = (c_{\boldsymbol{\nu}})_{\boldsymbol{\nu} \in \Lambda}$ for which $\|\boldsymbol{c}\|_{p,\boldsymbol{w};\mathcal{Z}} < \infty$, where

$$\|\boldsymbol{c}\|_{p,\boldsymbol{u};\mathcal{Z}} = \left( \sum_{\boldsymbol{\nu} \in \Lambda} w_{\boldsymbol{\nu}}^{2-p} \|c_{\boldsymbol{\nu}}\|_{\mathcal{Z}}^p \right)^{1/p}.$$

Notice that $\|\cdot\|_{2,\boldsymbol{u};\mathcal{Z}}$ coincides with the unweighted norm $\|\cdot\|_{2;\mathcal{Z}}$.

### C.1.4 Legendre polynomials

We write $\{P_n\}_{n \in \mathbb{N}_0}$ for the classical Legendre polynomials on $[-1, 1]$ with normalization $P_n(1) = 1$. Since $\int_{-1}^1 |P_n(x)|^2 \, \mathrm{d}x = (n + 1/2)^{-1}$, we define the orthonormal (with respect to the uniform probability measure) Legendre polynomials as

$$\psi_n(x) = \sqrt{2n+1} P_n(x), \quad x \in [-1, 1], \ n \in \mathbb{N}_0.$$

Let $D = [-1, 1]^{\mathbb{N}}$ as before,

$$\mathcal{F} = \left\{ \boldsymbol{\nu} = (\nu_i)_{i=1}^{\infty} \in \mathbb{N}_0^{\mathbb{N}} : |\mathrm{supp}(\boldsymbol{\nu})| < \infty \right\}$$

be the set of multi-indices with finitely-many nonzero terms and define the multivariate Legendre polynomials as

$$\Psi_{\boldsymbol{\nu}}(\boldsymbol{x}) = \prod_{i \in \mathbb{N}} \psi_{\nu_i}(y_i) \equiv \prod_{i \in \mathrm{supp}(\boldsymbol{\nu})} \psi_{\nu_i}(x_i), \quad \forall \boldsymbol{x} \in D, \ \boldsymbol{\nu} = (\nu_i)_{i=1}^{\infty} \in \mathcal{F}.$$

Here, the second equality follows from the fact that $\psi_0 = 1$. Then it is known that the set

$$\{\Psi_{\boldsymbol{\nu}}\}_{\boldsymbol{\nu} \in \mathcal{F}} \subset L^2_{\varrho}(D) \tag{C.2}$$

constitutes an orthonormal basis for $L^2_{\varrho}(D)$ [19, §3]. For later use, we also define the sequences

$$\boldsymbol{u} = (u_{\boldsymbol{\nu}})_{\boldsymbol{\nu} \in \mathcal{F}}, \quad \text{where } u_{\boldsymbol{\nu}} = \|\Psi_{\boldsymbol{\nu}}\|_{L^\infty_{\varrho}(D)} = \prod_{k \in \mathbb{N}} \sqrt{2\nu_k + 1}, \ \forall \boldsymbol{\nu} \in \mathcal{F}, \tag{C.3}$$

and

$$\boldsymbol{v} = (v_{\boldsymbol{\nu}})_{\boldsymbol{\nu} \in \mathcal{F}}, \quad \text{where } v_{\boldsymbol{\nu}} = u_{\boldsymbol{\nu}}^{5+\xi}, \ \forall \boldsymbol{\nu} \in \mathcal{F}. \tag{C.4}$$

Here $\xi \ge 0$ will be chosen later in the proof.

### C.1.5 Miscellaneous

Given an optimization problem $\min_t f(t)$, we say that $\hat{t}$ is a $(\sigma, \tau)$-*approximate minimizer* for some $\sigma \geq 1$ and $\tau \geq 0$ if $f(\hat{t}) \leq \sigma^2 \min_t f(t) + \tau^2$. (In the main paper, we consider only $\sigma = 1$, but for the proofs it is useful to allow $\sigma > 1$).

Finally, for convenience, given $X_1, \ldots, X_m \in \mathcal{X}$, we define the semi-norm

$$\|G\|_{\mathsf{disc},\mu} = \sqrt{\frac{1}{m} \sum_{i=1}^{m} \|G(X_i)\|_{\mathcal{Y}}^2}$$

of an operator $G : \mathcal{X} \to \mathcal{Y}$ and

$$\|g\|_{\mathsf{disc},\tilde{\varsigma}} = \|g \circ \mathcal{E}_{\mathcal{X}}\|_{\mathsf{disc},\mu} = \sqrt{\frac{1}{m} \sum_{i=1}^{m} \|g \circ \mathcal{E}_{\mathcal{X}}(X_i)\|_{\mathcal{Y}}^2}$$

of a function $g : \mathbb{R}^{\mathbb{N}} \to \mathcal{Y}$. Here, as in (A.III), $\tilde{\varsigma}$ denotes the pushforward measure $\mathcal{E}_{\mathcal{X}} \# \mu$.

## C.2 Overview of the proofs

### C.2.1 Theorem 3.1

Theorems 3.1-3.2 are based on polynomials, and specifically, procedures for learning efficient Legendre polynomial approximations to holomorphic operators. As observed, these results are based on recent work on learning holomorphic, Banach-valued functions [2, 5, 6]. See also Remark C.1 below.

The proof of Theorem 3.1 involves three mains steps.

(a) Formulation (§D.1) and analysis (§D.2-D.4) of a suitable polynomial learning procedure.
(b) Construction of a family of DNNs that approximately emulates the Legendre polynomials (§D.5).
(c) Analysis of the corresponding training problem (2.5) (§D.6–D.8).

Since our goal in this work is 'agnostic' DNN architectures (i.e., independent of the smoothness of the underlying operator), in step (a) we first define a nonlinear set (D.2) spanned by Legendre polynomials with nonzero indices in certain sets of bounded weighted cardinality. This is effectively a form of *sparse polynomial approximation*, and the analysis of the resulting learning procedure (D.3) relies heavily on techniques from compressed sensing. In order to bound the encoding-decoding error, we also require several results on Lipschitz continuity (Lemma D.2) and norm equivalences (Lemma D.4) for multivariate polynomials.

Step (b) relies on what have now become fairly standard results in DNN approximation theory: namely, the approximate emulation of orthogonal polynomials via DNNs of given width and depth. We present such a result in Lemma D.9, then use this to define the DNN family $\mathcal{N}$ in (D.19).

We then analyze the DNN training problem (2.5) in step (c). Using the emulation result of step (b), we first show that any approximate minimizer $\widehat{N}$ of (2.5) yields a polynomial that is also an approximate minimizer of the polynomial training problem (D.3) (Lemma D.12). We may then apply the results shown in Step (a) to prove a generalization bound (Theorem D.13). Up to this point, we have not used the holomorphy assumption. We now use this assumption to bound the various best polynomial approximation errors that arise in the previously-derived generalization bound (§D.7). Finally, in §D.8 we put all these estimates together to complete the proof.

**Remark C.1** Theorem 3.1 is a generalization and improvement of [5, Thms. 4.1 & 4.2], which deals with the case of learning Banach-valued functions rather than operators. Specifically, the setting of [5] can be considered a special case of this paper where $\mathcal{X} = \ell^{\infty}(\mathbb{N})$ and the encoding error $E_{\mathcal{X},q} = 0$. Moreover, Theorem 3.1 improves the main results of [5] in three key ways. First, the the DNN architectures bounds are much narrower: $\mathrm{width}(\mathcal{N}) \lesssim (m/L)^{1+\delta}$ versus $\mathrm{width}(\mathcal{N}) \lesssim m^{3+\log_2(m)}$ in the latter. Second, Theorem 3.1 considers standard training, i.e., $\ell^2$-loss minimization, whereas [5, Thms. 4.1 & 4.2] requires regularization. Third, for Banach spaces, the error decay rate with respect

to $m$ is roughly doubled: $E_{\mathsf{app},2} = \mathcal{O}(m^{1-1/p})$ in Theorem 3.1 versus $\mathcal{O}(m^{1/2(1-1/p)})$ in [5, Thm. 4.1]. Finally, Theorem 3.1 also provides bounds in the $L_\mu^\infty$-norm, whereas the results in [5] only consider the $L_\mu^2$-norm.

### C.2.2 Theorem 3.2

The proof of Theorem 3.2 relies of three key steps.

(a) Using a minimizer of the polynomial training problem (D.3) to construct a family of minimizers of the DNN training problem (2.5).

(b) Analysis of the corresponding minimizers using the previously-derived bound for polynomial minimizers.

(c) Using the Lipschitz continuity of DNNs to show stability of the DNN minimizer and a permutation argument to show the existence of many parameters that lead to equally 'good' minimizers.

Given any minimizer of the polynomial training problem (D.3), it is straightforward to define a DNN with the desired generalization bound. Unfortunately, this will generally not be a minimizer of (2.5). To achieve the aims of step (a) we proceed as follows. First, we note that a DNN will be a minimizer if the corresponding approximation satisfies $\widehat{F}(X_i) = \widetilde{Y}_i$, where $\widetilde{Y}_i$ are the closest points to the $Y_i$ from $\widetilde{\mathcal{Y}} = \mathcal{D}_{\mathcal{Y}}(\mathbb{R}^{d_{\mathcal{Y}}})$. To achieve this, we take the existing DNN then add on a suitable number of additional terms corresponding to the first $r > m$ order-one Legendre polynomials (§E.1). We show that by doing this, we can construct a DNN for which $\widehat{F}(X_i) = \widetilde{Y}_i$ (Lemma E.1).

In Step (b), we first bound this DNN minimizer in terms of the polynomial minimizer plus the contributions of these additional terms (Lemma E.2). The latter involves the minimal singular value of a certain $m \times r$ matrix $\boldsymbol{B}$, which is the matrix of the linear system that enforces the condition $\widehat{F}(X_i) = \widetilde{Y}_i$. We bound this minimal singular value in §E.3.

We then use this to complete the proof of part (A) of Theorem 3.2 in §E.4. In this section, we also complete step (c) to establish parts (B) and (C).

**Remark C.2** Like Theorem 3.1, Theorem 3.2 also relies on ideas from [5]. However, [5] does not address fully-connected DNN architectures. To address this challenge, the proof of Theorem 3.2 involves the technical construction described above.

### C.2.3 Theorem 4.1

Theorems 4.1 is based on [7, Thm. 4.4]. The basic idea is to consider a family of affine, holomorphic operators (F.4). This allows us to lower bound the quantity $\theta_m(\boldsymbol{b})$ by the so-called *Gelfand width* (F.1) of a certain weighted unit ball in a finite-dimensional space. Bounds for such Gelfand widths are known, and this allows us to derive the corresponding result. The main difference between this and [7, Thm. 4.4] is the setup leading to the construction in (F.4).

### C.2.4 Theorem 4.2

Theorems 4.2 employs similar ideas, but in a more technical manner. We consider a family of linear, holomorphic operators (G.2), which involves a sum over $r$ groups of $m+1$ coefficients. We restrict to coefficients that lie in the null space of the corresponding sampling operator. Then, through a series of inequalities, we can lower bound $\tilde{\theta}_m(\boldsymbol{b})$ by a sum over $r$ terms, each involving the $\ell^1$-norms of certain vectors in the null space of the matrix of the corresponding sampling operator (G.3). This matrix is a subgaussian random matrix. We now use a technical estimate from [75] for vectors in the null space of subgaussian random matrices, which shows that they cannot be 'spiky'. Applying this and a series of further inequalities yields the result.

## D Proof of Theorem 3.1

### D.1 Formulation of an approximate polynomial training problem

Let $\Lambda \subset \mathcal{F}$ with $\mathrm{supp}(\boldsymbol{\nu}) \subseteq \{1, \ldots, d_{\mathcal{X}}\}$, $\forall \boldsymbol{\nu} \in \Lambda$, and write $N = |\Lambda|$. Let $k > 0$ and define the set
$$\mathcal{S} = \mathcal{S}_{\Lambda,k} = \{S \subseteq \Lambda : |S|_{\boldsymbol{v}} \le k\}. \tag{D.1}$$

Both $\Lambda$ and $k$ will be chosen later in the proof. For any Banach space $(\mathcal{Z}, \|\cdot\|_{\mathcal{Z}})$ define the space

$$\mathcal{P}_{\mathcal{S};\mathcal{Z}} = \left\{ \sum_{\boldsymbol{\nu} \in S} c_{\boldsymbol{\nu}} \Psi_{\boldsymbol{\nu}} : c_{\boldsymbol{\nu}} \in \mathcal{Z}, S \in \mathcal{S} \right\}. \tag{D.2}$$

Then, given the training data (2.4), consider the problem

$$\min_{p \in \mathcal{P}_{\mathcal{S};\widetilde{\mathcal{Y}}}} \frac{1}{m} \sum_{i=1}^{m} \|Y_i - p \circ \mathcal{E}_{\mathcal{X}}(X_i)\|_{\mathcal{Y}}^2, \tag{D.3}$$

where $\widetilde{\mathcal{Y}} = \mathcal{D}_{\mathcal{Y}}(\mathbb{R}^{d_{\mathcal{Y}}})$. Here and throughout the proofs, we slightly abuse notation: the polynomial $p : \mathbb{R}^{\mathbb{N}} \to \mathbb{R}$, whereas $\mathcal{E}_{\mathcal{X}}$ has codomain $\mathbb{R}^{d_{\mathcal{X}}}$. However, by construction, $p$ is independent of all but the first $d_{\mathcal{X}}$ variables. Hence we may consider $p$ as a function $\mathbb{R}^{d_{\mathcal{X}}} \to \mathbb{R}$. This aside, if $\hat{p}$ is an approximate minimizer of (D.3), then we define the approximation to $F$ as

$$F \approx \widehat{F} = \hat{p} \circ \mathcal{E}_{\mathcal{X}}. \tag{D.4}$$

Notice that $p \circ \mathcal{E}_{\mathcal{X}} = \mathcal{D}_{\mathcal{Y}} \circ P \circ \mathcal{E}_{\mathcal{X}}$, where $P : \mathbb{R}^{d_{\mathcal{X}}} \to \mathbb{R}^{d_{\mathcal{Y}}}$ is a vector-valued polynomial, since, by (A.IV), the map $\mathcal{D}_{\mathcal{Y}}$ is linear. The idea exploited in this proof is to construct a class of DNNs $\mathcal{N}$ such that (i) all such polynomials $P$ are approximated by members of $\mathcal{N}$ and (ii) the polynomial training problem (D.3) is approximated by the DNN training problem (2.5). The first step in this analysis is therefore to analyze the polynomial training problem (D.3).

## D.2 Supporting lemmas

We require several lemmas. The first relates the $L_\varrho^\infty$-norm of a polynomial to its Pettis $L_\varrho^2$-norm.

**Lemma D.1** (Nikolskii inequality for polynomials). *Let $(\mathcal{Z}, \|\cdot\|_{\mathcal{Z}})$ be any Banach space and $p = \sum_{\boldsymbol{\nu} \in S} c_{\boldsymbol{\nu}} \Psi_{\boldsymbol{\nu}}$ for some finite set $S \subset \mathcal{F}$, where $c_{\boldsymbol{\nu}} \in \mathcal{Z}$. Then*

$$\|p\|_{L_\varrho^\infty(D;\mathcal{Z})} \leq \sqrt{|S|_{\boldsymbol{u}}} \|\!\|p\|\!\|_{L_\varrho^2(D;\mathcal{Z})}.$$

*Proof.* By definition

$$\|p\|_{L_\varrho^\infty(D;\mathcal{Z})} = \|\!\|p\|\!\|_{L_\varrho^\infty(D;\mathcal{Z})} = \sup_{z^* \in B(\mathcal{Z}^*)} \|z^*(p)\|_{L_\varrho^\infty(D)}.$$

Fix $z^* \in B(\mathcal{Z}^*)$ and write $z^*(p) = \sum_{\boldsymbol{\nu} \in S} z^*(c_{\boldsymbol{\nu}}) \Psi_{\boldsymbol{\nu}}$. By the triangle inequality and the definition of the weights (C.3), we have

$$\|z^*(p)\|_{L_\varrho^\infty(D)} \leq \sum_{\boldsymbol{\nu} \in S} |z^*(c_{\boldsymbol{\nu}})| \|\Psi_{\boldsymbol{\nu}}\|_{L_\varrho^\infty(D)} = \sum_{\boldsymbol{\nu} \in S} u_{\boldsymbol{\nu}} |z^*(c_{\boldsymbol{\nu}})|.$$

We now apply the Cauchy–Schwarz inequality, (C.1) and Parseval's identity to get

$$\|z^*(p)\|_{L_\varrho^\infty(D)} \leq \sqrt{|S|_{\boldsymbol{u}}} \sqrt{\sum_{\boldsymbol{\nu} \in S} |z^*(c_{\boldsymbol{\nu}})|^2} = \sqrt{|S|_{\boldsymbol{u}}} \|z^*(p)\|_{L_\varrho^2(D)}.$$

Since $z^*$ was arbitrary, we deduce that

$$\|p\|_{L_\varrho^\infty(D;\mathcal{Z})} \leq \sqrt{|S|_{\boldsymbol{u}}} \sup_{z^* \in B(\mathcal{Z}^*)} \|z^*(p)\|_{L_\varrho^2(D)} = \sqrt{|S|_{\boldsymbol{u}}} \|\!\|p\|\!\|_{L_\varrho^2(D;\mathcal{Z})},$$

as required. $\qquad\square$

Next, we require the following bound on the Lipschitz constant of a multivariate polynomial.

**Lemma D.2** (Lipschitz continuity for polynomials). *Let $(\mathcal{Z}, \|\cdot\|_{\mathcal{Z}})$ be any Banach space and suppose that $p = \sum_{\boldsymbol{\nu} \in S} c_{\boldsymbol{\nu}} \Psi_{\boldsymbol{\nu}}$ for some finite $S \subset \mathcal{F}$, where $c_{\boldsymbol{\nu}} \in \mathcal{Z}$. Then satisfies*

$$\mathrm{Lip}(p; B^\infty(\mathbb{N}), \mathcal{Z}) \leq \frac{1}{2} \sqrt{|S|_{\boldsymbol{v}}} \cdot \|\!\|p\|\!\|_{L_\varrho^2(D;\mathcal{Z})},$$

*where $B^\infty(\mathbb{N}) = \{\boldsymbol{x} \in \mathbb{R}^{\mathbb{N}} : \|\boldsymbol{x}\|_\infty \leq 1\}$ is the unit ball of $\ell^\infty(\mathbb{N})$.*

*Proof.* Fix $z^* \in B(\mathcal{Z}^*)$ and let $\tilde{p} = z^*(p) = \sum_{\boldsymbol{\nu} \in S} z^*(c_{\boldsymbol{\nu}}) \Psi_{\boldsymbol{\nu}} =: \sum_{\boldsymbol{\nu} \in S} \tilde{c}_{\boldsymbol{\nu}} \Psi_{\boldsymbol{\nu}}$ be the corresponding scalar-valued polynomial. Let $\boldsymbol{x}, \boldsymbol{x}' \in D$. Then the mean value theorem gives that

$$\tilde{p}(\boldsymbol{x}') - \tilde{p}(\boldsymbol{x}) = \sum_{i=1}^{\infty} (x_i' - x_i) \frac{\partial \tilde{p}}{\partial x_i}(t\boldsymbol{x} + (1-t)\boldsymbol{x}')$$

for some $0 \le t \le 1$. Since $B^{\infty}(\mathbb{N}) \equiv D$, this and the fact that $D$ is convex give that

$$|\tilde{p}(\boldsymbol{x}') - \tilde{p}(\boldsymbol{x})| \le \|\boldsymbol{x}' - \boldsymbol{x}\|_{\infty} \sum_{i=1}^{\infty} \left\| \frac{\partial \tilde{p}}{\partial x_i} \right\|_{L_{\varrho}^{\infty}(D)}. \tag{D.5}$$

We now consider the terms in the sum separately. Using the definition of the Legendre polynomials (see §C.1.4), we have

$$\frac{\partial \tilde{p}}{\partial x_i} = \sum_{\boldsymbol{\nu} \in S} \tilde{c}_{\boldsymbol{\nu}} u_{\boldsymbol{\nu}} P'_{\nu_i}(x_i) \prod_{j \neq i} P_{\nu_j}(x_j).$$

The unnormalized Legendre polynomials satisy $1 = P_n(1) = \|P_n\|_{L^{\infty}([-1,1])}$ and $n(n+1)/2 = P'_n(1) = \|P'_n\|_{L^{\infty}([-1,1])}$. Hence

$$\left\| \frac{\partial \tilde{p}}{\partial x_i} \right\|_{L_{\varrho}^{\infty}(D)} \le \sum_{\boldsymbol{\nu} \in S} |\tilde{c}_{\boldsymbol{\nu}}| u_{\boldsymbol{\nu}} \nu_i (\nu_i + 1)/2.$$

We deduce that

$$\sum_{i=1}^{\infty} \left\| \frac{\partial \tilde{p}}{\partial x_i} \right\|_{L_{\varrho}^{\infty}(D)} \le \sum_{\boldsymbol{\nu} \in S} |\tilde{c}_{\boldsymbol{\nu}}| u_{\boldsymbol{\nu}} \sum_{i=1}^{\infty} \nu_i (\nu_i + 1)/2.$$

Now, by definition of the weights $u_{\boldsymbol{\nu}}$ and $v_{\boldsymbol{\nu}}$ (see (C.3) and (C.4)), we have

$$(\nu_i + 1) \le \prod_{j \in \mathbb{N}} (2\nu_j + 1) = u_{\boldsymbol{\nu}}^2$$

and

$$\sum_{i=1}^{\infty} \nu_i \le \prod_{j \in \mathbb{N}} (2\nu_j + 1) = u_{\boldsymbol{\nu}}^2.$$

Hence

$$\sum_{i=1}^{\infty} u_{\boldsymbol{\nu}} \nu_i (\nu_i + 1) \le u_{\boldsymbol{\nu}}^5 \le v_{\boldsymbol{\nu}}.$$

Here we also used the fact that $\boldsymbol{u} \ge \boldsymbol{1}$. We now apply this, the Cauchy-Schwarz inequality and Parseval's identity to get

$$\sum_{i=1}^{\infty} \left\| \frac{\partial \tilde{p}}{\partial x_i} \right\|_{L_{\varrho}^{\infty}(D)} \le \frac{1}{2} \|\tilde{p}\|_{L_{\varrho}^2(D)} \sqrt{|S|_{\boldsymbol{v}}}.$$

Substituting this into (D.5) now gives

$$|\tilde{p}(\boldsymbol{x}') - \tilde{p}(\boldsymbol{x})| \le \frac{1}{2} \|\tilde{p}\|_{L_{\varrho}^2(D)} \sqrt{|S|_{\boldsymbol{v}}} \|\boldsymbol{x}' - \boldsymbol{x}\|_{\infty}.$$

We now recall that $\tilde{p} = z^*(p)$ and $z^* \in B(\mathcal{Z}^*)$ was arbitrary to get

$$\begin{aligned} \|p(\boldsymbol{x}') - p(\boldsymbol{x})\|_{\mathcal{Z}} &= \sup_{z^* \in B(\mathcal{Z}^*)} |z^*(p(\boldsymbol{x}') - p(\boldsymbol{x}))| \\ &\le \frac{1}{2} \sup_{z^* \in B(\mathcal{Z}^*)} \|z^*(p)\|_{L_{\varrho}^2(D)} \sqrt{|S|_{\boldsymbol{v}}} \|\boldsymbol{x}' - \boldsymbol{x}\|_{\infty} \\ &= \frac{1}{2} \||p\||_{L_{\varrho}^2(D;\mathcal{Z})} \sqrt{|S|_{\boldsymbol{v}}} \|\boldsymbol{x}' - \boldsymbol{x}\|_{\infty}, \end{aligned}$$

as required. $\qquad\square$

We now show that this result can be used to imply a norm equivalence for polynomials. For this we first require the following lemma. Note that in this and subsequent results, we abuse notation and write $\tilde{\varsigma}$ for both the measure on $[-1,1]^{d_{\mathcal{X}}}$ defined in (A.III) and the measure on $D = [-1,1]^{\mathbb{N}}$ defined by tensoring this measure (corresponding to the first $d_{\mathcal{X}}$ variables $x_1, \ldots, x_{d_{\mathcal{X}}}$) with the uniform measure on $D \backslash [-1,1]^{d_{\mathcal{X}}}$ (corresponding to the remaining variables $x_{d_{\mathcal{X}}+1}, x_{d_{\mathcal{X}}+2}, \ldots$).

**Lemma D.3** (Closeness of $L^2$ norms). *Let $(\mathcal{Z}, \|\cdot\|_{\mathcal{Z}})$ be any Banach space, $\varsigma$ be the measure defined in (A.I) and $\tilde{\varsigma}$ be the measure defined in (A.III). Suppose that $f \in L^2_{\varsigma}(D; \mathcal{Z})$ is Lipschitz continuous with constant $L = \text{Lip}(f; B^{\infty}(\mathbb{N}), \mathcal{Z}) < \infty$ and that $f$ depends only on its first $d_{\mathcal{X}}$ variables. Then $f \in L^2_{\tilde{\varsigma}}(D; \mathcal{Z})$ and*

$$\|\|f\|\|_{L^2_{\varsigma}(D;\mathcal{Z})} - \delta \leq \|\|f\|\|_{L^2_{\tilde{\varsigma}}(D;\mathcal{Z})} \leq \|\|f\|\|_{L^2_{\varsigma}(D;\mathcal{Z})} + \delta,$$

*where*

$$\delta = L \cdot \|\iota_{d_{\mathcal{X}}} - \iota_{d_{\mathcal{X}}} \circ \widetilde{\mathcal{D}}_{\mathcal{X}} \circ \widetilde{\mathcal{E}}_{\mathcal{X}}\|_{L^2_{\mu}(\mathcal{X};\ell^{\infty}(\mathbb{N}))}.$$

*Proof.* Fix $z^* \in B(\mathcal{Z}^*)$, let $g = z^*(f) \in L^2_{\varsigma}(D)$ and set

$$a = \|g\|_{L^2_{\tilde{\varsigma}}(D)}, \qquad b = \|g\|_{L^2_{\varsigma}(D)}.$$

Notice that

$$|g(\boldsymbol{x}) - g(\boldsymbol{x}')| \leq \|z^*\|_{\mathcal{Z}^*}\|f(\boldsymbol{x}) - f(\boldsymbol{x}')\|_{\mathcal{Z}} \leq L\|\boldsymbol{x} - \boldsymbol{x}'\|_{\infty}, \quad \forall \boldsymbol{x}, \boldsymbol{x}' \in B^{\infty}(\mathbb{N}).$$

Now, with slight abuse of notation, $g(\iota(X)) = g(\iota_{d_{\mathcal{X}}}(X))$ due to the assumption on $f$. Therefore, using this and the Cauchy–Schwarz inequality,

$$a^2 - b^2 = \int_D (g(\iota_{d_{\mathcal{X}}} \circ \widetilde{\mathcal{D}}_{\mathcal{X}} \circ \widetilde{\mathcal{E}}_{\mathcal{X}}(X)))^2 - (g(\iota_{d_{\mathcal{X}}}(X)))^2 \, \mathrm{d}\mu(X)$$

$$\leq L \int_D \|\iota_{d_{\mathcal{X}}}(X) - \iota_{d_{\mathcal{X}}} \circ \widetilde{\mathcal{D}}_{\mathcal{X}} \circ \widetilde{\mathcal{E}}_{\mathcal{X}}(X)\|_{\infty} \left(|g(\iota_{d_{\mathcal{X}}}(X))| + |g(\iota_{d_{\mathcal{X}}} \circ \widetilde{\mathcal{D}}_{\mathcal{X}} \circ \widetilde{\mathcal{E}}_{\mathcal{X}})|\right) \, \mathrm{d}\mu(X)$$

$$\leq L\|\iota_{d_{\mathcal{X}}} - \iota_{d_{\mathcal{X}}} \circ \widetilde{\mathcal{D}}_{\mathcal{X}} \circ \widetilde{\mathcal{E}}_{\mathcal{X}}\|_{L^2_{\mu}(\mathcal{X};\ell^{\infty}(\mathbb{N}))} (a + b).$$

We deduce that $a - b \leq \delta$. Since $z^*$ was arbitrary, we get

$$\|\|f\|\|_{L^2_{\tilde{\varsigma}}(D;\mathcal{Z})} = \sup_{z^* \in B(\mathcal{Z}^*)} \|z^*(f)\|_{L^2_{\tilde{\varsigma}}(D)} \leq \sup_{z^* \in B(\mathcal{Z}^*)} \|z^*(f)\|_{L^2_{\varsigma}(D)} + \delta = \|\|f\|\|_{L^2_{\varsigma}(D;\mathcal{Z})} + \delta,$$

which gives the upper bound. The same argument applied to $b^2 - a^2$ also gives the lower bound. $\square$

**Lemma D.4** (Norm equivalences for polynomials). *Let $(\mathcal{Z}, \|\cdot\|_{\mathcal{Z}})$ be any Banach space, $\varsigma$ be the measure defined in (A.I), $\tilde{\varsigma}$ be the measure defined in (A.III) and $S \subset \mathcal{F}$ with $\text{supp}(\boldsymbol{\nu}) \in \{1, \ldots, d_{\mathcal{X}}\}$, $\forall \boldsymbol{\nu} \in S$. Suppose that*

$$\sqrt{|S|_{\boldsymbol{v}}} \cdot \|\iota_{d_{\mathcal{X}}} - \iota_{d_{\mathcal{X}}} \circ \widetilde{\mathcal{D}}_{\mathcal{X}} \circ \widetilde{\mathcal{E}}_{\mathcal{X}}\|_{L^2_{\mu}(\mathcal{X};\ell^{\infty}(\mathbb{N}))} \leq c, \tag{D.6}$$

*for some sufficiently small universal constant $c > 0$. Then the norm equivalence*

$$\|\|p\|\|_{L^2_{\varrho}(D;\mathcal{Z})} \lesssim \|\|p\|\|_{L^2_{\tilde{\varsigma}}(D;\mathcal{Z})} \lesssim \|\|p\|\|_{L^2_{\varrho}(D;\mathcal{Z})}$$

*holds for all $p = \sum_{\boldsymbol{\nu} \in S} c_{\boldsymbol{\nu}} \Psi_{\boldsymbol{\nu}}$, where $c_{\boldsymbol{\nu}} \in \mathcal{Z}$.*

*Proof.* Combining Lemmas D.2 and D.3, we see that

$$\|\|p\|\|_{L^2_{\varsigma}(D;\mathcal{Z})} - \delta \leq \|\|p\|\|_{L^2_{\tilde{\varsigma}}(D;\mathcal{Z})} \leq \|\|p\|\|_{L^2_{\varsigma}(D;\mathcal{Z})} + \delta,$$

where $\delta = \frac{1}{2}\|\|p\|\|_{L^2_{\varrho}(D;\mathcal{Z})}\sqrt{|S|_{\boldsymbol{v}}}\|\iota_{d_{\mathcal{X}}} - \iota_{d_{\mathcal{X}}} \circ \widetilde{\mathcal{D}}_{\mathcal{X}} \circ \widetilde{\mathcal{E}}_{\mathcal{X}}\|_{L^2_{\mu}(\mathcal{X};\ell^{\infty}(\mathbb{N}))} \leq c\|\|p\|\|_{L^2_{\varrho}(D;\mathcal{Z})}/2$. By (A.I), there are constants $c_1 \geq c_2 > 0$ such that

$$c_1\|\|p\|\|_{L^2_{\varrho}(D;\mathcal{Z})} \leq \|\|p\|\|_{L^2_{\varsigma}(D;\mathcal{Z})} \leq c_2\|\|p\|\|_{L^2_{\varrho}(D;\mathcal{Z})}.$$

We now take $c = c_1/2$. $\square$

## D.3   Analysis of (D.3)

We now analyze (D.3). Our analysis relies on the following result, which shows an error bound subject to a certain discrete metric inequality (D.7).

**Lemma D.5** (Discrete metric inequality implies error bounds). *Let $\mathcal{Q} : \mathcal{Y} \to \widetilde{\mathcal{Y}} = \mathcal{D}_{\mathcal{Y}}(\mathbb{R}^{d_{\mathcal{Y}}})$ be a bounded linear operator and $\pi_{\mathcal{Q}} = \|\mathcal{Q}\|_{\mathcal{Y} \to \mathcal{Y}}$. Suppose that*

$$\|p - q\|_{\mathsf{disc},\bar{\varsigma}} \geq \alpha \max\{\|\|p - q\|\|_{L^2_\varrho(D;\mathcal{Y})}, \|\|p - q\|\|_{L^2_\varsigma(D;\mathcal{Y})}\}, \quad \forall p, q \in \mathcal{P}_{\mathcal{S};\widetilde{\mathcal{Y}}} \tag{D.7}$$

*for some $\alpha > 0$. Then, for any $F \in L^\infty_\mu(\mathcal{X}; \mathcal{Y})$, $p \in \mathcal{P}_{\mathcal{S};\widetilde{\mathcal{Y}}}$ and $q \in \mathcal{P}_{\mathcal{S};\mathcal{Y}}$, we have*

$$\|\|F - p \circ \mathcal{E}_{\mathcal{X}}\|\|_{L^2_\mu(\mathcal{X};\mathcal{Y})} \leq \|\|F - \mathcal{Q} \circ F\|\|_{L^2_\mu(\mathcal{X};\mathcal{Y})} + \alpha^{-1}\|p - \mathcal{Q} \circ q\|_{\mathsf{disc},\bar{\varsigma}}$$
$$+ \pi_{\mathcal{Q}}\|\|F - q \circ \mathcal{E}_{\mathcal{X}}\|\|_{L^2_\mu(\mathcal{X};\mathcal{Y})}$$

*and*

$$\|\|F - p \circ \mathcal{E}_{\mathcal{X}}\|\|_{L^\infty_\mu(\mathcal{X};\mathcal{Y})} \leq \|\|F - \mathcal{Q} \circ F\|\|_{L^\infty_\mu(\mathcal{X};\mathcal{Y})} + \sqrt{2k}/\alpha\|p - \mathcal{Q} \circ q\|_{\mathsf{disc},\bar{\varsigma}}$$
$$+ \pi_{\mathcal{Q}}\|\|F - q \circ \mathcal{E}_{\mathcal{X}}\|\|_{L^\infty_\mu(\mathcal{X};\mathcal{Y})}.$$

*Proof.* By the triangle inequality and properties of $\mathcal{Q}$, we have

$$\|\|F - p \circ \mathcal{E}_{\mathcal{X}}\|\|_{L^2_\mu(\mathcal{X};\mathcal{Y})}$$
$$\leq \|\|F - \mathcal{Q} \circ F\|\|_{L^2_\mu(\mathcal{X};\mathcal{Y})} + \|\|p \circ \mathcal{E}_{\mathcal{X}} - \mathcal{Q} \circ q \circ \mathcal{E}_{\mathcal{X}}\|\|_{L^2_\mu(\mathcal{X};\mathcal{Y})} + \|\|\mathcal{Q} \circ F - \mathcal{Q} \circ q \circ \mathcal{E}_{\mathcal{X}}\|\|_{L^2_\mu(\mathcal{X};\mathcal{Y})}$$
$$\leq \|\|F - \mathcal{Q} \circ F\|\|_{L^2_\mu(\mathcal{X};\mathcal{Y})} + \|\|p \circ \mathcal{E}_{\mathcal{X}} - \mathcal{Q} \circ q \circ \mathcal{E}_{\mathcal{X}}\|\|_{L^2_\mu(\mathcal{X};\mathcal{Y})} + \pi_{\mathcal{Q}}\|\|F - q \circ \mathcal{E}_{\mathcal{X}}\|\|_{L^2_\mu(\mathcal{X};\mathcal{Y})}.$$

Now, since $p, \mathcal{Q} \circ q \in \mathcal{P}_{\mathcal{S};\widetilde{\mathcal{Y}}}$, the second term can be bounded by

$$\|\|p \circ \mathcal{E}_{\mathcal{X}} - \mathcal{Q} \circ q \circ \mathcal{E}_{\mathcal{X}}\|\|_{L^2_\mu(\mathcal{X};\mathcal{Y})} = \|\|p - \mathcal{Q} \circ q\|\|_{L^2_\varsigma(D;\mathcal{Y})} \leq \alpha^{-1}\|p - \mathcal{Q} \circ q\|_{\mathsf{disc},\bar{\varsigma}}. \tag{D.8}$$

This yields the first result.

For the second result, we once more write

$$\|\|F - p \circ \mathcal{E}_{\mathcal{X}}\|\|_{L^\infty_\mu(\mathcal{X};\mathcal{Y})} \leq \|\|F - \mathcal{Q} \circ F\|\|_{L^\infty_\mu(\mathcal{X};\mathcal{Y})} + \|\|p \circ \mathcal{E}_{\mathcal{X}} - \mathcal{Q} \circ q \circ \mathcal{E}_{\mathcal{X}}\|\|_{L^\infty_\mu(\mathcal{X};\mathcal{Y})}$$
$$+ \pi_{\mathcal{Q}}\|\|F - q \circ \mathcal{E}_{\mathcal{X}}\|\|_{L^\infty_\mu(\mathcal{X};\mathcal{Y})}.$$

For the second term, we use (A.III) to write

$$\|\|p \circ \mathcal{E}_{\mathcal{X}} - \mathcal{Q} \circ q \circ \mathcal{E}_{\mathcal{X}}\|\|_{L^\infty_\mu(\mathcal{X};\mathcal{Y})} = \|\|p - \mathcal{Q} \circ q\|\|_{L^\infty_\varsigma(D;\mathcal{Y})} \leq \|\|p - \mathcal{Q} \circ q\|\|_{L^\infty_\varrho(D;\mathcal{Y})}.$$

Notice that $p - \mathcal{Q} \circ q$ is a polynomial supported in a set $S$ with $|S|_{\boldsymbol{u}} \leq 2k$. We now apply Lemma D.1 and (D.7) to obtain

$$\|\|p \circ \mathcal{E}_{\mathcal{X}} - \mathcal{Q} \circ q \circ \mathcal{E}_{\mathcal{X}}\|\|_{L^\infty_\mu(\mathcal{X};\mathcal{Y})} \leq \sqrt{2k}\|\|p - \mathcal{Q} \circ q\|\|_{L^2_\varrho(D;\mathcal{Y})} \leq \sqrt{2k}/\alpha\|p - \mathcal{Q} \circ q\|_{\mathsf{disc},\bar{\varsigma}}. \tag{D.9}$$

This gives the result. $\qquad\square$

**Theorem D.6** (Error bound for polynomial minimizers). *Let $\mathcal{Q} : \mathcal{Y} \to \widetilde{\mathcal{Y}} = \mathcal{D}_{\mathcal{Y}}(\mathbb{R}^{d_{\mathcal{Y}}})$ be a bounded linear operator, $\pi_{\mathcal{Q}} = \|\mathcal{Q}\|_{\mathcal{Y} \to \mathcal{Y}}$ and suppose that (D.7) holds. Let $\widehat{F}$ be as in (D.4) for some $(\sigma, \tau)$-approximate minimizer $\hat{p}$ of (D.3). Then*

$$\|\|F - \widehat{F}\|\|_{L^2_\mu(\mathcal{X};\mathcal{Y})} \leq \|\|F - \mathcal{Q} \circ F\|\|_{L^2_\mu(\mathcal{X};\mathcal{Y})} + \frac{\sigma+1}{\alpha}\|\|F - \mathcal{Q} \circ F\|\|_{\mathsf{disc},\mu}$$
$$+ \pi_{\mathcal{Q}}\left(\|\|F - q \circ \mathcal{E}_{\mathcal{X}}\|\|_{L^2_\mu(\mathcal{X};\mathcal{Y})} + \frac{\sigma+1}{\alpha}\|\|F - q \circ \mathcal{E}_{\mathcal{X}}\|\|_{\mathsf{disc},\mu}\right)$$
$$+ \frac{\tau}{\alpha} + \frac{\sigma+1}{\alpha\sqrt{m}}\|\boldsymbol{E}\|_{2;\mathcal{Y}}$$

*and*

$$\|F - \widehat{F}\|_{L^\infty_\mu(\mathcal{X};\mathcal{Y})} \le \|F - \mathcal{Q} \circ F\|_{L^\infty_\mu(\mathcal{X};\mathcal{Y})} + \frac{\sqrt{2k}(\sigma+1)}{\alpha}\|F - \mathcal{Q} \circ F\|_{\mathsf{disc},\mu}$$

$$+ \pi_\mathcal{Q}\left(\|F - q \circ \mathcal{E}_\mathcal{X}\|_{L^\infty_\mu(\mathcal{X};\mathcal{Y})} + \frac{\sqrt{2k}(\sigma+1)}{\alpha}\|F - q \circ \mathcal{E}_\mathcal{X}\|_{\mathsf{disc},\mu}\right)$$

$$+ \frac{\sqrt{2k}\tau}{\alpha} + \frac{\sqrt{2k}(\sigma+1)}{\alpha\sqrt{m}}\|\boldsymbol{E}\|_{2;\mathcal{Y}}$$

*for all $q \in \mathcal{P}_{\mathcal{S};\mathcal{Y}}$, where $\boldsymbol{E} = (E_i)_{i=1}^m \in \mathcal{Y}^m$ is the (Banach-valued) vector of noise terms.*

*Proof.* We apply the previous lemma with $p = \hat{p}$. This gives

$$\|F - \widehat{F}\|_{L^2_\mu(\mathcal{X};\mathcal{Y})} \le \|F - \mathcal{Q} \circ F\|_{L^2_\mu(\mathcal{X};\mathcal{Y})} + \alpha^{-1}\|\hat{p} - \mathcal{Q} \circ q\|_{\mathsf{disc},\varsigma}$$

$$+ \pi_\mathcal{Q}\|F - q \circ \mathcal{E}_\mathcal{X}\|_{L^2_\mu(\mathcal{X};\mathcal{Y})} \tag{D.10}$$

$$\|F - \widehat{F}\|_{L^\infty_\mu(\mathcal{X};\mathcal{Y})} \le \|F - \mathcal{Q} \circ F\|_{L^\infty_\mu(\mathcal{X};\mathcal{Y})} + \sqrt{2k}/\alpha\|\hat{p} - \mathcal{Q} \circ q\|_{\mathsf{disc},\varsigma}$$

$$+ \pi_\mathcal{Q}\|F - q \circ \mathcal{E}_\mathcal{X}\|_{L^\infty_\mu(\mathcal{X};\mathcal{Y})}.$$

Consider the second term. We have

$$\|\hat{p} - \mathcal{Q} \circ q\|_{\mathsf{disc},\varsigma} = \|\hat{p} \circ \mathcal{E}_\mathcal{X} - \mathcal{Q} \circ q \circ \mathcal{E}_\mathcal{X}\|_{\mathsf{disc},\mu} \le \|F - \mathcal{Q} \circ q \circ \mathcal{E}_\mathcal{X}\|_{\mathsf{disc},\mu} + \|F - \hat{p} \circ \mathcal{E}_\mathcal{X}\|_{\mathsf{disc},\mu}.$$

Consider the second term of this expression. By the triangle inequality and the facts that $\hat{p}$ is a $(\sigma, \tau)$-minimizer and $\mathcal{Q} \circ q$ is feasible, we obtain

$$\|F - \hat{p} \circ \mathcal{E}_\mathcal{X}\|_{\mathsf{disc},\mu} \le \sqrt{\frac{1}{m}\sum_{i=1}^m \|Y_i - \hat{p} \circ \mathcal{E}_\mathcal{X}\|_\mathcal{Y}^2} + \frac{1}{\sqrt{m}}\|\boldsymbol{E}\|_{2;\mathcal{Y}}$$

$$\le \sigma\sqrt{\frac{1}{m}\sum_{i=1}^m \|Y_i - \mathcal{Q} \circ q \circ \mathcal{E}_\mathcal{X}\|_\mathcal{Y}^2 + \tau} + \frac{1}{\sqrt{m}}\|\boldsymbol{E}\|_{2;\mathcal{Y}}$$

$$\le \sigma\|F - \mathcal{Q} \circ q \circ \mathcal{E}_\mathcal{X}\|_{\mathsf{disc},\mu} + \tau + \frac{\sigma+1}{\sqrt{m}}\|\boldsymbol{E}\|_{2;\mathcal{Y}}.$$

Therefore, we get

$$\|\hat{p} - \mathcal{Q} \circ q\|_{\mathsf{disc},\varsigma} \le (\sigma+1)\|F - \mathcal{Q} \circ q \circ \mathcal{E}_\mathcal{X}\|_{\mathsf{disc},\mu} + \tau + \frac{\sigma+1}{\sqrt{m}}\|\boldsymbol{E}\|_{2;\mathcal{Y}}.$$

We now estimate the first term in this expression as follows:

$$\|F - \mathcal{Q} \circ q \circ \mathcal{E}_\mathcal{X}\|_{\mathsf{disc},\mu} \le \|F - \mathcal{Q} \circ F\|_{\mathsf{disc},\mu} + \|\mathcal{Q} \circ F - \mathcal{Q} \circ q \circ \mathcal{E}_\mathcal{X}\|_{\mathsf{disc},\mu}$$

$$\le \|F - \mathcal{Q} \circ F\|_{\mathsf{disc},\mu} + \pi_\mathcal{Q}\|F - q \circ \mathcal{E}_\mathcal{X}\|_{\mathsf{disc},\mu}.$$

Therefore, we conclude that

$$\|\hat{p} - \mathcal{Q} \circ q\|_{\mathsf{disc},\varsigma} \le (\sigma+1)\|F - \mathcal{Q} \circ F\|_{\mathsf{disc},\mu} + (\sigma+1)\pi_\mathcal{Q}\|F - q \circ \mathcal{E}_\mathcal{X}\|_{\mathsf{disc},\mu} + \tau + \frac{\sigma+1}{\sqrt{m}}\|\boldsymbol{E}\|_{2;\mathcal{Y}}.$$

Combining this with (D.10) now gives the result. $\qquad\square$

### D.4  Ensuring (D.7) holds with high probability

For the proof of the next lemma and subsequent steps of the proof, we let $\Lambda = \{\boldsymbol{\nu}_1, \ldots, \boldsymbol{\nu}_N\}$ and define the matrix

$$\boldsymbol{A} = \left(\frac{\Psi_{\boldsymbol{\nu}_j}(\mathcal{E}_\mathcal{X}(X_i))}{\sqrt{m}}\right)_{i,j=1}^{m,N} \in \mathbb{R}^{m \times N}.$$

**Lemma D.7.** *Let $0 < \epsilon < 1$, $0 < \delta < 1$, $k > 0$ and suppose that*

$$m \geq c_0 \cdot \delta^{-2} \cdot k \cdot (\log(eN) \cdot \log^2(k/\delta) + \log(2/\epsilon)) \tag{D.11}$$

*for some universal constant $c_0 > 0$. Let $T = \{\boldsymbol{c} \in \mathbb{R}^N : \|\boldsymbol{c}\|_2 = 1, \ \|\boldsymbol{c}\|_{0,\boldsymbol{v}} \leq k\}$ and*

$$\theta_+ = \sup\left\{\mathbb{E}\|\boldsymbol{A}\boldsymbol{c}\|_2^2 : \boldsymbol{c} \in T\right\}, \quad \theta_- = \inf\left\{\mathbb{E}\|\boldsymbol{A}\boldsymbol{c}\|_2^2 : \boldsymbol{c} \in T\right\}.$$

*Then*

$$\|\boldsymbol{A}\boldsymbol{c}\|_2^2 \geq (\theta_- - (1+\theta_+)c_1\delta)\|\boldsymbol{c}\|_2^2, \quad \forall \boldsymbol{c} \in \mathbb{R}^N, \ \|\boldsymbol{c}\|_{0,\boldsymbol{v}} \leq k.$$

*with probability at least $1 - \epsilon$, where $c_1 > 0$ is a universal constant.*

*Proof.* The proof is based on [12, Thm. 2.13]. Since $\|\boldsymbol{c}\|_{1,\boldsymbol{v}} \leq \sqrt{\|\boldsymbol{c}\|_{0,\boldsymbol{v}}}\|\boldsymbol{c}\|_2 \leq \sqrt{k}$, we see that

$$T \subseteq \left\{\boldsymbol{c} \in \mathbb{R}^N : \|\boldsymbol{c}\|_{1,\boldsymbol{v}} \leq \sqrt{k}\right\}.$$

Define the random vector $\boldsymbol{X} \in \mathbb{R}^N$ by $\boldsymbol{X} = (\Psi_{\boldsymbol{\nu}_i}(\mathcal{E}_{\mathcal{X}}(X)))_{i=1}^N$ for $X \sim \mu$. Observe that

$$|\langle \boldsymbol{X}, \boldsymbol{e}_j \rangle| = |\Psi_{\boldsymbol{\nu}_i}(\mathcal{E}_{\mathcal{X}}(X))| \leq u_{\boldsymbol{\nu}_i} \leq v_{\boldsymbol{\nu}_i}$$

almost surely, by (A.III) and the definition of the weights. Therefore, by [12, Thm. 2.13], if

$$m \geq c_0 \cdot \delta^{-2} \cdot k \cdot \log(eN) \cdot \log^2(k/(c_1\delta)), \tag{D.12}$$

it holds that

$$\sup_{\boldsymbol{c} \in T}\left|\frac{1}{m}\sum_{i=1}^m |\langle \boldsymbol{c}, \boldsymbol{X}_i \rangle|^2 - \mathbb{E}|\langle \boldsymbol{c}, \boldsymbol{X} \rangle|^2\right| \leq c_1\delta\left(1 + \sup_{\boldsymbol{c} \in T}\mathbb{E}|\langle \boldsymbol{c}, \boldsymbol{X} \rangle|^2\right)$$

with probability at least $1 - 2\exp(-c_2\delta^2 m/k)$. Here $c_0, c_1, c_2 > 0$ are universal constants. Now observe that

$$\frac{1}{m}\sum_{i=1}^m |\langle \boldsymbol{c}, \boldsymbol{X}_i \rangle|^2 = \|\boldsymbol{A}\boldsymbol{c}\|_2^2, \qquad \mathbb{E}|\langle \boldsymbol{c}, \boldsymbol{X} \rangle|^2 = \mathbb{E}(\|\boldsymbol{A}\boldsymbol{c}\|_2^2).$$

Therefore, we have shown that

$$\|\boldsymbol{A}\boldsymbol{c}\|_2^2 \geq \theta_- - (1+\theta_+)c_1\delta, \quad \forall \boldsymbol{c} \in T,$$

with probability at least $1 - 2\exp(-c_2\delta^2 m/k)$, provided $m$ satisfies (D.12). To conclude the result, we observe that (D.11) implies (D.12), up to a possible change in the universal constant $c_0$. Moreover, it also implies that

$$2\exp(-c_2\delta^2 m/k) \leq \epsilon.$$

Hence we obtain the result. $\qquad\square$

**Lemma D.8.** *There exist universal constants $c_0, c_1, c_2 > 0$ such that the following holds. Suppose that*

$$m \geq c_0 \cdot k \cdot (\log(eN) \cdot \log^2(k) + \log(2/\epsilon)) \tag{D.13}$$

*and*

$$\sqrt{k} \cdot \|\iota_{d_{\mathcal{X}}} - \iota_{d_{\mathcal{X}}} \circ \widetilde{\mathcal{D}}_{\mathcal{X}} \circ \widetilde{\mathcal{E}}_{\mathcal{X}}\|_{L_\mu^2(\mathcal{X};\ell^\infty(\mathbb{N}))} \leq c_1. \tag{D.14}$$

*Then* (D.7) *holds with probability at least $1 - \epsilon$ and constant $\alpha \geq c_2$.*

*Proof.* We shall apply the previous lemma with $k$ replaced by $2k$. First, we estimate $\theta_+$ and $\theta_-$. Let $\boldsymbol{c} \in T$, with $T$ as defined therein with $2k$ in place of $k$. Write $p = \sum_{\boldsymbol{\nu} \in \Lambda} c_{\boldsymbol{\nu}}\Psi_{\boldsymbol{\nu}}$ for the corresponding (scalar-valued) polynomial. Then

$$\|\boldsymbol{A}\boldsymbol{c}\|_2^2 = \frac{1}{m}\sum_{i=1}^m |p \circ \mathcal{E}_{\mathcal{X}}(X_i)|^2,$$

and therefore

$$\mathbb{E}\|\boldsymbol{A}\boldsymbol{c}\|_2^2 = \|p\|_{L_\varsigma^2(D)}^2.$$

Since $\|c\|_{0,v} \leq 2k$, (D.14) implies (D.6). We now apply Lemma D.4 and the fact that $\|p\|_{L^2_\varrho(D)} = \|c\|_2 = 1$ to get

$$c_3 \leq \mathbb{E}\|Ac\|_2^2 \leq c_4$$

for universal constants $c_4 \geq c_3 > 0$. Since $c \in T$ was arbitrary to deduce that

$$c_3 \leq \theta_- \leq \theta_+ \leq c_4. \tag{D.15}$$

We now show that (D.7) holds with the desired probability. Let $p, q \in \mathcal{P}_{\mathcal{S};\widetilde{\mathcal{Y}}}$ be arbitrary. Then their difference $h = p - q$ can be expressed as

$$h = \sum_{\boldsymbol{\nu} \in \Lambda} d_{\boldsymbol{\nu}} \Psi_{\boldsymbol{\nu}} \tag{D.16}$$

where $\boldsymbol{d} = (d_{\boldsymbol{\nu}})_{\boldsymbol{\nu} \in \Lambda} \in \widetilde{\mathcal{Y}}^N$ satisfies

$$\|\boldsymbol{d}\|_{0,u} \leq \|\boldsymbol{d}\|_{0,v} \leq 2k.$$

Now, this, (D.14) and Lemma D.4 imply that

$$\|\|h\|\|_{L^2_{\check{\varsigma}}(D;\mathcal{Y})} \leq c_4 \|\|h\|\|_{L^2_\varrho(D;\mathcal{Y})}.$$

Therefore, in order to prove (D.7), it suffices to show that $\|\|h\|\|_{L^2_\varrho(D;\mathcal{Y})} \lesssim \|h\|_{\mathsf{disc},\check{\varsigma}}$.

Observe that

$$\|h\|_{\mathsf{disc},\check{\varsigma}} = \|\boldsymbol{Ad}\|_{2;\mathcal{Y}} \geq \|\|\boldsymbol{Ad}\|\|_{2;\mathcal{Y}} = \sup_{y^* \in B(\mathcal{Y}^*)} \|y^*(\boldsymbol{Ad})\|_2,$$

where we recall that for a vector $\boldsymbol{z} = (z_i)_{i=1}^N \in \mathcal{Y}^N$, we write $y^*(\boldsymbol{z}) = (y^*(z_i))_{i=1}^N \in \mathbb{R}^N$. By linearity, we have $y^*(\boldsymbol{Ad}) = Ay^*(\boldsymbol{d})$. Therefore, by Lemma D.7,

$$
\begin{aligned}
\|h\|_{\mathsf{disc},\check{\varsigma}}^2 = \|\boldsymbol{Ad}\|_{2;\mathcal{Y}}^2 \\
&\geq \sup_{y^* \in B(\mathcal{Y}^*)} \|Ay^*(\boldsymbol{d})\|_2^2 \\
&\geq \sup_{y^* \in B(\mathcal{Y}^*)} (\theta_- - (1 + \theta_+)c_5\delta) \|y^*(\boldsymbol{d})\|_2^2 \\
&\geq (c_3 - (1 + c_4)c_5\delta) \sup_{y^* \in B(\mathcal{Y}^*)} \|y^*(h)\|_{L^2_\varrho(D)}^2 \\
&= (c_3 - (1 + c_4)c_5\delta) \|\|h\|\|_{L^2_\varrho(D;\mathcal{Y})}^2.
\end{aligned}
$$

for some universal constant $c_5 > 0$, provided $m$ satisfies (D.11). We now set $\delta = c_3/(2(1 + c_4)c_5)$ to get

$$\|h\|_{\mathsf{disc},\check{\varsigma}}^2 \geq c_3/2 \|\|h\|\|_{L^2_\varrho(D;\mathcal{Y})}^2,$$

provided $m$ satisfies (D.11) with this value of $\delta$. However, this is implied by the condition (D.13). We deduce the result. $\qquad\square$

## D.5  Construction of the DNN family $\mathcal{N}$

Recall that $\Lambda \subset \mathcal{F}$, $|\Lambda| = N$ is an arbitrary set and $\mathcal{S}$ is defined by (D.1). In this and what follows, we slightly abuse notation and consider a DNN $N : \mathbb{R}^d \to \mathbb{R}$ as a function $\mathbb{R}^{\mathbb{N}} \to \mathbb{R}$ which depends on only the first $d$ variables.

**Lemma D.9** (Approximating Legendre polynomials with tanh DNNs). *Let $\Gamma \subset \Lambda$ with $\mathrm{supp}(\boldsymbol{\nu}) \subseteq \{1, \ldots, d\}$, $\forall \boldsymbol{\nu} \in \Gamma$, and $m(\Gamma) = \max_{\boldsymbol{\nu} \in \Gamma} \|\boldsymbol{\nu}\|_1 < \infty$. There exists a fully-connected family $\mathcal{N}_o$ of tanh DNNs $N : \mathbb{R}^d \to \mathbb{R}$ with*

$$\mathrm{width}(\mathcal{N}_o) \lesssim m(\Gamma), \quad \mathrm{depth}(\mathcal{N}_o) \lesssim \log(m(\Gamma)),$$

*such that, for any $0 < \delta < 1$ and $\boldsymbol{\nu} \in \Gamma$, there is a DNN $N_{\boldsymbol{\nu}} \in \mathcal{N}_o$ with*

$$\|N_{\boldsymbol{\nu}} - \Psi_{\boldsymbol{\nu}}\|_{L^\infty_\varrho(D)} \leq \delta. \tag{D.17}$$

*Moreover, the zero network $0 : \boldsymbol{x} \mapsto 0$ also belongs to $\mathcal{N}_o$ (trivially, since $\tanh(0) = 0$).*

*Proof.* We follow the argument of [5, Thm. 7.4], which is based on [79, Prop. 2.6]. Let $\boldsymbol{\nu} \in \Gamma$. Then via the fundamental theorem of algebra we can write $\Psi_{\boldsymbol{\nu}}(\boldsymbol{x})$ as a product of $\|\boldsymbol{\nu}\|_1$ numbers as

$$\Psi_{\boldsymbol{\nu}}(\boldsymbol{x}) = \prod_{i \in \text{supp}(\boldsymbol{\nu})} \prod_{j=1}^{\nu_i} d_i(x_i - r_{ij}).$$

Here $\{r_{ij}\}_{j=1}^{\nu_i}$ are the roots of the univariate Legendre polynomial $P_{\nu_i}$. We append $m(\Gamma) - \|\boldsymbol{\nu}\|_1$ ones and write $\Psi_{\boldsymbol{\nu}}(\boldsymbol{x})$ as a product of exactly $m(\Gamma)$ numbers. Now define the affine map

$$\mathcal{A}_{\boldsymbol{\nu}} : \mathbb{R}^d \to \mathbb{R}^{m(\Gamma)}, \tag{D.18}$$

so that $\mathcal{A}_{\boldsymbol{\nu}}(\boldsymbol{x})$ is the vector consisting of the values $d_i(x_i - r_{ij})$ for $j = 1, \ldots, \nu_i$ and $i \in \text{supp}(\boldsymbol{\nu})$ and 1 otherwise. To complete the proof, we need to construct a tanh DNN mapping $\mathbb{R}^{m(\Gamma)} \to \mathbb{R}$ that approximately multiplies these numbers. To do this, we argue as in the proof of [5, Thm. 7.4] to see that there is a tanh DNN $N_{\boldsymbol{\nu}}$ with

$$\text{width}(N_{\boldsymbol{\nu}}) \lesssim m(\Gamma), \quad \text{depth}(N_{\boldsymbol{\nu}}) \lesssim \log(m(\Gamma))$$

that satisfies the desired bound (D.17). Since these width and depth bounds are independent of $\boldsymbol{\nu}$, we deduce the result. $\qquad\square$

Fix $\delta > 0$, let $\Gamma = \cup_{S \in \mathcal{S}} S$, $d = d_{\mathcal{X}}$ and consider the corresponding family $\mathcal{N}_o$ and DNNs $N_{\boldsymbol{\nu}}$, $\boldsymbol{\nu} \in \Gamma$, asserted by this lemma. For any $\boldsymbol{\nu} \in \Gamma$, we have $\boldsymbol{\nu} \in S$ for some $S \in \mathcal{S}$, and any such $S$ satisfies $|S|_{\boldsymbol{v}} \leq k$. Therefore $u_{\boldsymbol{\nu}}^{2(5+\xi)} = v_{\boldsymbol{\nu}}^2 \leq k$ for any $\boldsymbol{\nu} \in S$ and we deduce that

$$\|\boldsymbol{\nu}\|_1 \leq \prod_{j=1}^d (2\nu_j + 1) = u_{\boldsymbol{\nu}}^2 \leq k^{1/(5+\xi)}.$$

This implies that $m(\Gamma) \leq k^{1/(5+\xi)}$ and therefore

$$\text{width}(\mathcal{N}_o) \lesssim k^{1/(5+\xi)}, \quad \text{depth}(\mathcal{N}_o) \lesssim \log(k).$$

With this in hand, we now define the family $\mathcal{N}$ of DNNs $N : \mathbb{R}^{d_{\mathcal{X}}} \to \mathbb{R}^{d_{\mathcal{Y}}}$ by

$$\mathcal{N} = \left\{ N = \boldsymbol{C} \begin{bmatrix} N_{\boldsymbol{\nu}_1} \\ \vdots \\ N_{\boldsymbol{\nu}_{|S|}} \\ 0 \\ \vdots \\ 0 \end{bmatrix} : S = \{\boldsymbol{\nu}_1, \ldots, \boldsymbol{\nu}_{|S|}\} \in \mathcal{S}, \ \boldsymbol{C} \in \mathbb{R}^{d_{\mathcal{Y}} \times \lfloor k \rfloor} \right\}. \tag{D.19}$$

Here we also use the fact that $|S| \leq |S|_{\boldsymbol{v}} \leq k$. Notice that this family satisfies

$$\text{width}(\mathcal{N}) \leq k^{1+1/(5+\xi)}, \quad \text{depth}(\mathcal{N}) \lesssim \log(k), \tag{D.20}$$

due to the bounds for $\mathcal{N}_o$.

Now let $N \in \mathcal{N}$ and write $\boldsymbol{C} = [\boldsymbol{c}_1 | \cdots | \boldsymbol{c}_{\lfloor k \rfloor}]$, where $\boldsymbol{c}_i \in \mathbb{R}^{d_{\mathcal{Y}}}$. Then

$$\mathcal{D}_{\mathcal{Y}} \circ N = \sum_{i=1}^{|S|} \mathcal{D}_{\mathcal{Y}}(\boldsymbol{c}_i) N_{\boldsymbol{\nu}_i} = \sum_{i=1}^{|S|} c_i N_{\boldsymbol{\nu}_i}, \quad \text{where } c_i \in \widetilde{\mathcal{Y}}.$$

Therefore, we can associate $\mathcal{N}$ with the space of functions $\widetilde{P}_{\mathcal{S};\widetilde{\mathcal{Y}}}$, where, for any arbitrary Banach space $(\mathcal{Z}, \|\cdot\|_{\mathcal{Z}})$,

$$\widetilde{\mathcal{P}}_{\mathcal{S};\mathcal{Z}} = \left\{ \sum_{\boldsymbol{\nu} \in S} c_{\boldsymbol{\nu}} N_{\boldsymbol{\nu}} : c_{\boldsymbol{\nu}} \in \mathcal{Z}, \ S \in \mathcal{S} \right\}.$$

We now require the following lemma, which relates the distance between a function in $\widetilde{\mathcal{P}}_{\mathcal{S};\mathcal{Z}}$ and the corresponding polynomial in $\mathcal{P}_{\mathcal{S};\mathcal{Z}}$.

**Lemma D.10** (Discrete norms of polynomials and their approximating DNNs)**.** *Let $S \subset \mathcal{F}$, $(\mathcal{Z}, \|\cdot\|_{\mathcal{Z}})$ be any Banach space and suppose that $p = \sum_{\boldsymbol{\nu} \in S} c_{\boldsymbol{\nu}} \Psi_{\boldsymbol{\nu}}$, where $c_{\boldsymbol{\nu}} \in \mathcal{Z}$. Define*

$$\tilde{p} = \sum_{\boldsymbol{\nu} \in S} c_{\boldsymbol{\nu}} N_{\boldsymbol{\nu}}.$$

*Then*

$$\|p - \tilde{p}\|_{\mathsf{disc}, \tilde{\varsigma}} \leq \|p - \tilde{p}\|_{L^{\infty}_{\varrho}(D; \mathcal{Z})} \leq \delta \sqrt{|S|} \|\!|p|\!\|_{L^2_{\varrho}(D; \mathcal{Z})}. \tag{D.21}$$

*Moreover, if $\mathcal{Z} = \widetilde{\mathcal{Y}}$, $S \in \mathcal{S}$ and (D.7) holds with $\alpha$ satisfying $\delta \sqrt{|S|_{\boldsymbol{u}}}/\alpha < 1$ then*

$$\|\!|p|\!\|_{L^2_{\varrho}(D; \mathcal{Y})} \leq \frac{1}{\alpha - \delta \sqrt{|S|}} \|\tilde{p}\|_{\mathsf{disc}, \tilde{\varsigma}}.$$

*Proof.* By (A.III) and the definition of the $N_{\boldsymbol{\nu}}$ we have

$$\begin{aligned}
\|p - \tilde{p}\|_{\mathsf{disc}, \tilde{\varsigma}} &\leq \|p - \tilde{p}\|_{L^{\infty}_{\varrho}(D; \mathcal{Z})} \\
&= \sup_{z^* \in B(\mathcal{Z}^*)} \|z^*(p - \tilde{p})\|_{L^{\infty}_{\varrho}(D)} \\
&\leq \sup_{z^* \in B(\mathcal{Z}^*)} \sum_{\boldsymbol{\nu} \in S} |z^*(c_{\boldsymbol{\nu}})| \|\Psi_{\boldsymbol{\nu}} - N_{\boldsymbol{\nu}}\|_{L^{\infty}_{\varrho}(D)} \\
&\leq \delta \|\!|\boldsymbol{c}|\!\|_{1; \mathcal{Z}}.
\end{aligned}$$

We now apply the Cauchy–Schwarz inequality and Parseval's identity to obtain

$$\|p - \tilde{p}\|_{\mathsf{disc}, \tilde{\varsigma}} \leq \delta \sqrt{|S|} \|\!|\boldsymbol{c}|\!\|_{2; \mathcal{Z}} = \delta \sqrt{|S|_{\boldsymbol{u}}} \|\!|p|\!\|_{L^2_{\varrho}(D; \mathcal{Z})},$$

which gives the first result.

For the second result, we apply (D.7) with $q = 0$ to get

$$\|\!|p|\!\|_{L^2_{\varrho}(D; \mathcal{Y})} \leq \alpha^{-1} \|p\|_{\mathsf{disc}, \tilde{\varsigma}} \leq \alpha^{-1} \left( \|p - \tilde{p}\|_{\mathsf{disc}, \tilde{\varsigma}} + \|\tilde{p}\|_{\mathsf{disc}, \tilde{\varsigma}} \right).$$

Using (D.21) and the fact that $\delta \sqrt{|S|}/\alpha < 1$ now gives the result. $\qquad\square$

**Remark D.11 (Other activation functions)** As seen in this section, a key step in our proofs is emulating the polynomials via DNNs of quantifiable width and depth. There is an extensive literature on this topic. See, e.g., [5, 21, 23, 25, 38, 57, 63, 67, 72, 78, 79, 85, 86, 92, 97] and references therein. The proof of Lemma D.9 reduces this to the task of emulating the multiplication operation $(x_1, \ldots, x_d) \in \mathbb{R}^d \mapsto x_1 \cdots x_d \in \mathbb{R}$ via a DNN. As shown in the proof of [5, Thm. 7.4] (which is based on [79, Prop. 2.6]), this can in turn be achieved using a binary tree of $\lceil \log_2(d) \rceil$ DNNs that approximately compute the multiplication of two numbers $(x, y) \in \mathbb{R}^2 \mapsto xy \in \mathbb{R}$. Further, this task can be achieved via the identity $xy = ((x + y)^2 - (x - y)^2)/4$ by using a DNN that approximately computes the squaring function $x \in \mathbb{R} \mapsto x^2 \in \mathbb{R}$. To summarize, provide a DNN of quantifiable width and depth can approximately compute the squaring function, it can also approximately emulate the multivariate Legendre polynomials.

In view of this, we can adapt our main theorems to various other activation functions without change. This includes Rectified Polynomial Units (RePUs), where the emulation is, in fact, exact (see, e.g., [57, Lem. 2.1]). It also includes the Exponential Linear Unit (ELU) used in our numerical experiments and many others. See, for instance, Proposition 4.7 of [38] and the ensuing discussion. Rectified Linear Units (ReLUs) are slightly different, as in this case the depth of the DNN that performs the approximation multiplication depends on the desired accuracy (see, e.g., [79, Prop. 2.6]). One could modify Theorem 3.1 to consider ReLU DNNs, with the result being a worse depth bound than that presented for tanh DNNs.

### D.6  Analysis of (2.5)

**Lemma D.12** (Approximate minimizers of (2.5) yield approximate minimizers of (D.3))**.** *Suppose that (D.7) holds, let $\mathcal{N}$ be the family of DNNs defined in §D.5 and $\widehat{N}$ be any $(\sigma, \tau)$-approximate minimizer of (2.5). Let*

$$\mathcal{D}_{\mathcal{Y}} \circ \widehat{N} = \sum_{\boldsymbol{\nu} \in S} \hat{c}_{\boldsymbol{\nu}} N_{\boldsymbol{\nu}} \in \widetilde{\mathcal{P}}_{\mathcal{S}; \widetilde{\mathcal{Y}}},$$

*where $S \in \mathcal{S}$ and $\hat{c}_{\boldsymbol{\nu}} \in \widetilde{\mathcal{Y}}$, and define*

$$\hat{p} = \sum_{\boldsymbol{\nu} \in S} \hat{c}_{\boldsymbol{\nu}} \Psi_{\boldsymbol{\nu}}.$$

*Then $\hat{p}$ is a $(\sigma', \tau')$-approximate minimizer of* (D.3), *where*

$$
\begin{aligned}
\sigma' &\leq \sigma(1 + \delta\sqrt{k}/\alpha) \\
\tau' &\leq \tau + \sigma\delta\sqrt{k}/\alpha \left( \|F\|_{L^\infty_\mu(\mathcal{X};\mathcal{Y})} + \frac{1}{\sqrt{m}}\|\boldsymbol{E}\|_{2;\mathcal{Y}} \right) + \delta\sqrt{k}\|\hat{p}\|_{L^2_\varrho(D;\mathcal{Y})}
\end{aligned}
\tag{D.22}
$$

*Proof.* Let $p = \sum_{\boldsymbol{\nu} \in S} c_{\boldsymbol{\nu}} \Psi_{\boldsymbol{\nu}} \in P_{\mathcal{S};\widetilde{\mathcal{Y}}}$ be arbitrary and $N \in \mathcal{N}$ be the corresponding DNN so that $\mathcal{D}_{\mathcal{Y}} \circ N = \sum_{\boldsymbol{\nu} \in S} c_{\boldsymbol{\nu}} N_{\boldsymbol{\nu}}$. Then by the triangle inequality and the fact that $\widehat{N}$ is an approximate minimizer, we have

$$\sqrt{\frac{1}{m} \sum_{i=1}^m \|Y_i - \hat{p} \circ \mathcal{E}_{\mathcal{X}}(X_i)\|_{\mathcal{Y}}^2}$$

$$\leq \sqrt{\frac{1}{m} \sum_{i=1}^m \|Y_i - \mathcal{D}_{\mathcal{Y}} \circ \widehat{N} \circ \mathcal{E}_{\mathcal{X}}(X_i)\|_{\mathcal{Y}}^2} + \|\hat{p} - \mathcal{D}_{\mathcal{Y}} \circ \widehat{N}\|_{\mathsf{disc},\widetilde{\varsigma}}$$

$$\leq \sigma \sqrt{\frac{1}{m} \sum_{i=1}^m \|Y_i - \mathcal{D}_{\mathcal{Y}} \circ N \circ \mathcal{E}_{\mathcal{X}}(X_i)\|_{\mathcal{Y}}^2} + \tau + \|\hat{p} - \mathcal{D}_{\mathcal{Y}} \circ \widehat{N}\|_{\mathsf{disc},\widetilde{\varsigma}}$$

$$\leq \sigma \sqrt{\frac{1}{m} \sum_{i=1}^m \|Y_i - p \circ \mathcal{E}_{\mathcal{X}}(X_i)\|_{\mathcal{Y}}^2} + \tau + \sigma\|p - \mathcal{D}_{\mathcal{Y}} \circ N\|_{\mathsf{disc},\widetilde{\varsigma}} + \|\hat{p} - \mathcal{D}_{\mathcal{Y}} \circ \widehat{N}\|_{\mathsf{disc},\widetilde{\varsigma}}.$$

We now apply Lemma D.10 to the last two terms, noting that $|S| \leq |S|_{\boldsymbol{v}} \leq k$ since $S \in \mathcal{S}$, to obtain

$$\sqrt{\frac{1}{m} \sum_{i=1}^m \|Y_i - \hat{p} \circ \mathcal{E}_{\mathcal{X}}(X_i)\|_{\mathcal{Y}}^2} \leq \sigma \sqrt{\frac{1}{m} \sum_{i=1}^m \|Y_i - p \circ \mathcal{E}_{\mathcal{X}}(X_i)\|_{\mathcal{Y}}^2}$$

$$+ \tau + \sigma\delta\sqrt{k}\|p\|_{L^2_\varrho(D;\mathcal{Y})} + \delta\sqrt{k}\|\hat{p}\|_{L^2_\varrho(D;\mathcal{Y})}.$$

Consider the third term. We first apply (D.7) with $q = 0$ to get

$$\|p\|_{L^2_\varrho(D;\mathcal{Y})} \leq \alpha^{-1}\|p\|_{\mathsf{disc},\widetilde{\varsigma}} = \alpha^{-1}\|p \circ \mathcal{E}_{\mathcal{X}}\|_{\mathsf{disc},\mu}.$$

We then use the triangle inequality to get

$$\|p \circ \mathcal{E}_{\mathcal{X}}\|_{\mathsf{disc},\mu} \leq \|F - p \circ \mathcal{E}_{\mathcal{X}}\|_{\mathsf{disc},\mu} + \|F\|_{\mathsf{disc},\mu}$$

$$\leq \sqrt{\frac{1}{m} \sum_{i=1}^m \|Y_i - p \circ \mathcal{E}_{\mathcal{X}}(X_i)\|_{\mathcal{Y}}^2} + \|F\|_{L^\infty_\mu(\mathcal{X};\mathcal{Y})} + \frac{1}{\sqrt{m}}\|\boldsymbol{E}\|_{2;\mathcal{Y}}.$$

Therefore, we obtain

$$\sqrt{\frac{1}{m} \sum_{i=1}^m \|Y_i - \hat{p} \circ \mathcal{E}_{\mathcal{X}}(X_i)\|_{\mathcal{Y}}^2} \leq \sigma(1 + \delta\sqrt{k}/\alpha) \sqrt{\frac{1}{m} \sum_{i=1}^m \|Y_i - p \circ \mathcal{E}_{\mathcal{X}}(X_i)\|_{\mathcal{Y}}^2} + \tau$$

$$+ \sigma\delta\sqrt{k}/\alpha \left( \|F\|_{L^\infty_\mu(\mathcal{X};\mathcal{Y})} + \frac{1}{\sqrt{m}}\|\boldsymbol{E}\|_{2;\mathcal{Y}} \right) + \delta\sqrt{k}\|\hat{p}\|_{L^2_\varrho(D;\mathcal{Y})}.$$

Since $p \in P_{\mathcal{S};\widetilde{\mathcal{Y}}}$ was arbitrary we get the result. $\qquad\square$

**Theorem D.13** (Error bound for DNN minimizers). *Let $\mathcal{Q} : \mathcal{Y} \to \widetilde{\mathcal{Y}} = \mathcal{D}_{\mathcal{Y}}(\mathbb{R}^{d_{\mathcal{Y}}})$ be a bounded linear operator, $\pi_{\mathcal{Q}} = \|\mathcal{Q}\|_{\mathcal{Y} \to \mathcal{Y}}$ and suppose that* (D.7) *holds with $\alpha \geq c_0$ and $\alpha - \delta\sqrt{k} \geq c_1$ for suitable universal constants $c_0, c_1 > 0$. Let $\mathcal{N}$ be the family of DNNs defined in §D.5 and $\widehat{N}$ be any $(\sigma, \tau)$-approximate minimizer of* (2.5). *Then the approximation $\widehat{F} = \mathcal{D}_{\mathcal{Y}} \circ \widehat{N} \circ \mathcal{E}_{\mathcal{X}}$ satisfies*

$$
\begin{aligned}
\|\!|F - \widehat{F}|\!\|_{L^2_{\mu}(\mathcal{X};\mathcal{Y})} &\lesssim \|\!|F - \mathcal{Q} \circ F|\!\|_{L^2_{\mu}(\mathcal{X};\mathcal{Y})} + \sigma\|\!|F - \mathcal{Q} \circ F|\!\|_{\mathsf{disc},\mu} \\
&\quad + \pi_{\mathcal{Q}} \left( \|\!|F - q \circ \mathcal{E}_{\mathcal{X}}|\!\|_{L^2_{\mu}(\mathcal{X};\mathcal{Y})} + \sigma\|\!|F - q \circ \mathcal{E}_{\mathcal{X}}|\!\|_{\mathsf{disc},\mu} \right) \\
&\quad + \tau + \sigma\delta\sqrt{k}\|F\|_{L^\infty_{\mu}(\mathcal{X};\mathcal{Y})} + \frac{\sigma}{\sqrt{m}}\|\boldsymbol{E}\|_{2;\mathcal{Y}}
\end{aligned}
$$

*and*

$$
\begin{aligned}
\|F - \widehat{F}\|_{L^\infty_{\mu}(\mathcal{X};\mathcal{Y})} &\lesssim \|F - \mathcal{Q} \circ F\|_{L^\infty_{\mu}(\mathcal{X};\mathcal{Y})} + \sqrt{k}\sigma\|\!|F - \mathcal{Q} \circ F|\!\|_{\mathsf{disc},\mu} \\
&\quad + \pi_{\mathcal{Q}} \left( \|F - q \circ \mathcal{E}_{\mathcal{X}}\|_{L^\infty_{\mu}(\mathcal{X};\mathcal{Y})} + \sqrt{k}\sigma\|\!|F - q \circ \mathcal{E}_{\mathcal{X}}|\!\|_{\mathsf{disc},\mu} \right) \\
&\quad + \sqrt{k}\tau + \sigma\delta k\|F\|_{L^\infty_{\mu}(\mathcal{X};\mathcal{Y})} + \frac{\sqrt{k}\sigma}{\sqrt{m}}\|\boldsymbol{E}\|_{2;\mathcal{Y}}
\end{aligned}
$$

*for all $q \in \mathcal{P}_{\mathcal{S};\mathcal{Y}}$.*

*Proof.* Let $\hat{p}$ be as in Lemma D.12. Then

$$
\begin{aligned}
\|\!|F - \widehat{F}|\!\|_{L^2_{\mu}(\mathcal{X};\mathcal{Y})} &\leq \|\!|F - \hat{p} \circ \mathcal{E}_{\mathcal{X}}|\!\|_{L^2_{\mu}(\mathcal{X};\mathcal{Y})} + \|\!|\hat{p} \circ \mathcal{E}_{\mathcal{X}} - \mathcal{D}_{\mathcal{Y}} \circ \widehat{N} \circ \mathcal{E}_{\mathcal{X}}|\!\|_{L^2_{\mu}(\mathcal{X};\mathcal{Y})} \\
\|F - \widehat{F}\|_{L^\infty_{\mu}(\mathcal{X};\mathcal{Y})} &\leq \|F - \hat{p} \circ \mathcal{E}_{\mathcal{X}}\|_{L^\infty_{\mu}(\mathcal{X};\mathcal{Y})} + \|\hat{p} \circ \mathcal{E}_{\mathcal{X}} - \mathcal{D}_{\mathcal{Y}} \circ \widehat{N} \circ \mathcal{E}_{\mathcal{X}}\|_{L^\infty_{\mu}(\mathcal{X};\mathcal{Y})}
\end{aligned} \tag{D.23}
$$

For the second term, we have, by (A.III) and Lemma D.10,

$$
\begin{aligned}
\|\!|\hat{p} \circ \mathcal{E}_{\mathcal{X}} - \mathcal{D}_{\mathcal{Y}} \circ \widehat{N} \circ \mathcal{E}_{\mathcal{X}}|\!\|_{L^2_{\mu}(\mathcal{X};\mathcal{Y})} &\leq \|\hat{p} \circ \mathcal{E}_{\mathcal{X}} - \mathcal{D}_{\mathcal{Y}} \circ \widehat{N} \circ \mathcal{E}_{\mathcal{X}}\|_{L^\infty_{\mu}(\mathcal{X};\mathcal{Y})} \\
&= \|\hat{p} - \mathcal{D}_{\mathcal{Y}} \circ \widehat{N}\|_{L^\infty_{\tilde{\xi}}(D;\mathcal{Y})} \\
&\leq \|\hat{p} - \mathcal{D}_{\mathcal{Y}} \circ \widehat{N}\|_{L^\infty_{\varrho}(D;\mathcal{Y})} \\
&\leq \delta\sqrt{k}\|\!|\hat{p}|\!\|_{L^2_{\varrho}(D;\mathcal{Y})}.
\end{aligned}
$$

We now apply this, Lemma D.12, Theorem D.6 and the facts that $\alpha^{-1} \lesssim 1$ and $\sigma' \geq 1$ to obtain

$$
\begin{aligned}
\|\!|F - \widehat{F}|\!\|_{L^2_{\mu}(\mathcal{X};\mathcal{Y})} &\lesssim \|\!|F - \mathcal{Q} \circ F|\!\|_{L^2_{\mu}(\mathcal{X};\mathcal{Y})} + \sigma'\|\!|F - \mathcal{Q} \circ F|\!\|_{\mathsf{disc},\mu} \\
&\quad + \pi_{\mathcal{Q}} \left( \|\!|F - q \circ \mathcal{E}_{\mathcal{X}}|\!\|_{L^2_{\mu}(\mathcal{X};\mathcal{Y})} + \sigma'\|\!|F - q \circ \mathcal{E}_{\mathcal{X}}|\!\|_{\mathsf{disc},\mu} \right) \\
&\quad + \tau' + \frac{\sigma'}{\sqrt{m}}\|\boldsymbol{E}\|_{2;\mathcal{Y}}
\end{aligned}
$$

*and*

$$
\begin{aligned}
\|F - \widehat{F}\|_{L^\infty_{\mu}(\mathcal{X};\mathcal{Y})} &\lesssim \|F - \mathcal{Q} \circ F\|_{L^\infty_{\mu}(\mathcal{X};\mathcal{Y})} + \sqrt{k}\sigma'\|\!|F - \mathcal{Q} \circ F|\!\|_{\mathsf{disc},\mu} \\
&\quad + \pi_{\mathcal{Q}} \left( \|F - q \circ \mathcal{E}_{\mathcal{X}}\|_{L^\infty_{\mu}(\mathcal{X};\mathcal{Y})} + \sqrt{k}\sigma'\|\!|F - q \circ \mathcal{E}_{\mathcal{X}}|\!\|_{\mathsf{disc},\mu} \right) \\
&\quad + \sqrt{k}\tau' + \frac{\sqrt{k}\sigma'}{\sqrt{m}}\|\boldsymbol{E}\|_{2;\mathcal{Y}}
\end{aligned}
$$

for any $q \in \mathcal{P}_{\mathcal{S};\mathcal{Y}}$, where $\sigma'$ and $\tau'$ are as in (D.22). Due to the various assumptions, these satisfy

$$
\sigma' \lesssim \sigma, \quad \tau' \lesssim \tau + \sigma\delta\sqrt{k}\|F\|_{L^\infty_{\mu}(\mathcal{X};\mathcal{Y})} + \frac{\sigma}{\sqrt{m}}\|\boldsymbol{E}\|_{2;\mathcal{Y}} + \delta\sqrt{k}\|\!|\hat{p}|\!\|_{L^2_{\varrho}(D;\mathcal{Y})}.
$$

We deduce that

$$
\begin{aligned}
\|\!|F - \widehat{F}|\!\|_{L^2_{\mu}(\mathcal{X};\mathcal{Y})} &\lesssim \|\!|F - \mathcal{Q} \circ F|\!\|_{L^2_{\mu}(\mathcal{X};\mathcal{Y})} + \sigma\|\!|F - \mathcal{Q} \circ F|\!\|_{\mathsf{disc},\mu} \\
&\quad + \pi_{\mathcal{Q}} \left( \|\!|F - q \circ \mathcal{E}_{\mathcal{X}}|\!\|_{L^2_{\mu}(\mathcal{X};\mathcal{Y})} + \sigma\|\!|F - q \circ \mathcal{E}_{\mathcal{X}}|\!\|_{\mathsf{disc},\mu} \right) \\
&\quad + \tau + \sigma\delta\sqrt{k}\|F\|_{L^\infty_{\mu}(\mathcal{X};\mathcal{Y})} + \frac{\sigma}{\sqrt{m}}\|\boldsymbol{E}\|_{2;\mathcal{Y}} + \delta\sqrt{k}\|\!|\hat{p}|\!\|_{L^2_{\varrho}(D;\mathcal{Y})}
\end{aligned}
$$

and

$$\|F - \widehat{F}\|_{L^\infty_\mu(\mathcal{X};\mathcal{Y})} \lesssim \|F - \mathcal{Q} \circ F\|_{L^\infty_\mu(\mathcal{X};\mathcal{Y})} + \sqrt{k}\sigma \|\!|F - \mathcal{Q} \circ F\|\!|_{\mathsf{disc},\mu}$$

$$+ \pi_{\mathcal{Q}} \left( \|F - q \circ \mathcal{E}_{\mathcal{X}}\|_{L^\infty_\mu(\mathcal{X};\mathcal{Y})} + \sqrt{k}\sigma \|\!|F - q \circ \mathcal{E}_{\mathcal{X}}\|\!|_{\mathsf{disc},\mu} \right)$$

$$+ \sqrt{k}\tau + \sigma\delta k \|F\|_{L^\infty_\mu(\mathcal{X};\mathcal{Y})} + \frac{\sqrt{k}\sigma}{\sqrt{m}} \|\boldsymbol{E}\|_{2;\mathcal{Y}} + \delta k \|\!|\hat{p}\|\!|_{L^2_\varrho(D;\mathcal{Y})}.$$

We now bound the final term. Using Lemma D.10 once more, in combination with the fact that $\alpha - \delta\sqrt{k} \gtrsim 1$, we see that

$$\|\!|\hat{p}\|\!|_{L^2_\varrho(D;\mathcal{Y})} \le \frac{1}{\alpha - \delta\sqrt{k}} \|\!|\mathcal{D}_{\mathcal{Y}} \circ \widehat{N}\|\!|_{\mathsf{disc},\tilde{\varsigma}} \lesssim \|\!|\mathcal{D}_{\mathcal{Y}} \circ \widehat{N}\|\!|_{\mathsf{disc},\tilde{\varsigma}}.$$

$$\le \sqrt{\frac{1}{m}\sum_{i=1}^m \|Y_i - \mathcal{D}_{\mathcal{Y}} \circ \widehat{N} \circ \mathcal{E}_{\mathcal{X}}(X_i)\|_{\mathcal{Y}}^2} + \frac{1}{\sqrt{m}}\|\boldsymbol{Y}\|_{2;\mathcal{Y}}$$

$$\le \sigma \sqrt{\frac{1}{m}\sum_{i=1}^m \|Y_i - 0\|_{\mathcal{Y}}^2} + \tau + \frac{1}{\sqrt{m}}\|\boldsymbol{Y}\|_{2;\mathcal{Y}}$$

$$\le (1+\sigma)\left( \|F\|_{L^\infty_\mu(\mathcal{X};\mathcal{Y})} + \frac{1}{\sqrt{m}}\|\boldsymbol{E}\|_{2;\mathcal{Y}} \right) + \tau.$$

Here, we also used the fact that $\widehat{N}$ is an approximate minimizer in the fourth step, as well as the facts that the zero network $0 \in \mathcal{N}$ and that $\mathcal{D}_{\mathcal{Y}}$ is a linear map. Plugging this into the previous expressions now gives the result. $\qquad\square$

## D.7 Bounding the best polynomial approximation error terms

When $\sigma = 1$ (as will be the case when we come to prove Theorem 3.1), the error bounds in Theorem D.13 involve best polynomial approximation error terms of the form

$$\|F - q \circ \mathcal{E}_{\mathcal{X}}\|_{L^2_\mu(\mathcal{X};\mathcal{Y})} + \|\!|F - q \circ \mathcal{E}_{\mathcal{X}}\|\!|_{\mathsf{disc},\mu}, \quad \|F - q \circ \mathcal{E}_{\mathcal{X}}\|_{L^\infty_\mu(\mathcal{X};\mathcal{Y})} + \sqrt{k}\|\!|F - q \circ \mathcal{E}_{\mathcal{X}}\|\!|_{\mathsf{disc},\mu}$$

for arbitrary $q \in \mathcal{P}_{\mathcal{S};\mathcal{Y}}$. By the triangle inequality, these are bounded by

$$E_2(F, q) := \|\!|F - q \circ \iota\|\!|_{L^2_\mu(\mathcal{X};\mathcal{Y})} + \|\!|F - q \circ \iota\|\!|_{\mathsf{disc},\mu}$$
$$+ \|\!|q \circ \iota - q \circ \mathcal{E}_{\mathcal{X}}\|\!|_{L^2_\mu(\mathcal{X};\mathcal{Y})} + \|\!|q \circ \iota - q \circ \mathcal{E}_{\mathcal{X}}\|\!|_{\mathsf{disc},\mu},$$

$$E_\infty(F, q) := \|F - q \circ \iota\|_{L^\infty_\mu(\mathcal{X};\mathcal{Y})} + \sqrt{k}\|\!|F - q \circ \iota\|\!|_{\mathsf{disc},\mu} \tag{D.24}$$
$$+ \|q \circ \iota - q \circ \mathcal{E}_{\mathcal{X}}\|_{L^\infty_\mu(\mathcal{X};\mathcal{Y})} + \sqrt{k}\|\!|q \circ \iota - q \circ \mathcal{E}_{\mathcal{X}}\|\!|_{\mathsf{disc},\mu}.$$

In this section, we construct a suitable polynomial $q$ in the case where (A.II) holds and thereby derive a bound for these term. We first require the following lemma.

**Lemma D.14.** *Let $G \in L^\infty_\mu(\mathcal{X};\mathcal{Y})$ and $0 < \epsilon < 1$. Then the following hold.*

*(a) With probability at least $1 - \epsilon$ on the draw of the $X_i$, we have*

$$\|\!|G\|\!|_{\mathsf{disc},\mu} \le \|G\|_{L^2_\mu(\mathcal{X};\mathcal{Y})}/\sqrt{\epsilon}.$$

*(b) Suppose that $m \ge 2r\log(2/\epsilon)$ for some $r > 0$. Then, with probability at least $1 - \epsilon$ on the draw of the $X_i$, we have*

$$\|\!|G\|\!|_{\mathsf{disc},\mu} \le \sqrt{2}\left( \|G\|_{L^\infty_\mu(\mathcal{X};\mathcal{Y})}/\sqrt{r} + \|G\|_{L^2_\mu(\mathcal{X};\mathcal{Y})} \right).$$

*Proof.* Observe that the random variable $\|\!|G\|\!|_{\mathsf{disc},\mu}^2$ satisfies

$$\mathbb{E}\|\!|G\|\!|_{\mathsf{disc},\mu}^2 = \|G\|_{L^2_\mu(\mathcal{X};\mathcal{Y})}^2.$$

For (a), we use Markov's inequality to get

$$\mathbb{P}\left(\|\|G\|\|_{\mathsf{disc},\mu} \geq \|G\|_{L^2_\mu(\mathcal{X};\mathcal{Y})}/\sqrt{\epsilon}\right) \leq \frac{\mathbb{E}\|\|G\|\|^2_{\mathsf{disc},\mu}}{\|G\|^2_{L^2_\mu(\mathcal{X};\mathcal{Y})}/\epsilon} = \epsilon,$$

as required.

The proof of (b) is based on [3, Lem. 7.11]. We repeat it here for convenience. Define the random variable $Z_i = \|G(X_i)\|^2_{\mathcal{Y}}$ and observe that

$$\mathbb{E}(Z_i) = \mathbb{E}_{X \sim \mu}\|G(X)\|^2_{\mathcal{Y}} = \|G\|^2_{L^2_\mu(\mathcal{X};\mathcal{Y})} =: a.$$

Let $X_i = Z_i - \mathbb{E}(Z_i)$ so that

$$\|\|G\|\|^2_{\mathsf{disc},\mu} = \frac{1}{m}\sum_{i=1}^m Z_i = \frac{1}{m}\sum_{i=1}^m X_i + a.$$

Now let $b = \|G\|^2_{L^\infty_\mu(\mathcal{X};\mathcal{Y})}$ and observe that

$$X_i \leq Z_i \leq b, \quad -X_i \leq \mathbb{E}(Z_i) \leq b, \quad \text{a.s..}$$

We also have

$$\sum_{i=1}^m \mathbb{E}(X_i^2) \leq \sum_{i=1}^m \mathbb{E}(Z_i^2) \leq b\sum_{i=1}^m \mathbb{E}(Z_i) = abm.$$

Therefore, Bernstein's inequality for bounded random variables (see, e.g., [29, Cor. 7.31]) implies that

$$\mathbb{P}\left(\left|\frac{1}{m}\sum_{i=1}^m X_i\right| \geq t\right) \leq 2\exp\left(-\frac{t^2 m/2}{ab + bt/3}\right)$$

for any $t > 0$. We now set $t = a + b/r$ and notice that $\frac{t^2 m/2}{ab+bt/3} \geq \frac{3m}{5r} \geq \log(2/\epsilon)$. Therefore,

$$\left|\frac{1}{m}\sum_{i=1}^m X_i\right| < a + b/r$$

with probability at least $1 - \epsilon$. It follows that

$$\|\|G\|\|_{\mathsf{disc},\mu} \leq \sqrt{2a + b/r} \leq \sqrt{2}\left(\sqrt{a} + \sqrt{b/r}\right)$$

with the same probability. Substituting the values for $a$ and $b$ now gives the result. $\qquad\square$

**Lemma D.15.** *Let* $F \in L^\infty_\mu(\mathcal{X};\mathcal{Y})$, $q \in \mathcal{P}_{S;\mathcal{Y}}$ *be arbitrary and* $m \geq 2r\log(6/\epsilon)$ *for some* $r > 0$ *and* $0 < \epsilon < 1$. *Then*

$$E_2(F,q) \lesssim \|F - q \circ \iota\|_{L^\infty_\mu(\mathcal{X};\mathcal{Y})}/\sqrt{r} + \|F - q \circ \iota\|_{L^2_\mu(\mathcal{X};\mathcal{Y})}$$
$$+ \sqrt{k/\epsilon}\|\|q\|\|_{L^2_\varrho(D;\mathcal{Y})}\|\iota_{d_\mathcal{X}} - \iota_{d_\mathcal{X}} \circ \widetilde{\mathcal{D}}_\mathcal{X} \circ \widetilde{\mathcal{E}}_\mathcal{X}\|_{L^2_\mu(\mathcal{X};\ell^\infty(\mathbb{R}^{d_\mathcal{X}}))}.$$

*and*

$$E_\infty(F,q) \lesssim (1 + \sqrt{k/r})\|F - q \circ \iota\|_{L^\infty_\mu(\mathcal{X};\mathcal{Y})} + \sqrt{k}\|F - q \circ \iota\|_{L^2_\mu(\mathcal{X};\mathcal{Y})}$$
$$+ k\|\|q\|\|_{L^2_\varrho(D;\mathcal{Y})}\|\iota_{d_\mathcal{X}} - \iota_{d_\mathcal{X}} \circ \widetilde{\mathcal{D}}_\mathcal{X} \circ \widetilde{\mathcal{E}}_\mathcal{X}\|_{L^2_\mu(\mathcal{X};\ell^\infty(\mathbb{R}^{d_\mathcal{X}}))}$$
$$+ \sqrt{k}(1 + \sqrt{k/r})\|\|q\|\|_{L^2_\varrho(D;\mathcal{Y})}\|\iota_{d_\mathcal{X}} - \iota_{d_\mathcal{X}} \circ \widetilde{\mathcal{D}}_\mathcal{X} \circ \widetilde{\mathcal{E}}_\mathcal{X}\|_{L^\infty_\mu(\mathcal{X};\ell^\infty(\mathbb{R}^{d_\mathcal{X}}))}$$

*with probability at least* $1 - \epsilon$.

*Proof.* We apply part (b) of Lemma D.14 with $\epsilon$ replaced by $\epsilon/3$ to the term $\||F - q \circ \iota\||_{\mathsf{disc},\mu}$ and parts (a) and (b) of Lemma D.14 with $\epsilon$ replaced by $\epsilon/3$ to the term $\||q \circ \iota - q \circ \mathcal{E}_{\mathcal{X}}\||_{\mathsf{disc},\mu}$. Using the union bound, this gives that

$$\||F - q \circ \iota\||_{\mathsf{disc},\mu} \lesssim \|F - q \circ \iota\|_{L^2_\mu(\mathcal{X};\mathcal{Y})} + \|F - q \circ \iota\|_{L^\infty_\mu(\mathcal{X};\mathcal{Y})}/\sqrt{r}$$

$$\||q \circ \iota - q \circ \mathcal{E}_{\mathcal{X}}\||_{\mathsf{disc},\mu} \lesssim \|q \circ \iota - q \circ \mathcal{E}_{\mathcal{X}}\|_{L^2_\mu(\mathcal{X};\mathcal{Y})}/\sqrt{\epsilon}$$

$$\||q \circ \iota - q \circ \mathcal{E}_{\mathcal{X}}\||_{\mathsf{disc},\mu} \lesssim \|q \circ \iota - q \circ \mathcal{E}_{\mathcal{X}}\|_{L^\infty_\mu(\mathcal{X};\mathcal{Y})}//\sqrt{r} + \|q \circ \iota - q \circ \mathcal{E}_{\mathcal{X}}\|_{L^2_\mu(\mathcal{X};\mathcal{Y})}$$

with probability at least $1 - \epsilon$. This yields

$$E_2(F,q) \lesssim \|F - q \circ \iota\|_{L^2_\mu(\mathcal{X};\mathcal{Y})} + \|F - q \circ \iota\|_{L^\infty_\mu(\mathcal{X};\mathcal{Y})}/\sqrt{r}$$
$$+ \|q \circ \iota - q \circ \mathcal{E}_{\mathcal{X}}\|_{L^2_\mu(\mathcal{X};\mathcal{Y})}/\sqrt{\epsilon}$$

$$E_\infty(F,q) \lesssim (1 + \sqrt{k/r})\|F - q \circ \iota\|_{L^\infty_\mu(\mathcal{X};\mathcal{Y})} + \sqrt{k}\|F - q \circ \iota\|_{L^2_\mu(\mathcal{X};\mathcal{Y})}$$
$$+ (1 + \sqrt{k/r})\|q \circ \iota - q \circ \mathcal{E}_{\mathcal{X}}\|_{L^\infty_\mu(\mathcal{X};\mathcal{Y})} + \sqrt{k}\|q \circ \iota - q \circ \mathcal{E}_{\mathcal{X}}\|_{L^2_\mu(\mathcal{X};\mathcal{Y})}$$

with the same probability. It remains to bound the terms involving $q \circ \iota - q \circ \mathcal{E}_{\mathcal{X}}$. Using (A.III), Lemma D.2 and the fact that $q$ depends on its first $d_{\mathcal{X}}$ variables only, we see that

$$\|q \circ \iota(X) - q \circ \mathcal{E}_{\mathcal{X}}(X)\|_{\mathcal{Y}} \leq \frac{1}{2}\sqrt{k}\||q\||_{L^2_\varrho(D;\mathcal{Y})}\|\iota_{d_{\mathcal{X}}}(X) - \mathcal{E}_{\mathcal{X}}(X)\|_\infty.$$

Therefore

$$\|q \circ \iota - q \circ \mathcal{E}_{\mathcal{X}}\|_{L^2_\mu(\mathcal{X};\mathcal{Y})} \lesssim \sqrt{k}\||q\||_{L^2_\varrho(D;\mathcal{Y})}\|\iota_{d_{\mathcal{X}}} - \mathcal{E}_{\mathcal{X}}\|_{L^2_\mu(\mathcal{X};\ell^\infty(\mathbb{N}))}$$

$$\|q \circ \iota - q \circ \mathcal{E}_{\mathcal{X}}\|_{L^\infty_\mu(\mathcal{X};\mathcal{Y})} \lesssim \sqrt{k}\||q\||_{L^2_\varrho(D;\mathcal{Y})}\|\iota_{d_{\mathcal{X}}} - \mathcal{E}_{\mathcal{X}}\|_{L^\infty_\mu(\mathcal{X};\ell^\infty(\mathbb{N}))}$$

Substituting this into the previous bounds and using the definition of $\mathcal{E}_{\mathcal{X}}$ from (A.III) now gives the result. $\qquad\square$

We are now ready to choose the polynomial $q$. First, we require the following technical lemma, which shows the existence of multi-index sets of weighted cardinality $k$ which achieve the desired algebraic rates of convergence.

**Lemma D.16.** *Let $f = \sum_{\nu \in \mathcal{F}} c_\nu \Psi_\nu$ satisfy $f \in \mathcal{H}(\mathbf{b})$ for some $\mathbf{b} \in \ell^p(\mathbb{N})$ with $\mathbf{b} \geq \mathbf{0}$. Then there are index sets $S_1, S_2 \subset \mathcal{F}$ with $|S_1|_{\mathbf{v}}, |S_2|_{\mathbf{v}} \leq k$ such that*

$$\|\mathbf{c} - \mathbf{c}_{S_1}\|_{2;\mathcal{Y}} \leq C(\mathbf{b},p,\xi) \cdot k^{1/2-1/p}, \quad \|\mathbf{c} - \mathbf{c}_{S_2}\|_{1,\mathbf{v};\mathcal{Y}} \leq C(\mathbf{b},p,\xi) \cdot k^{1-1/p},$$

*where $C(\mathbf{b},p,\xi) \geq 0$ depends on $\mathbf{b}$, $p$ and $\xi$ only.*

*Proof.* We first show that $\mathbf{c} \in \ell^p_{\mathbf{v}}(\mathcal{F};\mathcal{Y})$. By definition of $\mathcal{H}(\mathbf{b})$ (see (2.3)), $f$ is holomorphic in every Bernstein polyellipse $\mathcal{E}(\boldsymbol{\rho})$ for which $\boldsymbol{\rho}$ satisfies

$$\boldsymbol{\rho} \geq \mathbf{1}, \quad \sum_{j=1}^\infty \left(\frac{\rho_j + \rho_j^{-1}}{2} - 1\right) b_j \leq 1. \tag{D.25}$$

Using [4, Lem. 5.3] (which is based on [102, Cor. B.2.7]) with $\boldsymbol{\alpha} = \boldsymbol{\beta} = \mathbf{0}$, we get that $\|c_\mathbf{0}\|_{\mathcal{Y}} \leq 1$ and

$$\|c_\nu\|_{\mathcal{Y}} \leq \prod_{k \in I(\nu,\boldsymbol{\rho})} \frac{\rho_k^{-\nu_k+1}}{(\rho_k - 1)^2}(\nu_k + 1), \quad \nu \in \mathcal{F}\backslash\{\mathbf{0}\}$$

for all such $\boldsymbol{\rho}$, where $I(\nu,\boldsymbol{\rho}) = \mathrm{supp}(\nu) \cap \{k : \rho_k > 1\}$. Define the sequence $d_\mathbf{0} = 1$ and

$$d_\nu = v_\nu^{2/p-1} \cdot \inf\left\{\prod_{k \in I(\nu,\boldsymbol{\rho})} \frac{\rho_k^{-\nu_k+1}}{(\rho_k - 1)^2}(\nu_k + 1) : \boldsymbol{\rho} \text{ satisfies (D.25)}\right\}, \quad \nu \in \mathcal{F}\backslash\{\mathbf{0}\}.$$

Using (C.3)-(C.4), we see that

$$v_{\boldsymbol{\nu}} = \prod_{k \in \mathrm{supp}(\boldsymbol{\nu})} (2\nu_k + 1)^{(5+\xi)/2}$$

and therefore

$$d_{\boldsymbol{\nu}} \leq \prod_{k \in I(\boldsymbol{\nu},\boldsymbol{\rho})} \frac{\rho_k^{-\nu_k+1}}{(\rho_k - 1)^2} (2\nu_k + 1)^{\gamma}$$

for all $\boldsymbol{\rho}$ satisfying (D.25) and some $\gamma = \gamma(p, \xi) \geq 0$. We now use [4, Lem. 5.4] to deduce that $\boldsymbol{d} = (d_{\boldsymbol{\nu}})_{\boldsymbol{\nu} \in \mathcal{F}} \in \ell^p(\mathbb{N})$ with $\|\boldsymbol{d}\|_p \leq C(\boldsymbol{b}, p, \xi)$ for some $C(\boldsymbol{b}, p, \xi) \geq 0$ depending on $\boldsymbol{b}$, $p$ and $\xi$ only. Returning to $\boldsymbol{c}$, this gives

$$\|\boldsymbol{c}\|_{p,\boldsymbol{v};\mathcal{Y}}^p = \sum_{\boldsymbol{\nu} \in \mathcal{F}} v_{\boldsymbol{\nu}}^{2-p} \|c_{\boldsymbol{\nu}}\|_{\mathcal{Y}}^p \leq \sum_{\boldsymbol{\nu} \in \mathcal{F}} |d_{\boldsymbol{\nu}}|^p \leq C(\boldsymbol{b}, p, \xi)^p.$$

Hence $\boldsymbol{c} \in \ell_{\boldsymbol{v}}^p(\mathcal{F}; \mathcal{Y})$ with $\|\boldsymbol{c}\|_{p,\boldsymbol{v};\mathcal{Y}} \leq C(\boldsymbol{b}, p, \xi)$, as required.

The second step involves the application of the weighted Stechkin's inequality (see [3, Lem. 3.12]). This gives that

$$\min \left\{ \|\boldsymbol{c} - \boldsymbol{c}_S\|_{q,\boldsymbol{v};\mathcal{Y}} : S \subset \mathcal{F}, \ |S|_{\boldsymbol{v}} \leq k \right\} =: \sigma_k(\boldsymbol{c})_{q,\boldsymbol{v};\mathcal{Y}} \leq \|\boldsymbol{c}\|_{p,\boldsymbol{v};\mathcal{Y}} k^{1/q-1/p},$$

for any $q \in (p, 2]$ and $k > 0$. Applying this result with $q = 2$ implies the existence of the set $S_1$ (recall that $\| \cdot \|_{2,\boldsymbol{v};\mathcal{Y}} = \| \cdot \|_{2;\mathcal{Y}}$) and applying it with $q = 1$ implies the existence of the set $S_2$. $\square$

We now define the set

$$\Lambda_n^{\mathsf{HCI}} = \left\{ \boldsymbol{\nu} = (\nu_k)_{k=1}^{\infty} \in \mathcal{F} : \prod_{k:\nu_k \neq 0} (\nu_k + 1) \leq n, \ \nu_k = 0, \ k > n \right\} \subset \mathcal{F}. \tag{D.26}$$

Notice that $\Lambda_n^{\mathsf{HCI}}$ is isomorphic to an index set in $\mathbb{N}_0^n$ by the natural restriction map.

**Lemma D.17.** *Let $k > 0$ and suppose that $\Lambda \supseteq \Lambda_n^{\mathsf{HCI}}$ for some $n \in \mathbb{N}$. Let $f = \sum_{\boldsymbol{\nu} \in \mathcal{F}} c_{\boldsymbol{\nu}} \Psi_{\boldsymbol{\nu}}$ satisfy $f \in \mathcal{H}(\boldsymbol{b})$ for some $\boldsymbol{b} \in \ell_{\mathsf{M}}^p(\mathbb{N})$ with $\boldsymbol{b} \geq \boldsymbol{0}$. Then there exists an index set $S \in \mathcal{S}$ such that*

$$\|f - f_S\|_{L_{\varrho}^{\infty}(D;\mathcal{Y})} \leq C(\boldsymbol{b}, p, \xi) \cdot \left( k^{1-1/p} + n^{1-1/p} \right), \tag{D.27}$$

*where $f_S = \sum_{\boldsymbol{\nu} \in S} c_{\boldsymbol{\nu}} \Psi_{\boldsymbol{\nu}}$. Moreover, if $\mathcal{Y}$ is a Hilbert space, then we also have that*

$$\|f - f_S\|_{L_{\varrho}^2(D;\mathcal{Y})} \leq C(\boldsymbol{b}, p, \xi) \cdot \left( k^{1/2-1/p} + n^{1/2-1/p} \right). \tag{D.28}$$

*Here $C(\boldsymbol{b}, p, \xi) > 0$ is a constant depending on $\boldsymbol{b}$, $p$ and $\xi$ only.*

*Proof of Lemma D.17.* The previous lemma implies that there exist index sets $S_1, S_2 \subset \mathcal{F}$ with $|S_1|_{\boldsymbol{v}}, |S_2|_{\boldsymbol{v}} \leq k/2$ such that

$$\|\boldsymbol{c} - \boldsymbol{c}_{S_1}\|_{2;\mathcal{Y}} \leq C(\boldsymbol{b}, p, \xi) \cdot k^{1/2-1/p}, \quad \|\boldsymbol{c} - \boldsymbol{c}_{S_2}\|_{1,\boldsymbol{v};\mathcal{Y}} \leq C(\boldsymbol{b}, p, \xi) \cdot k^{1-1/p}.$$

Now define $S = S_1 \cup S_2 \cap \Lambda$ and notice that $S \subseteq \Lambda$ and $|S|_{\boldsymbol{v}} \leq |S_1|_{\boldsymbol{v}} + |S_2|_{\boldsymbol{v}} \leq k$. Hence $S \in \mathcal{S}$. Since $v_{\boldsymbol{\nu}} \geq u_{\boldsymbol{\nu}} = \|\Psi_{\boldsymbol{\nu}}\|_{L_{\varrho}^{\infty}(D)}$, we have (using [5, Lem. 5.1])

$$\|f - f_S\|_{L_{\varrho}^{\infty}(D;\mathcal{Y})} \leq \|\boldsymbol{c} - \boldsymbol{c}_S\|_{1,\boldsymbol{u};\mathcal{Y}} \leq \|\boldsymbol{c} - \boldsymbol{c}_{S_2}\|_{1,\boldsymbol{v};\mathcal{Y}} + \|\boldsymbol{c} - \boldsymbol{c}_{\Lambda}\|_{1,\boldsymbol{u};\mathcal{Y}}.$$

Hence, to complete the proof of the first result, we need only show that

$$\|\boldsymbol{c} - \boldsymbol{c}_{\Lambda}\|_{1,\boldsymbol{u};\mathcal{Y}} \leq C(\boldsymbol{b}, p) \cdot n^{1-1/p}$$

for some constant $C(\boldsymbol{b}, p) \geq 0$. First, by construction, $\Lambda \supseteq \Lambda_n^{\mathsf{HCI}}$ contains every anchored set of size at most $n$. See, e.g.. Therefore $\|\boldsymbol{c} - \boldsymbol{c}_{\Lambda}\|_{1,\boldsymbol{u};\mathcal{Y}} \leq \|\boldsymbol{c} - \boldsymbol{c}_S\|_{1,\boldsymbol{u};\mathcal{Y}}$ for any such set $S$. The result now follows from [5, Cor. 8.2].

Now suppose that $\mathcal{Y}$ is a Hilbert space. Then Parseval's identity gives that

$$\|f - f_S\|_{L_{\varrho}^2(D;\mathcal{Y})} = \|\boldsymbol{c} - \boldsymbol{c}_S\|_{2;\mathcal{Y}} \leq \|\boldsymbol{c} - \boldsymbol{c}_{S_1}\|_{2;\mathcal{Y}} + \|\boldsymbol{c} - \boldsymbol{c}_{\Lambda}\|_{2;\mathcal{Y}}.$$

As before, it suffices to show that

$$\|\boldsymbol{c} - \boldsymbol{c}_{\Lambda}\|_{2;\mathcal{Y}} \leq C(\boldsymbol{b}, p) \cdot n^{1/2-1/p}.$$

This follows from the same approach and [5, Cor. 8.2] once more. $\square$

## D.8 Final arguments

We are now, finally, ready to prove Theorem 3.1. We first consider the case where $\mathcal{Y}$ is a Banach space, then treat the case where $\mathcal{Y}$ is a Hilbert space afterwards.

*Proof of Theorem 3.1 when $\mathcal{Y}$ is a Banach space.* We divide the proof into a series of steps.

*Step 1: Setup and DNN width/depth bounds.* Let $m$, $\delta$, $\epsilon$ and $L$ be as in the theorem statement. We may without loss of generality assume that $\delta \leq 1/5$. Now let

$$n = \left\lceil \frac{m}{L} \right\rceil \leq d_{\mathcal{X}} \tag{D.29}$$

and

$$\Lambda = \Lambda_n^{\mathsf{HCl}},$$

where $\Lambda_n^{\mathsf{HCl}}$ is as in (D.26). We also set

$$\xi = 1/\delta - 5 \geq 0 \tag{D.30}$$

and

$$k = \frac{m}{cL}, \tag{D.31}$$

where $c \geq 1$ is a universal constant that will be chosen in the next step. Finally, let $\delta = 2^{-m}/m$ and $\mathcal{N}$ by the tanh DNN family (D.19), where the $N_\nu$ are as in Lemma D.9 for this $\Lambda$ and value of $\delta$.

By (D.20) and the definition of $\xi$ and $k$, we immediately see that

$$\mathrm{width}(\mathcal{N}) \lesssim (m/L)^{1+\delta}, \qquad \mathrm{depth}(\mathcal{N}) \lesssim \log(m/L).$$

This yields the width and depth bounds (3.1). The rest of the proof is therefore devoted to showing the error bounds (3.3)-(3.4).

*Step 2: Ensuring* (D.7) *holds with probability at least* $1 - \epsilon/2$. A standard bound (see, e.g., the proof of Lemma 6.4 in [5]) gives that $N = |\Lambda|$ satisfies

$$\log(eN) \leq 4\log^2(en) \lesssim \log^2(m). \tag{D.32}$$

Here, in the final step we used the fact that $m \geq 3$ and $L(m, \epsilon) \geq 1$. This and the fact that $c \geq 1$ also implies that $k \leq m$. Therefore, the right-hand side of (D.13) with $\epsilon$ replaced by $\epsilon/2$ satisfies

$$c_0 \cdot k \cdot (\log(eN) \cdot \log^2(k) + \log(4/\epsilon)) \lesssim k \cdot L(m, \epsilon). \tag{D.33}$$

Hence, for sufficiently large $c \geq 1$, we deduce that (D.13) holds with $\epsilon/2$. Using (A.I), we see that

$$\|\iota_{d_{\mathcal{X}}} - \iota_{d_{\mathcal{X}}} \circ \widetilde{\mathcal{D}}_{\mathcal{X}} \circ \widetilde{\mathcal{E}}_{\mathcal{X}}\|_{L_\mu^q(\mathcal{X};\ell^\infty(\mathbb{N}))} \leq L_\iota \|\mathcal{I}_{\mathcal{X}} - \widetilde{\mathcal{D}}_{\mathcal{X}} \circ \widetilde{\mathcal{E}}_{\mathcal{X}}\|_{L_\mu^q(\mathcal{X};\mathcal{X})}, \quad q = 2, \infty. \tag{D.34}$$

Hence (D.14) is implied by (3.2). We conclude from Lemma D.8 that (D.7) holds with probability at least $1 - \epsilon/2$ and $\alpha \gtrsim 1$.

*Step 3: Error analysis.* Let $f \in \mathcal{H}(\boldsymbol{b})$ be the function asserted by (A.II) and define $q = f_S$ as the polynomial asserted by Lemma D.17 with $n$ as in (D.29). We also observe that

$$\|F\|_{L_\mu^\infty(\mathcal{X};\mathcal{Y})} = \|f\|_{L_\xi^\infty(D;\mathcal{Y})} \lesssim \|f\|_{L_\varrho^\infty(D;\mathcal{Y})} \lesssim 1, \tag{D.35}$$

since $f \in \mathcal{H}(\boldsymbol{b})$. We now apply Theorem D.13 with $\sigma = 1$ and use (D.31), the definition of $\delta$ and (3.8) to see that

$$\begin{aligned}
\||F - \widehat{F}\||_{L_\mu^2(\mathcal{X};\mathcal{Y})} &\lesssim \||F - \mathcal{Q} \circ F\||_{L_\mu^2(\mathcal{X};\mathcal{Y})} + \||F - \mathcal{Q} \circ F\||_{\mathsf{disc},\mu} \\
&\quad + \pi_{\mathcal{Q}} E_2(F, q) + E_{\mathsf{opt},2} + E_{\mathsf{samp},2} \\
\||F - \widehat{F}\||_{L_\mu^\infty(\mathcal{X};\mathcal{Y})} &\lesssim \||F - \mathcal{Q} \circ F\||_{L_\mu^\infty(\mathcal{X};\mathcal{Y})} + \sqrt{k}\||F - \mathcal{Q} \circ F\||_{\mathsf{disc},\mu} \\
&\quad + \pi_{\mathcal{Q}} E_\infty(F, q) + E_{\mathsf{opt},\infty} + E_{\mathsf{samp},\infty}
\end{aligned} \tag{D.36}$$

with probability at least $1 - \epsilon/2$, where $E_2(F, q)$ and $E_\infty(F, q)$ are as in (D.24), $\mathcal{Q} : \mathcal{Y} \to \widetilde{\mathcal{Y}} = \mathcal{D}_{\mathcal{Y}}(\mathbb{R}^{d_{\mathcal{Y}}})$ is any bounded linear operator and $\pi_{\mathcal{Q}} = \|\mathcal{Q}\|_{\mathcal{Y} \to \mathcal{Y}}$.

Notice that Parseval's identity and (D.35) imply that
$$\|q\|_{L^2_\varrho(D;\mathcal{Y})} \le \|f\|_{L^2_\varrho(D;\mathcal{Y})} \le \|f\|_{L^\infty_\varrho(D;\mathcal{Y})} \lesssim 1.$$

Hence, this, the previous bounds, Lemma D.15 with $\epsilon$ replaced by $\epsilon/4$ and the union bound yield

$$
\begin{aligned}
\|F - \widehat{F}\|_{L^2_\mu(\mathcal{X};\mathcal{Y})} \lesssim\; & \|F - \mathcal{Q} \circ F\|_{L^2_\mu(\mathcal{X};\mathcal{Y})} + \|F - \mathcal{Q} \circ F\|_{\mathsf{disc},\mu} \\
& + \pi_\mathcal{Q} \left( \|F - q \circ \iota\|_{L^\infty_\mu(\mathcal{X};\mathcal{Y})}/\sqrt{r} + \|F - q \circ \iota\|_{L^2_\mu(\mathcal{X};\mathcal{Y})} \right) \\
& + \pi_\mathcal{Q} \sqrt{k/\epsilon} \|\iota_{d_\mathcal{X}} - \iota_{d_\mathcal{X}} \circ \widetilde{\mathcal{D}}_\mathcal{X} \circ \widetilde{\mathcal{E}}_\mathcal{X}\|_{L^2_\mu(\mathcal{X};\ell^\infty(\mathbb{R}^{d_\mathcal{X}}))} \\
& + E_{\mathsf{opt},2} + E_{\mathsf{samp},2}
\end{aligned}
$$

and

$$
\begin{aligned}
\|F - \widehat{F}\|_{L^\infty_\mu(\mathcal{X};\mathcal{Y})} \lesssim\; & \|F - \mathcal{Q} \circ F\|_{L^\infty_\mu(\mathcal{X};\mathcal{Y})} + \sqrt{k}\|F - \mathcal{Q} \circ F\|_{\mathsf{disc},\mu} \\
& + \pi_\mathcal{Q} \left( (1 + \sqrt{k/r})\|F - q \circ \iota\|_{L^\infty_\mu(\mathcal{X};\mathcal{Y})} + \sqrt{k}\|F - q \circ \iota\|_{L^2_\mu(\mathcal{X};\mathcal{Y})} \right) \\
& + \pi_\mathcal{Q} k \|\iota_{d_\mathcal{X}} - \iota_{d_\mathcal{X}} \circ \widetilde{\mathcal{D}}_\mathcal{X} \circ \widetilde{\mathcal{E}}_\mathcal{X}\|_{L^2_\mu(\mathcal{X};\ell^\infty(\mathbb{R}^{d_\mathcal{X}}))} \\
& + \pi_\mathcal{Q} \sqrt{k}(1 + \sqrt{k/r})\|\iota_{d_\mathcal{X}} - \iota_{d_\mathcal{X}} \circ \widetilde{\mathcal{D}}_\mathcal{X} \circ \widetilde{\mathcal{E}}_\mathcal{X}\|_{L^\infty_\mu(\mathcal{X};\ell^\infty(\mathbb{R}^{d_\mathcal{X}}))} \\
& + E_{\mathsf{opt},\infty} + E_{\mathsf{samp},\infty}
\end{aligned}
$$

with probability at least $1 - 3\epsilon/4$, for any $r$ such that $m \ge 2r\log(24/\epsilon)$. In particular, we may choose $r = k$ due the definition of $k$ (D.31). We next bound the discrete error $\|F - \mathcal{Q} \circ F\|_{\mathsf{disc},\mu}$. Applying Lemma D.14 with $\epsilon$ replaced by $\epsilon/8$ and the union bound, we see that

$$
\begin{aligned}
\|F - \mathcal{Q} \circ F\|_{\mathsf{disc},\mu} &\sim \|F - \mathcal{Q} \circ F\|_{L^2_\mu(\mathcal{X};\mathcal{Y})}/\sqrt{\epsilon} \\
\|F - \mathcal{Q} \circ F\|_{\mathsf{disc},\mu} &\lesssim \|F - \mathcal{Q} \circ F\|_{L^\infty_\mu(\mathcal{X};\mathcal{Y})}/\sqrt{r} + \|F - \mathcal{Q} \circ F\|_{L^2_\mu(\mathcal{X};\mathcal{Y})}
\end{aligned}
$$

with probability at least $1 - \epsilon/4$, provided $m \ge 2r\log(16/\epsilon)$. In particular, we may take $r = k$ once more. Substituting this into the previous expressions, setting $r = k$ throughout, using the union bound once more and recalling (3.7), (D.31) and (D.34), we deduce that

$$
\begin{aligned}
\|F - \widehat{F}\|_{L^2_\mu(\mathcal{X};\mathcal{Y})} \lesssim\; & \|F - \mathcal{Q} \circ F\|_{L^2_\mu(\mathcal{X};\mathcal{Y})}/\sqrt{\epsilon} \\
& + \pi_\mathcal{Q} \left( \|F - q \circ \iota\|_{L^\infty_\mu(\mathcal{X};\mathcal{Y})}/\sqrt{m/L} + \|F - q \circ \iota\|_{L^2_\mu(\mathcal{X};\mathcal{Y})} \right) \\
& + (\pi_\mathcal{Q}/a_\mathcal{Y}) \cdot E_{\mathcal{X},2} + E_{\mathsf{opt},2} + E_{\mathsf{samp},2}
\end{aligned}
$$

and

$$
\begin{aligned}
\|F - \widehat{F}\|_{L^\infty_\mu(\mathcal{X};\mathcal{Y})} \lesssim\; & \|F - \mathcal{Q} \circ F\|_{L^\infty_\mu(\mathcal{X};\mathcal{Y})} + \sqrt{m/L}\|F - \mathcal{Q} \circ F\|_{L^2_\mu(\mathcal{X};\mathcal{Y})} \\
& + \pi_\mathcal{Q} \left( \|F - q \circ \iota\|_{L^\infty_\mu(\mathcal{X};\mathcal{Y})} + \sqrt{m/L}\|F - q \circ \iota\|_{L^2_\mu(\mathcal{X};\mathcal{Y})} \right) \\
& + (\pi_\mathcal{Q}/a_\mathcal{Y}) \cdot E_{\mathcal{X},\infty} + E_{\mathsf{opt},\infty} + E_{\mathsf{samp},\infty}
\end{aligned}
$$

with probability at least $1 - \epsilon$. This holds for any bounded linear operator $\mathcal{Q} : \mathcal{Y} \to \widetilde{\mathcal{Y}} = D_\mathcal{Y}(\mathbb{R}^{d_\mathcal{Y}})$. We now set $\mathcal{Q} = \mathcal{D}_\mathcal{Y} \circ \mathcal{E}_\mathcal{Y}$, which is linear and bounded by (A.IV) with $\pi_\mathcal{Q} = \|\mathcal{D}_\mathcal{Y} \circ \mathcal{E}_\mathcal{Y}\|_{\mathcal{Y} \to \mathcal{Y}} = a_\mathcal{Y}$ by definition. Using this, we observe that

$$\|F - \mathcal{Q} \circ F\|_{L^q_\mu(\mathcal{X};\mathcal{Y})} = \|\mathcal{I}_\mathcal{Y} - \mathcal{D}_\mathcal{Y} \circ \mathcal{E}_\mathcal{Y}\|_{L^q_{F_\sharp \mu}(\mathcal{Y};\mathcal{Y})}, \quad q = 2, \infty.$$

Substituting this into the previous expressions and recalling (3.7) now gives

$$
\begin{aligned}
\|F - \widehat{F}\|_{L^2_\mu(\mathcal{X};\mathcal{Y})} \lesssim\; & a_\mathcal{Y} \left( \|F - q \circ \iota\|_{L^\infty_\mu(\mathcal{X};\mathcal{Y})}/\sqrt{m/L} + \|F - q \circ \iota\|_{L^2_\mu(\mathcal{X};\mathcal{Y})} \right) \\
& + E_{\mathcal{X},2} + E_{\mathcal{Y},2} + E_{\mathsf{opt},2} + E_{\mathsf{samp},2} \\
\|F - \widehat{F}\|_{L^\infty_\mu(\mathcal{X};\mathcal{Y})} \lesssim\; & a_\mathcal{Y} \left( \|F - q \circ \iota\|_{L^\infty_\mu(\mathcal{X};\mathcal{Y})} + \sqrt{m/L}\|F - q \circ \iota\|_{L^2_\mu(\mathcal{X};\mathcal{Y})} \right) \\
& + E_{\mathcal{X},\infty} + E_{\mathcal{Y},\infty} + E_{\mathsf{opt},\infty} + E_{\mathsf{samp},\infty}.
\end{aligned}
$$
(D.37)

*Step 4: Bounding the polynomial error terms.* It remains to bound the error terms $F - q \circ \iota$ in (D.37). Using (A.I), (A.II) and Lemma D.17, we now notice that

$$\|F - q \circ \iota\|_{L^\infty_\mu(\mathcal{X};\mathcal{Y})} = \|f - q\|_{L^\infty_\varsigma(D;\mathcal{Y})} \lesssim \|f - q\|_{L^\infty_\varrho(D;\mathcal{Y})} \lesssim C(\boldsymbol{b}, p, \xi) \cdot (k^{1-1/p} + n^{1-1/p}).$$

Recall that $n \geq k$. Therefore, using this, (D.31) and (3.6), we see that

$$\begin{aligned}
a_{\mathcal{Y}} \left( \|F - q \circ \iota\|_{L^\infty_\mu(\mathcal{X};\mathcal{Y})}/\sqrt{m/L} + \|F - q \circ \iota\|_{L^2_\mu(\mathcal{X};\mathcal{Y})} \right) &\lesssim a_{\mathcal{Y}}\|F - q \circ \iota\|_{L^\infty_\mu(\mathcal{X};\mathcal{Y})} \\
&\lesssim a_{\mathcal{Y}} \cdot C(\boldsymbol{b}, p, \xi) \cdot (m/L)^{1-1/p} \\
&= E_{\mathsf{app},2}
\end{aligned}$$

and likewise for the $L^\infty_\mu$-norm bound. Combining this with (D.37) now completes the proof. $\qquad\square$

*Proof of Theorem 3.1 when $\mathcal{Y}$ is a Hilbert space.* The two differences in theorem statement when $\mathcal{Y}$ is a Hilbert space are: (i) the $L^2$-norm error is with respect to the stronger Bochner norm, and (ii) the approximation error terms $E_{\mathsf{app},q}$, $q = 2, \infty$, are smaller by a factor of $1/2$ in the exponent (recall (3.6)). We treat both issues separately.

For the (i), we commence with the supporting results in §D.2. First, we note that Lemmas D.1 and D.2 also hold in the Bochner $L^2$-norm, since these results already give upper bounds involving the Pettis $L^2$-norm. Next, we observe that the proof of Lemma D.3 is readily adapted to yield an equivalent result in the Bochner $L^2$-norm with the same constant $\delta$. The same therefore applies to Lemma D.4.

We next consider the analysis of (D.3) in §D.3. If we replace (D.7) by the condition

$$\|p - q\|_{\mathsf{disc},\varsigma} \geq \alpha \max\{\|p - q\|_{L^2_\varrho(D;\mathcal{Y})}, \|p - q\|_{L^2_\varsigma(D;\mathcal{Y})}\}, \quad \forall p, q \in \mathcal{P}_{\mathcal{S};\widetilde{\mathcal{Y}}} \tag{D.38}$$

then the proof of Lemma D.5 yields the same error bounds, except with the Pettis $L^2$-norm replaced by the Bochner $L^2$-norm. Theorem D.6 is likewise modified to provide a bound in the Bochner $L^2$-norm.

Up to this point, we have not used the fact that $\mathcal{Y}$ is a Hilbert space. We now need this property. As in §D.4, the next step is to establish that (D.38) holds with high probability. Lemma D.7 is unchanged, therefore our focus is on Lemma D.8. We now describe the steps needed to modify the proof of this lemma to assert (D.38) subject to the same conditions (D.13)-(D.14). First, let $\{\varphi_i\}_{i=1}^{d_{\mathcal{Y}}}$ be an orthonormal basis of $\widetilde{\mathcal{Y}}$ and write each coefficient $c_{\boldsymbol{\nu}_i}$ of the function $h$ in (D.16) as $c_{\boldsymbol{\nu}_i} = \sum_{j=1}^{d_{\mathcal{Y}}} b_{ij}\varphi_j$ for scalars $b_{ij}$. Then it is a short exercise to write

$$\|h\|^2_{\mathsf{disc},\varsigma} = \|\boldsymbol{A}\boldsymbol{c}\|^2_{2;\mathcal{Y}} = \sum_{j=1}^{d_{\mathcal{Y}}} \|\boldsymbol{A}\boldsymbol{b}_j\|^2_2,$$

where $\boldsymbol{b}_j = (b_{ij})_{i=1}^N$. Since $b_{ij} = 0$, $\forall j$, whenever $c_{\boldsymbol{\nu}_i} = 0$, this vector also satisfies $\|\boldsymbol{b}_j\|_{0,\boldsymbol{v}} \leq 2k$. Hence, by Lemma D.7 and Parseval's identity twice,

$$\begin{aligned}
\|h\|^2_{\mathsf{disc},\varsigma} &\geq \sum_{j=1}^{d_{\mathcal{Y}}} (\theta_- - (1 + \theta_+ c_5\delta))\|\boldsymbol{b}_j\|^2_2 \\
&= (\theta_- - (1 + \theta_+ c_5\delta)) \sum_{i=1}^N \|c_{\boldsymbol{\nu}_i}\|^2_2 \\
&= (\theta_- - (1 + \theta_+ c_5\delta))\|h\|^2_{L^2_\varrho(D;\mathcal{Y})}.
\end{aligned}$$

We now use the bounds (D.15) (which are unchanged) and set $\delta = c_3/(2(1 + c_4)c_5)$ once more to get

$$\|h\|^2_{\mathsf{disc},\varsigma} \geq c_3/2\|h\|^2_{L^2_\varrho(D;\mathcal{Y})}.$$

This gives the desired result.

This completes the changes needed in order to analyze the polynomial training problem (D.3). We next consider the DNN training problem (2.5). §D.5 remains unchanged. After reviewing their proofs,

we see that Lemma D.12 and Theorem D.13 both hold with Pettis norms replaced by Bochner norms whenever (D.38) holds instead of (D.7). Finally, we also observe that Lemma D.15 also holds with Pettis norms replaced by Bochner norms, both in the bound and in the definition (D.24) of $E_2(F, q)$ and $E_\infty(F, q)$.

Having completed the changes needed in all the preparatory results, we now follow the same steps as above in the Banach space case. Steps 1 and 2 are unchanged. For Step 3, we go through and replace Pettis norms by Bochner norms throughout. Using this, we obtain (D.37), except with the Bochner norm on the left-hand side in the first inequality, i.e.,

$$
\begin{aligned}
\|F - \widehat{F}\|_{L^2_\mu(\mathcal{X};\mathcal{Y})} &\lesssim a_\mathcal{Y} \left( \|F - q \circ \iota\|_{L^\infty_\mu(\mathcal{X};\mathcal{Y})} / \sqrt{m/L} + \|F - q \circ \iota\|_{L^2_\mu(\mathcal{X};\mathcal{Y})} \right) \\
&\quad + E_{\mathcal{X},2} + E_{\mathcal{Y},2} + E_{\mathsf{opt},2} + E_{\mathsf{samp},2} \\
\|F - \widehat{F}\|_{L^\infty_\mu(\mathcal{X};\mathcal{Y})} &\lesssim a_\mathcal{Y} \left( \|F - q \circ \iota\|_{L^\infty_\mu(\mathcal{X};\mathcal{Y})} + \sqrt{m/L} \|F - q \circ \iota\|_{L^2_\mu(\mathcal{X};\mathcal{Y})} \right) \\
&\quad + E_{\mathcal{X},\infty} + E_{\mathcal{Y},\infty} + E_{\mathsf{opt},\infty} + E_{\mathsf{samp},\infty}.
\end{aligned}
\tag{D.39}
$$

This concludes the changes needed to address (i). To address (ii), we bound the terms $F - q \circ \iota$. Using (A.I), (A.II), Lemma D.17, the fact that $\mathcal{Y}$ is a Hilbert space and the definitions (D.31) and (D.29) of $k$ and $n$, we see that

$$
\|F - q \circ \iota\|_{L^\infty_\mu(\mathcal{X};\mathcal{Y})} = \|f - q\|_{L^\infty_\varsigma(D;\mathcal{Y})} \lesssim \|f - q\|_{L^\infty_\varrho(D;\mathcal{Y})} \lesssim C(\boldsymbol{b}, p, \xi) \cdot (m/L)^{1-1/p}
$$

and

$$
\|F - q \circ \iota\|_{L^2_\mu(\mathcal{X};\mathcal{Y})} = \|f - q\|_{L^2_\varsigma(D;\mathcal{Y})} \lesssim \|f - q\|_{L^2_\varrho(D;\mathcal{Y})} \lesssim C(\boldsymbol{b}, p, \xi) \cdot (m/L)^{1/2-1/p}
$$

Substituting this into (D.39) and recalling (3.6) now completes the proof. $\qquad\square$

**Remark D.18 (Differences between the Banach and Hilbert space case)** Having seen the proof, we now summarize these differences as follows. First, the matter of whether the Pettis versus Bochner norm can be used reduces to the choice of such norm in the discrete metric inequality (D.7). When $\mathcal{Y}$ is a Banach space, we are able to establish this in terms of the Pettis norm subject to a log-linear scaling between $m$ and $k$ (see Lemma D.8 and (D.13)). However, when $\mathcal{Y}$ is also a Hilbert space, we can establish the stronger version (D.38) of this inequality by exploiting the additional structure. This, in short, is what leads to the stronger norm bound in this case.

Second, in the Hilbert space case, we get an improved approximation error. This stems from (D.17) and, specifically, the fact that when $\mathcal{Y}$ is a Hilbert space we may use Parseval's identity in the Bochner space $L^2_\varrho(D;\mathcal{Y})$ to bound the $L^2$-norm error term via (D.28). This is not possible when $\mathcal{Y}$ is a Banach space, so we settle for bounding this term via (D.27) instead.

# E   Proof of Theorem 3.2

## E.1   Setup

As in §D.1, let $\Lambda \subset \mathcal{F}$ with $\mathrm{supp}(\boldsymbol{\nu}) \subseteq \{1, \ldots, d_\mathcal{X}\}$, $\forall \boldsymbol{\nu} \in \Lambda$, and write $N = |\Lambda|$. Let $\mathcal{S}$ be as in (D.1), $r \in \mathbb{N}$, $r > \max\{m, k\}$ (its precise value will be chosen later in the proof) and define the set

$$
\Gamma = \Gamma_o \cup \bigcup_{S \in \mathcal{S}} S \subset \mathcal{F}, \quad \text{where } \Gamma_o = \{\boldsymbol{e}_i : i = 1, \ldots, r\}.
\tag{E.1}
$$

Finally, let $0 < \delta < 1$ and consider the family $\mathcal{N}_o$ and the tanh DNNs $\{N_{\boldsymbol{\nu}}\}_{\boldsymbol{\nu} \in \Gamma}$ whose existence is implied by Lemma D.9. We will specify $\Lambda$, $k$, $r$ and $\delta$ later in the proof.

Next, let

$$
\hat{p} = \sum_{\boldsymbol{\nu} \in S} \hat{c}_{\boldsymbol{\nu}} \Psi_{\boldsymbol{\nu}}
$$

be any minimizer of (D.3), where $|S|_{\boldsymbol{v}} \leq k$ and define

$$
\tilde{p} = \sum_{\boldsymbol{\nu} \in S} \hat{c}_{\boldsymbol{\nu}} N_{\boldsymbol{\nu}}.
$$

Let

$$\boldsymbol{B} = \frac{1}{\sqrt{r}} \left( N_{\boldsymbol{e}_j} \circ \mathcal{E}_{\mathcal{X}}(X_i) \right)_{i,j=1}^{m,r}.$$

Now, $\widetilde{\mathcal{Y}} := \mathcal{D}_{\mathcal{Y}}(\mathbb{R}^{d_{\mathcal{Y}}})$ is a finite-dimensional subspace of the Banach space $\mathcal{Y}$. Hence, for any $Y \in \mathcal{Y}$ there exists a closest point $\widetilde{Y} \in \widetilde{\mathcal{Y}}$, i.e., a point satisfying

$$\|Y - \widetilde{Y}\|_{\mathcal{Y}} = \inf \left\{ \|Y - Z\|_{\mathcal{Y}} : Z \in \widetilde{\mathcal{Y}} \right\}.$$

Given $Y_1, \ldots, Y_m$, let $\widetilde{Y}_1, \ldots, \widetilde{Y}_m \in \widetilde{\mathcal{Y}}$ be the corresponding closest points. Now define

$$\boldsymbol{e} = \frac{1}{\sqrt{r}} \left( \widetilde{Y}_i - \tilde{p} \circ \mathcal{E}_{\mathcal{X}}(X_i) \right)_{i=1}^{m} \in \widetilde{\mathcal{Y}}^m$$

and

$$\bar{p} = \tilde{p} + \sum_{i=1}^{r} (\boldsymbol{B}^\dagger \boldsymbol{e} + y\boldsymbol{z})_i N_{\boldsymbol{e}_i}, \tag{E.2}$$

where $\boldsymbol{z} \in N(\boldsymbol{B}) \backslash \{\boldsymbol{0}\}$ and $y \in \widetilde{\mathcal{Y}}$, $\|y\|_{\mathcal{Y}} = 1$, are arbitrary. Note that such a $\boldsymbol{z}$ exists, since $r > m$ by assumption. Notice that

$$\bar{p} = \sum_{\boldsymbol{\nu} \in S \cup \Gamma_o} \bar{c}_{\boldsymbol{\nu}} N_{\boldsymbol{\nu}}$$

for coefficients $\bar{c}_{\boldsymbol{\nu}} \in \widetilde{\mathcal{Y}}$. Therefore, we can write

$$\bar{p} = \mathcal{D}_{\mathcal{Y}} \circ \widehat{N}, \tag{E.3}$$

where

$$\widehat{N} = \boldsymbol{C} \begin{bmatrix} N_{\boldsymbol{\nu}_1} \\ \vdots \\ N_{\boldsymbol{\nu}_{|S \cup \Gamma_o|}} \\ 0 \\ \vdots \\ 0 \end{bmatrix} \tag{E.4}$$

and $\boldsymbol{C} \in \mathbb{R}^{d_{\mathcal{Y}} \times (\lfloor k \rfloor + r)}$. Finally, we define the approximation

$$\widehat{F} = \mathcal{D}_{\mathcal{Y}} \circ \widehat{N} \circ \mathcal{E}_{\mathcal{X}} \tag{E.5}$$

and, for convenience,

$$\check{F} = \hat{p} \circ \mathcal{E}_{\mathcal{X}}.$$

## E.2  Estimation of the DNN minimizer

**Lemma E.1** ($\widehat{N}$ is a minimizer). *If $\boldsymbol{B}$ is full rank, then*

$$\widehat{F}(X_i) = \widetilde{Y}_i, \quad \forall i = 1, \ldots, m.$$

*Therefore, $\widehat{N}$ is a minimizer of* (2.5).

*Proof.* Observe that

$$\begin{aligned} \widehat{F}(X_i) &= \bar{p} \circ \mathcal{E}_{\mathcal{X}}(X_i) \\ &= \tilde{p} \circ \mathcal{E}_{\mathcal{X}}(X_i) + \sqrt{r} \sum_{i=1}^{r} (\boldsymbol{B})_{ij} (\boldsymbol{B}^\dagger \boldsymbol{e} + y\boldsymbol{z})_j \\ &= \tilde{p} \circ \mathcal{E}_{\mathcal{X}}(X_i) + \sqrt{r} (\boldsymbol{B}(\boldsymbol{B}^\dagger \boldsymbol{e} + y\boldsymbol{z}))_i \\ &= \tilde{p} \circ \mathcal{E}_{\mathcal{X}}(X_i) + \sqrt{r}(\boldsymbol{e})_i \\ &= \widetilde{Y}_i. \end{aligned}$$

Here, in the penultimate step we use the facts that $\boldsymbol{Byz} = y\boldsymbol{Bz} = \boldsymbol{0}$ since $\boldsymbol{z} \in N(\boldsymbol{B})$ and $\boldsymbol{BB}^\dagger = \boldsymbol{I}$ since $r \geq m$ and $\boldsymbol{B}$ is full rank by assumption. This gives the first result.

For the second result, we recall that $\widetilde{Y}_i$ is a closest point to $Y_i$ from $\widetilde{\mathcal{Y}} = \mathcal{D}_{\mathcal{Y}}(\mathbb{R}^{d_{\mathcal{Y}}})$. Therefore, for any DNN $N$,

$$\frac{1}{m}\sum_{i=1}^{m}\|Y_i - \mathcal{D}_{\mathcal{Y}} \circ N \circ \mathcal{E}_{\mathcal{X}}(X_i)\|_{\mathcal{Y}}^2 \geq \frac{1}{m}\sum_{i=1}^{m}\|Y_i - \widetilde{Y}_i\|_{\mathcal{Y}}^2 = \frac{1}{m}\sum_{i=1}^{m}\|Y_i - \mathcal{D}_{\mathcal{Y}} \circ \widehat{N} \circ \mathcal{E}_{\mathcal{X}}(X_i)\|_{\mathcal{Y}}^2$$

as required. $\qquad\square$

**Lemma E.2** (Bounding $\widehat{F}$ in terms of $\breve{F}$). *Suppose that* (D.7) *holds with* $\alpha \geq c_0$ *and* $\alpha - \delta\sqrt{k} \geq c_1$, *and also that*

$$\sqrt{r}\|\iota_{d_{\mathcal{X}}} - \iota_{d_{\mathcal{X}}} \circ \widetilde{\mathcal{D}}_{\mathcal{X}} \circ \widetilde{\mathcal{E}}_{\mathcal{X}}\|_{L_\mu^2(\mathcal{X};\ell^\infty(\mathbb{N}))} \leq c_2,$$

*where* $c_0, c_1, c_2 > 0$ *are suitable universal constants. Then the approximation* $\widehat{F}$ *satisfies*

$$\begin{aligned}
\|\!|F - \widehat{F}|\!\|_{L_\mu^2(\mathcal{X};\mathcal{Y})} &\lesssim \|\!|F - \breve{F}|\!\|_{L_\mu^2(\mathcal{X};\mathcal{Y})} \\
&\quad + (1 + \delta\sqrt{r})\delta\sqrt{k}\left(1 + \frac{\sqrt{m}}{\sqrt{r}\sigma_{\min}(\boldsymbol{B})}\right)\left(\|F\|_{L_\mu^\infty(\mathcal{X};\mathcal{Y})} + \frac{1}{\sqrt{m}}\|\boldsymbol{E}\|_{2;\mathcal{Y}}\right) \\
&\quad + \frac{\sqrt{m}(1 + \delta\sqrt{r})}{\sqrt{r}\sigma_{\min}(\boldsymbol{B})}\left(\sqrt{\frac{1}{m}\sum_{i=1}^{m}\|\widetilde{Y}_i - \breve{F}(X_i)\|_{\mathcal{Y}}^2} + \frac{1}{\sqrt{m}}\|\boldsymbol{E}\|_{2;\mathcal{Y}}\right) \\
&\quad + (1 + \delta\sqrt{r})\|\boldsymbol{z}\|_2
\end{aligned}$$

*and*

$$\begin{aligned}
\|\!|F - \widehat{F}|\!\|_{L_\mu^\infty(\mathcal{X};\mathcal{Y})} &\lesssim \|\!|F - \breve{F}|\!\|_{L_\mu^\infty(\mathcal{X};\mathcal{Y})} \\
&\quad + (1 + \delta)\sqrt{r}\delta\sqrt{k}\left(1 + \frac{\sqrt{m}}{\sqrt{r}\sigma_{\min}(\boldsymbol{B})}\right)\left(\|F\|_{L_\mu^\infty(\mathcal{X};\mathcal{Y})} + \frac{1}{\sqrt{m}}\|\boldsymbol{E}\|_{2;\mathcal{Y}}\right) \\
&\quad + \frac{\sqrt{m}(1 + \delta)}{\sigma_{\min}(\boldsymbol{B})}\left(\sqrt{\frac{1}{m}\sum_{i=1}^{m}\|\widetilde{Y}_i - \breve{F}(X_i)\|_{\mathcal{Y}}^2} + \frac{1}{\sqrt{m}}\|\boldsymbol{E}\|_{2;\mathcal{Y}}\right) \\
&\quad + (1 + \delta)\sqrt{r}\|\boldsymbol{z}\|_2.
\end{aligned}$$

*Proof.* By the triangle inequality,

$$\begin{aligned}
\|\!|F - \widehat{F}|\!\|_{L_\mu^2(\mathcal{X};\mathcal{Y})} &\leq \|\!|F - \breve{F}|\!\|_{L_\mu^2(\mathcal{X};\mathcal{Y})} + \|\!|\breve{F} - \widehat{F}|\!\|_{L_\mu^2(\mathcal{X};\mathcal{Y})} \\
\|\!|F - \widehat{F}|\!\|_{L_\mu^\infty(\mathcal{X};\mathcal{Y})} &\leq \|\!|F - \breve{F}|\!\|_{L_\mu^\infty(\mathcal{X};\mathcal{Y})} + \|\!|\breve{F} - \widehat{F}|\!\|_{L_\mu^\infty(\mathcal{X};\mathcal{Y})}
\end{aligned} \tag{E.6}$$

Consider the second term. We have

$$\begin{aligned}
\|\!|\breve{F} - \widehat{F}|\!\|_{L_\mu^2(\mathcal{X};\mathcal{Y})} &= \|\!|\hat{p} - \mathcal{D}_{\mathcal{Y}} \circ \widehat{N}|\!\|_{L_{\breve{\xi}}^2(D;\mathcal{Y})} \leq \|\!|\hat{p} - \tilde{p}|\!\|_{L_{\breve{\xi}}^2(D;\mathcal{Y})} + \|\!|q|\!\|_{L_{\breve{\xi}}^2(D;\mathcal{Y})}, \\
\|\!|\breve{F} - \widehat{F}|\!\|_{L_\mu^\infty(\mathcal{X};\mathcal{Y})} &= \|\!|\hat{p} - \mathcal{D}_{\mathcal{Y}} \circ \widehat{N}|\!\|_{L_{\breve{\xi}}^\infty(D;\mathcal{Y})} \leq \|\!|\hat{p} - \tilde{p}|\!\|_{L_{\breve{\xi}}^\infty(D;\mathcal{Y})} + \|\!|q|\!\|_{L_{\breve{\xi}}^\infty(D;\mathcal{Y})},
\end{aligned} \tag{E.7}$$

where $q = \bar{p} - \tilde{p} = \sum_{i=1}^{r}(\boldsymbol{B}^\dagger\boldsymbol{e} + y\boldsymbol{z})_i N_{\boldsymbol{e}_i}$. Lemma D.10 and (A.III) give that

$$\|\!|\hat{p} - \tilde{p}|\!\|_{L_{\breve{\xi}}^2(D;\mathcal{Y})} = \|\!|\hat{p} - \tilde{p}|\!\|_{L_{\breve{\xi}}^\infty(D;\mathcal{Y})} \leq \delta\sqrt{k}\|\!|\hat{p}|\!\|_{L_{\varrho}^2(D;\mathcal{Y})}. \tag{E.8}$$

Now consider the other term in (E.7). Define $\tilde{q} = \sum_{i=1}^{r}(\boldsymbol{B}^\dagger\boldsymbol{e} + y\boldsymbol{z})_i \Psi_{\boldsymbol{e}_i}$. Then Lemma D.10 and (A.III) once more give that

$$\begin{aligned}
\|\!|q|\!\|_{L_{\breve{\xi}}^2(D;\mathcal{Y})} &\leq \|\!|\tilde{q}|\!\|_{L_{\breve{\xi}}^2(D;\mathcal{Y})} + \|\!|q - \tilde{q}|\!\|_{L_{\breve{\xi}}^2(D;\mathcal{Y})} \leq \|\!|\tilde{q}|\!\|_{L_{\breve{\xi}}^2(D;\mathcal{Y})} + \delta\sqrt{r}\|\!|\tilde{q}|\!\|_{L_{\varrho}^2(D;\mathcal{Y})} \\
\|\!|q|\!\|_{L_{\breve{\xi}}^\infty(D;\mathcal{Y})} &\leq \|\!|\tilde{q}|\!\|_{L_{\breve{\xi}}^\infty(D;\mathcal{Y})} + \|\!|q - \tilde{q}|\!\|_{L_{\breve{\xi}}^\infty(D;\mathcal{Y})} \leq \|\!|\tilde{q}|\!\|_{L_{\breve{\xi}}^\infty(D;\mathcal{Y})} + \delta\sqrt{r}\|\!|\tilde{q}|\!\|_{L_{\varrho}^2(D;\mathcal{Y})}.
\end{aligned}$$

Here, we also used the fact that $|\Gamma_o| = r$. We now apply Lemmas D.1 and D.4 and (A.III) once more to get

$$\||q\||_{L^2_{\tilde{\varsigma}}(D;\mathcal{Y})} \lesssim (1 + \delta\sqrt{r})\||\tilde{q}\||_{L^2_{\varrho}(D;\mathcal{Y})}, \qquad \||q\||_{L^\infty_{\tilde{\varsigma}}(D;\mathcal{Y})} \lesssim (1 + \delta)\sqrt{r}\||\tilde{q}\||_{L^2_{\varrho}(D;\mathcal{Y})}. \tag{E.9}$$

We next analyze the term $\||\tilde{q}\||_{L^2_{\varrho}(D;\mathcal{Y})}$. Let $y^* \in \mathcal{Y}^*$. Since $y^*((\boldsymbol{B}^\dagger\boldsymbol{e} + y\boldsymbol{z})_i) = (\boldsymbol{B}^\dagger y^*(\boldsymbol{e}) + y^*(y)\boldsymbol{z})_i$, we have

$$y^*(\tilde{q}) = \sum_{i=1}^r (\boldsymbol{B}^\dagger y^*(\boldsymbol{e}) + y^*(y)\boldsymbol{z})_i \Psi_{\boldsymbol{e}_i}.$$

Hence, by Parseval's identity,

$$\begin{aligned}
\||\tilde{q}\||_{L^2_{\varrho}(D;\mathcal{Y})} &= \sup_{y^* \in B(\mathcal{Y}^*)} \|y^*(\tilde{q})\|_{L^2(D)} \\
&= \sup_{y^* \in B(\mathcal{Y}^*)} \|\boldsymbol{B}^\dagger y^*(\boldsymbol{e}) + y^*(y)\boldsymbol{z}\|_2 \\
&\leq \frac{1}{\sigma_{\min}(\boldsymbol{B})} \sup_{y^* \in B(\mathcal{Y}^*)} \|y^*(\boldsymbol{e})\|_2 + \sup_{y^* \in B(\mathcal{Y}^*)} |y^*(y)|\|\boldsymbol{z}\|_2 \\
&= \frac{1}{\sigma_{\min}(\boldsymbol{B})} \||\boldsymbol{e}\||_{2;\mathcal{Y}} + \|\boldsymbol{z}\|_2.
\end{aligned}$$

We now use the definition of $\boldsymbol{e}$ and the inequality $\||\boldsymbol{e}\||_{2;\mathcal{Y}} \leq \|\boldsymbol{e}\|_{2;\mathcal{Y}}$ to obtain

$$\||\tilde{q}\||_{L^2_{\varrho}(D;\mathcal{Y})} \leq \frac{\sqrt{m}}{\sqrt{r}\sigma_{\min}(\boldsymbol{B})} \sqrt{\frac{1}{m}\sum_{i=1}^m \|\widetilde{Y}_i - \tilde{p} \circ \mathcal{E}_{\mathcal{X}}(X_i)\|_{\mathcal{Y}}^2} + \|\boldsymbol{z}\|_2.$$

We next apply the triangle inequality and Lemma D.10 once more to get

$$\begin{aligned}
\||\tilde{q}\||_{L^2_{\varrho}(D;\mathcal{Y})} &\leq \frac{\sqrt{m}}{\sqrt{r}\sigma_{\min}(\boldsymbol{B})} \left( \sqrt{\frac{1}{m}\sum_{i=1}^m \|\widetilde{Y}_i - \hat{p} \circ \mathcal{E}_{\mathcal{X}}(X_i)\|_{\mathcal{Y}}^2} + \|\hat{p} - \tilde{p}\|_{\mathsf{disc},\tilde{\varsigma}} \right) + \|\boldsymbol{z}\|_2 \\
&\leq \frac{\sqrt{m}}{\sqrt{r}\sigma_{\min}(\boldsymbol{B})} \left( \sqrt{\frac{1}{m}\sum_{i=1}^m \|\widetilde{Y}_i - \hat{p} \circ \mathcal{E}_{\mathcal{X}}(X_i)\|_{\mathcal{Y}}^2} + \delta\sqrt{k}\||\hat{p}\||_{L^2_{\varrho}(D;\mathcal{Y})} \right) + \|\boldsymbol{z}\|_2.
\end{aligned}$$

Combining this with (E.7) and (E.9), we deduce that

$$\begin{aligned}
\||\widecheck{F} - \widehat{F}\||_{L^2_\mu(\mathcal{X};\mathcal{Y})} \lesssim (1 + \delta\sqrt{r}) &\left[ \delta\sqrt{k}\left(1 + \frac{\sqrt{m}}{\sqrt{r}\sigma_{\min}(\boldsymbol{B})}\right)\||\hat{p}\||_{L^2_{\varrho}(D;\mathcal{Y})} + \|\boldsymbol{z}\|_2 \right. \\
&\left. + \frac{\sqrt{m}}{\sqrt{r}\sigma_{\min}(\boldsymbol{B})} \left( \sqrt{\frac{1}{m}\sum_{i=1}^m \|\widetilde{Y}_i - \hat{p} \circ \mathcal{E}_{\mathcal{X}}(X_i)\|_{\mathcal{Y}}^2} + \frac{1}{\sqrt{m}}\|\boldsymbol{E}\|_{2;\mathcal{Y}} \right) \right]
\end{aligned}$$

and

$$\begin{aligned}
\||\widecheck{F} - \widehat{F}\||_{L^\infty_\mu(\mathcal{X};\mathcal{Y})} \lesssim (1 + \delta)\sqrt{r} &\left[ \delta\sqrt{k}\left(1 + \frac{\sqrt{m}}{\sqrt{r}\sigma_{\min}(\boldsymbol{B})}\right)\||\hat{p}\||_{L^2_{\varrho}(D;\mathcal{Y})} + \|\boldsymbol{z}\|_2 \right. \\
&\left. + \frac{\sqrt{m}}{\sqrt{r}\sigma_{\min}(\boldsymbol{B})} \left( \sqrt{\frac{1}{m}\sum_{i=1}^m \|\widetilde{Y}_i - \hat{p} \circ \mathcal{E}_{\mathcal{X}}(X_i)\|_{\mathcal{Y}}^2} + \frac{1}{\sqrt{m}}\|\boldsymbol{E}\|_{2;\mathcal{Y}} \right) \right].
\end{aligned}$$

It remains to bound the term $\|\hat{p}\|_{L^2_\varrho(D;\mathcal{Y})}$. Using (D.7) and the fact that $\hat{p}$ is a minimizer of (D.3) and that the zero polynomial is feasible for (D.3), we get

$$\|\hat{p}\|_{L^2_\varrho(D;\mathcal{Y})} \lesssim \|\hat{p}\|_{\mathsf{disc},\tilde{\xi}}$$

$$\leq \sqrt{\frac{1}{m}\sum_{i=1}^{m}\|Y_i - \hat{p}\circ\mathcal{E}_{\mathcal{X}}(X_i)\|_{\mathcal{Y}}^2 + \frac{1}{\sqrt{m}}\|\boldsymbol{Y}\|_{2;\mathcal{Y}}}$$

$$\leq \frac{2}{\sqrt{m}}\|\boldsymbol{Y}\|_{2;\mathcal{Y}}$$

$$\leq 2\|F\|_{L^\infty_\mu(\mathcal{X};\mathcal{Y})} + \frac{2}{\sqrt{m}}\|\boldsymbol{E}\|_{2;\mathcal{Y}}.$$

Combining this with the previous bound and (E.6) now completes the proof. $\qquad\square$

### E.3   Estimation of $\sigma_{\min}(\boldsymbol{B})$

Recall that a matrix $\boldsymbol{A} \in \mathbb{R}^{m\times n}$ is a *subgaussian random matrix* if its entries are i.i.d. subgaussian random variables with mean zero and variance one (see, e.g., [29, Def. 9.1]). The following result can be found in, e.g., [29, Ex. 9.3].

**Lemma E.3** (Smallest singular value of a subgaussian random matrix)**.** *Let* $\boldsymbol{A} \in \mathbb{R}^{m\times n}$ *be a subgaussian random matrix and* $\sigma_{\min}$ *be the smallest singular value of* $\frac{1}{\sqrt{m}}\boldsymbol{A}$*. Then, for all* $0 < t < 1$,

$$\mathbb{P}\left(\sigma_{\min} \leq 1 - c_1\sqrt{n/m} - t\right) \leq 2\exp(-c_2 m t^2),$$

*where* $c_1, c_2 > 0$ *are universal constants.*

**Lemma E.4** (Bounding $\sigma_{\min}(\boldsymbol{B})$)**.** *Suppose that* $\sqrt{m}\delta \leq \sqrt{\omega}/8$,

$$\frac{3^{(5+\xi)/2}\sqrt{r}}{2}\|\iota_{d_{\mathcal{X}}} - \iota_{d_{\mathcal{X}}}\circ\widetilde{\mathcal{D}}_{\mathcal{X}}\circ\widetilde{\mathcal{E}}_{\mathcal{X}}\|_{L^\infty_\mu(\mathcal{X},\ell^\infty(\mathbb{N}))} \leq \frac{\sqrt{\omega}}{8}, \tag{E.10}$$

*where* $\omega$ *is the variance of the univariate probability measure as specified in Theorem 3.2 and*

$$d_{\mathcal{X}} \geq r \geq c\left(m + \log(2/\epsilon)\right) \tag{E.11}$$

*for some universal constant* $c \geq 1$*. Then, with probability at least* $1 - \epsilon$*, the matrix* $\boldsymbol{B}$ *is full rank and*

$$\sigma_{\min}(\boldsymbol{B}) \geq \sqrt{\omega}/4.$$

*Proof.* Define the matrices

$$\boldsymbol{B}' = \frac{1}{\sqrt{r}}\left(\Psi_{\boldsymbol{e}_j}\circ\mathcal{E}_{\mathcal{X}}(X_i)\right)_{i,j=1}^{m,r}, \quad \boldsymbol{B}'' = \frac{1}{\sqrt{r}}\left(\Psi_{\boldsymbol{e}_j}\circ\iota_{d_{\mathcal{X}}}(X_i)\right)_{i,j=1}^{m,r}.$$

Then, since $r \geq m$,

$$\sigma_{\min}(\boldsymbol{B}) = \inf\left\{\|\boldsymbol{B}^\top\boldsymbol{d}\|_2 : \boldsymbol{d}\in\mathbb{C}^m, \|\boldsymbol{d}\|_2 = 1\right\}$$

$$\geq \sigma_{\min}(\boldsymbol{B}') - \|(\boldsymbol{B} - \boldsymbol{B}')^\top\|_2$$

$$= \sigma_{\min}(\boldsymbol{B}') - \|\boldsymbol{B} - \boldsymbol{B}'\|_2$$

$$\geq \sigma_{\min}(\boldsymbol{B}'') - \|\boldsymbol{B} - \boldsymbol{B}'\|_2 - \|\boldsymbol{B}' - \boldsymbol{B}''\|_2.$$

Now, for any $\boldsymbol{c}\in\mathbb{C}^r$,

$$\|(\boldsymbol{B} - \boldsymbol{B}')\boldsymbol{c}\|_2^2 = \frac{1}{r}\sum_{i=1}^{m}\left(\sum_{j=1}^{r}\left(N_{\boldsymbol{e}_j}(\boldsymbol{x}_i) - \Psi_{\boldsymbol{e}_j}(\boldsymbol{x}_i)\right)c_j\right)^2$$

$$\leq \frac{1}{r}\sum_{i=1}^{m}\delta^2\|\boldsymbol{c}\|_1^2$$

$$\leq m\delta^2\|\boldsymbol{c}\|_2^2.$$

We deduce that $\|\boldsymbol{B} - \boldsymbol{B}'\|_2 \leq \sqrt{m}\delta \leq \sqrt{\omega}/8$. Hence

$$\sigma_{\min}(\boldsymbol{B}) \geq \sigma_{\min}(\boldsymbol{B}'') - \sqrt{\omega}/8 - \|\boldsymbol{B}' - \boldsymbol{B}''\|_2.$$

Now let $\boldsymbol{c} \in \mathbb{C}^r$ and $p = \sum_{i=1}^r c_i \Psi_{\boldsymbol{e}_i}$ be the corresponding polynomial. Then, by (A.I), (A.III) and Lemma D.2,

$$
\begin{aligned}
\|(\boldsymbol{B}' - \boldsymbol{B}'')\boldsymbol{c}\|_2 &= \sqrt{\frac{1}{r}\sum_{i=1}^m \left| p \circ \iota_{d_{\mathcal{X}}}(X_i) - p \circ \iota_{d_{\mathcal{X}}} \circ \widetilde{\mathcal{D}}_{\mathcal{X}} \circ \widetilde{\mathcal{E}}_{\mathcal{X}}(X_i) \right|^2} \\
&\leq \mathrm{Lip}(p, B^\infty(\mathbb{N}), \mathbb{R}) \| \iota_{d_{\mathcal{X}}} - \iota_{d_{\mathcal{X}}} \circ \widetilde{\mathcal{D}}_{\mathcal{X}} \circ \widetilde{\mathcal{E}}_{\mathcal{X}} \|_{L^\infty_\mu(\mathcal{X}, \ell^\infty(\mathbb{N}))} \\
&\leq \frac{3^{(5+\xi)/2}\sqrt{r}}{2} \|p\|_{L^2_\varrho(D)} \| \iota_{d_{\mathcal{X}}} - \iota_{d_{\mathcal{X}}} \circ \widetilde{\mathcal{D}}_{\mathcal{X}} \circ \widetilde{\mathcal{E}}_{\mathcal{X}} \|_{L^\infty_\mu(\mathcal{X}, \ell^\infty(\mathbb{N}))} \\
&\leq \frac{\sqrt{\omega}}{8} \|\boldsymbol{c}\|_2.
\end{aligned}
$$

Here, in the third step we used the fact that $|\Gamma_o|_{\boldsymbol{v}} = 3^{5+\xi}r$, since $u_{\boldsymbol{e}_i} = \sqrt{3}$. We deduce that $\|\boldsymbol{B}' - \boldsymbol{B}''\|_2 \leq \sqrt{\omega}/8$ and therefore

$$\sigma_{\min}(\boldsymbol{B}) \geq \sigma_{\min}(\boldsymbol{B}'') - \sqrt{\omega}/4.$$

It remains to show that $\sigma_{\min}(\boldsymbol{B}'') \geq \sqrt{\omega}/2$ with high probability. By construction, $\Psi_{\boldsymbol{e}_j}(\boldsymbol{x}) = \sqrt{3}x_j$. Now recall that the pushforward $\varsigma$ is a tensor-product of a univariate probability measure supported in $[-1, 1]$ with mean zero and variance $\omega > 0$. Therefore

$$(\boldsymbol{B}'')^\top = \sqrt{3}\frac{\sqrt{\omega}}{\sqrt{r}}\boldsymbol{A},$$

where $\boldsymbol{A} = (\iota(X_j)_i/\sqrt{\omega})_{i,j=1}^{r,m} \in \mathbb{R}^{r \times m}$. By construction, the entries of $\boldsymbol{A}$ are i.i.d. subgaussian random variables with mean zero and variance one. Hence $\boldsymbol{A}$ is a subgaussian random matrix. We now apply Lemma E.3. Let $t = 1/4$ and observe that

$$r \geq 4c_1^2 m, \quad r \geq \frac{16}{c_2}\log(2/\epsilon),$$

by assumption. Therefore,

$$\mathbb{P}(\sigma_{\min}(\boldsymbol{B}'') \leq \sqrt{\omega}/2) \leq \mathbb{P}(\sigma_{\min}(\boldsymbol{A}/\sqrt{r}) \leq 1/(2\sqrt{3})) \leq \epsilon.$$

This gives the result. $\qquad\square$

## E.4 Final arguments

We are now ready to complete the proof of Theorem 3.2.

*Proof of Theorem 3.2, Statement (A).* We divide the proof into a series of steps.

*Step 1: Setup.* Let $m$, $\delta$, $\epsilon$ and $L$ be as in the theorem statement. We once more assume without loss of generality that $\delta \leq 1/5$. Let $n$ be as in (D.29) and $\Lambda = \Lambda_n^{\mathsf{HCl}}$, let $\xi$ be as in (D.30) and

$$k = \frac{m}{c_1 L}, \tag{E.12}$$

where $c_1 \geq 1$ is a constant that will be chosen in the next step. Let

$$\delta = \min\left\{2^{-m}/r^2, \sqrt{\omega}/(8\sqrt{m})\right\},$$

where $\omega$ is the variance of the univariate probability measure and

$$r = \lceil c_2(m + \log(1/\epsilon)) \rceil, \tag{E.13}$$

where $c_2 \geq 2$ will also be chosen in the next step. Note that $d_{\mathcal{X}} \geq r > m$ by assumption. Finally, let $\mathcal{S}$ be as in (D.1), $\Gamma$ be as in (E.1) and $\{N_{\boldsymbol{\nu}}\}_{\boldsymbol{\nu} \in \Gamma}$ be the corresponding family of tanh DNNs ensured by Lemma D.9. Finally, let $\widehat{F}$ be given by (E.5), with DNN $\widehat{N}$ as in (E.4).

*Step 2: $\widehat{N}$ is a minimizer.* By construction,
$$\text{width}(\widehat{N}) \lesssim (k+r) \cdot m(\Gamma), \quad \text{depth}(\widehat{N}) \lesssim \log(k).$$
If $\boldsymbol{\nu} = \boldsymbol{e}_i$, then $\|\boldsymbol{\nu}\|_1 = 1$. Hence $m(\Gamma) \leq 1 + \max_{S \in \mathcal{S}} m(S) \leq 1 + k^{1/(5+\xi)} = 1 + k^\delta$ (see §D.5). Using the values of $k$ and $r$, we see that
$$\text{width}(\widehat{N}) \lesssim (m + \log(1/\epsilon))(m/L)^\delta, \quad \text{depth}(\widehat{N}) \lesssim \log(m/L).$$
Therefore, $\widehat{N} \in \mathcal{N}$ is feasible for (2.5). Lemma E.1 now implies that $\widehat{N}$ is a minimizer, provided $\boldsymbol{B}$ is full rank. We will show that this holds in the next step.

*Step 3: Ensuring that $\sigma_{\min}(\boldsymbol{B}) \geq \sqrt{\omega}/4$ with probability at least $1 - \epsilon/4$.* We seek to use Lemma E.4. By definition of $\delta$, we have that $\sqrt{m}\delta \leq \sqrt{\omega}/8$. Now (D.34), (E.13) and (3.9) imply that (E.10)-(E.11) hold, the latter with $\epsilon$ replaced by $\epsilon/4$. Hence Lemma E.4 implies the result.

*Step 4: Ensuring* (D.7) *holds with probability at least $1 - \epsilon/4$.* This step is very similar to Step 2 of the proof of Theorem 3.1. The only difference comes in the estimation of $N$ in (D.32), since now $N = |\Gamma| \leq |\Lambda| + r$. Since $m \geq 3$, we have
$$\log(eN) \leq \log(e|\Lambda|(r+1)) \lesssim \log^2(m) + \log(m + \log(1/\epsilon)).$$
If $m \leq \log(1/\epsilon)$ then
$$\log(eN) \lesssim \log^2(m) + \log(\log(1/\epsilon)) \leq \log^2(m) + \sqrt{\log(1/\epsilon)},$$
where in the second step we use the fact that $\log(t) \leq \sqrt{t}$ for $t > 0$. Conversely, if $m \geq \log(1/\epsilon)$, then
$$\log(eN) \lesssim \log^2(m) \leq \log^2(m) + \sqrt{\log(1/\epsilon)}.$$
Therefore, (D.33) with $\epsilon/4$ reads
$$c_0 \cdot k \cdot (\log(eN) \cdot \log^2(k) + \log(6/\epsilon)) \lesssim k \cdot \left( \log^2(m) \left( \log^2(m) + \sqrt{\log(1/\epsilon)} \right) + \log(1/\epsilon) \right)$$
$$\lesssim k \cdot \left( \log^4(m) + \log(1/\epsilon) \right).$$
$$= k \cdot L(m, \epsilon).$$
Hence, due to the definition of $k$, we get that (D.13) holds once more with probability at least $1 - \epsilon/4$. The remainder of this step is identical to Step 2 of the proof of Theorem 3.1.

*Step 5: Error analysis.* We now apply Lemma E.2 to the approximation $\widehat{F}$. Since $\sigma_{\min}(\boldsymbol{B}) \gtrsim \sqrt{\omega} \gtrsim 1$, $\delta \leq \delta\sqrt{r} \leq 1$ and $r \geq m$, we deduce that
$$\|F - \widehat{F}\|_{L^2_\mu(\mathcal{X};\mathcal{Y})} \lesssim \|F - \check{F}\|_{L^2_\mu(\mathcal{X};\mathcal{Y})} + \delta\sqrt{k} \left( \|F\|_{L^\infty_\mu(\mathcal{X};\mathcal{Y})} + \frac{1}{\sqrt{m}}\|\boldsymbol{E}\|_{2;\mathcal{Y}} \right)$$
$$+ \|\boldsymbol{z}\|_2 + \sqrt{\frac{1}{m}\sum_{i=1}^m \|\widetilde{Y}_i - \check{F}(X_i)\|_\mathcal{Y}^2 + \frac{1}{\sqrt{m}}\|\boldsymbol{E}\|_{2;\mathcal{Y}}}$$
and
$$\|F - \widehat{F}\|_{L^\infty_\mu(\mathcal{X};\mathcal{Y})} \lesssim \|F - \check{F}\|_{L^\infty_\mu(\mathcal{X};\mathcal{Y})} + \sqrt{r}\delta\sqrt{k} \left( \|F\|_{L^\infty_\mu(\mathcal{X};\mathcal{Y})} + \frac{1}{\sqrt{m}}\|\boldsymbol{E}\|_{2;\mathcal{Y}} \right)$$
$$+ \sqrt{r}\|\boldsymbol{z}\|_2 + \sqrt{m} \left( \sqrt{\frac{1}{m}\sum_{i=1}^m \|\widetilde{Y}_i - \check{F}(X_i)\|_\mathcal{Y}^2} + \frac{1}{\sqrt{m}}\|\boldsymbol{E}\|_{2;\mathcal{Y}} \right)$$
with probability at least $1 - \epsilon/2$, where $\check{F} = \hat{p} \circ \mathcal{E}_\mathcal{X}$ and $\hat{p}$ is the corresponding minimizer of (D.3). We now appeal to Theorem D.6 with $\sigma = 1$ and $\tau = 0$, recalling that $\alpha \gtrsim 1$ and $\delta\sqrt{k} \lesssim 1$, to get that
$$\|F - \widehat{F}\|_{L^2_\mu(\mathcal{X};\mathcal{Y})} \lesssim \|F - \mathcal{Q} \circ F\|_{L^2_\mu(\mathcal{X};\mathcal{Y})} + \|F - \mathcal{Q} \circ F\|_{\text{disc},\mu}$$
$$+ \pi_\mathcal{Q} \left( \|F - q \circ \mathcal{E}_\mathcal{X}\|_{L^2_\mu(\mathcal{X};\mathcal{Y})} + \|F - q \circ \mathcal{E}_\mathcal{X}\|_{\text{disc},\mu} \right)$$
$$+ \frac{1}{\sqrt{m}}\|\boldsymbol{E}\|_{2;\mathcal{Y}} + \delta\sqrt{k}\|F\|_{L^\infty_\mu(\mathcal{X};\mathcal{Y})}$$
$$+ \|\boldsymbol{z}\|_2 + \sqrt{\frac{1}{m}\sum_{i=1}^m \|\widetilde{Y}_i - \check{F}(X_i)\|_\mathcal{Y}^2}$$

and, since $\delta\sqrt{r} \lesssim 1$,

$$\|F - \widehat{F}\|_{L_\mu^\infty(\mathcal{X};\mathcal{Y})} \lesssim \|F - \mathcal{Q} \circ F\|_{L_\mu^\infty(\mathcal{X};\mathcal{Y})} + \sqrt{k}\|\|F - \mathcal{Q} \circ F\|\|_{\mathsf{disc},\mu}$$
$$+ \pi_\mathcal{Q}\left(\|F - q \circ \mathcal{E}_\mathcal{X}\|_{L_\mu^\infty(\mathcal{X};\mathcal{Y})} + \sqrt{k}\|\|F - q \circ \mathcal{E}_\mathcal{X}\|\|_{\mathsf{disc},\mu}\right)$$
$$+ \|\boldsymbol{E}\|_{2;\mathcal{Y}} + \sqrt{r}\delta\sqrt{k}\|F\|_{L_\mu^\infty(\mathcal{X};\mathcal{Y})}$$
$$+ \sqrt{r}\|\boldsymbol{z}\|_2 + \sqrt{m}\sqrt{\frac{1}{m}\sum_{i=1}^m \|\widetilde{Y}_i - \breve{F}(X_i)\|_\mathcal{Y}^2}$$

for any $q \in \mathcal{P}_{\mathcal{S};\mathcal{Y}}$ and linear operator $\mathcal{Q} : \mathcal{Y} \to \widetilde{\mathcal{Y}} = \mathcal{D}_\mathcal{Y}(\mathbb{R}^{d_\mathcal{Y}})$ with $\pi_\mathcal{Q} = \|\mathcal{Q}\|_{\mathcal{Y} \to \mathcal{Y}}$. Consider the final term. Since $\widetilde{Y}_i \in \widetilde{\mathcal{Y}}$ is the closest point to $Y_i = F(X_i) + E_i$ we have

$$\|\widetilde{Y}_i - Y_i\|_\mathcal{Y} \le \|F(X_i) - \mathcal{Q} \circ F(X_i)\|_\mathcal{Y} + \|E_i\|_\mathcal{Y}.$$

We now use this, the fact that $\hat{p}$ is a minimizer and $\mathcal{Q} \circ q$ is feasible in combination with triangle inequality to get

$$\sqrt{\frac{1}{m}\sum_{i=1}^m \|\widetilde{Y}_i - \breve{F}(X_i)\|_\mathcal{Y}^2} \le \sqrt{\frac{1}{m}\sum_{i=1}^m \|Y_i - \mathcal{Q} \circ q \circ \mathcal{E}_\mathcal{X}(X_i)\|_\mathcal{Y}^2} + \|\|F - \mathcal{Q} \circ F\|\|_{\mathsf{disc},\mu} + \frac{1}{\sqrt{m}}\|\boldsymbol{E}\|_{2;\mathcal{Y}}$$

$$\le \|\|F - \mathcal{Q} \circ q \circ \mathcal{E}_\mathcal{X}\|\|_{\mathsf{disc},\mu} + \|\|F - \mathcal{Q} \circ F\|\|_{\mathsf{disc},\mu} + \frac{2}{\sqrt{m}}\|\boldsymbol{E}\|_{2;\mathcal{Y}}$$

$$\le \pi_\mathcal{Q}\|\|F - q \circ \mathcal{E}_\mathcal{X}\|\|_{\mathsf{disc},\mu} + 2\|\|F - \mathcal{Q} \circ F\|\|_{\mathsf{disc},\mu} + \frac{2}{\sqrt{m}}\|\boldsymbol{E}\|_{2;\mathcal{Y}}.$$

Now let $f \in \mathcal{H}(\boldsymbol{b})$ be the function asserted by (A.II) and $q = f_S$ be the polynomial asserted by Lemma D.17. Substituting this into the previous expression, recalling (D.35) and using the definition of $\delta$, we deduce that

$$\|F - \widehat{F}\|_{L_\mu^2(\mathcal{X};\mathcal{Y})} \lesssim \|\|F - \mathcal{Q} \circ F\|\|_{L_\mu^2(\mathcal{X};\mathcal{Y})} + \|\|F - \mathcal{Q} \circ F\|\|_{\mathsf{disc},\mu}$$
$$+ \pi_\mathcal{Q} E_2(F, q) + \|\boldsymbol{z}\|_2 + 2^{-m} + E_{\mathsf{samp},2}$$
$$\|F - \widehat{F}\|_{L_\mu^\infty(\mathcal{X};\mathcal{Y})} \lesssim \|F - \mathcal{Q} \circ F\|_{L_\mu^\infty(\mathcal{X};\mathcal{Y})} + \sqrt{m}\|\|F - \mathcal{Q} \circ F\|\|_{\mathsf{disc},\mu}$$
$$+ \pi_\mathcal{Q} \widetilde{E}_\infty(F, q) + \sqrt{r}\|\boldsymbol{z}\|_2 + 2^{-m} + E'_{\mathsf{samp},\infty}$$

with probability at least $1 - \epsilon/2$. Here $E_2(F, q)$ is as in (D.24), $\widetilde{E}_\infty(F, q)$ is as in (D.24) with $k$ replaced by $m$ and $E'_{\mathsf{samp},\infty} = \|\boldsymbol{E}\|_{2;\mathcal{Y}} = \sqrt{L}E_{\mathsf{samp},\infty}$.

Now observe that the first bound is identical to the corresponding bound in (D.36), except with $E_{\mathsf{opt},2}$ replaced by $\|\boldsymbol{z}\|_2 + 2^{-m}$. Following the same arguments as in Step 3 of the proof of Theorem 3.1, this gives the corresponding bound in (D.37), which is

$$\|F - \widehat{F}\|_{L_\mu^2(\mathcal{X};\mathcal{Y})} \lesssim a_\mathcal{Y}\left(\|F - q \circ \iota\|_{L_\mu^\infty(\mathcal{X};\mathcal{Y})}/\sqrt{m/L} + \|F - q \circ \iota\|_{L_\mu^2(\mathcal{X};\mathcal{Y})}\right) \tag{E.14}$$
$$+ E_{\mathcal{X},2} + E_{\mathcal{Y},2} + \|\boldsymbol{z}\|_2 + 2^{-m} + E_{\mathsf{samp},2}.$$

The second bound above is identical to the corresponding bound in (D.36), except with $E_{\mathsf{opt},\infty}$ replaced by $\sqrt{r}\|\boldsymbol{z}\|_2 + 2^{-m}$, $E_{\mathsf{samp},\infty}$ and $E_\infty(F, q)$ replaced by $E'_{\mathsf{samp},\infty}$ and $\widetilde{E}_\infty(F, q)$, respectively, and with $\sqrt{k}$ replaced by $\sqrt{m}$. We once more follow the same arguments as in Step 3, with these changes. This yields the corresponding version of (D.37), which is

$$\|F - \widehat{F}\|_{L_\mu^\infty(\mathcal{X};\mathcal{Y})} \lesssim a_\mathcal{Y}\left(\sqrt{L}\|F - q \circ \iota\|_{L_\mu^\infty(\mathcal{X};\mathcal{Y})} + \sqrt{m}\|F - q \circ \iota\|_{L_\mu^2(\mathcal{X};\mathcal{Y})}\right) \tag{E.15}$$
$$+ E'_{\mathcal{X},\infty} + E'_{\mathcal{Y},\infty} + \sqrt{r}\|\boldsymbol{z}\|_2 + 2^{-m} + E'_{\mathsf{samp},\infty}.$$

Here $E'_{\mathcal{X},\infty} = LE_{\mathcal{X},\infty}$ and $E'_{\mathcal{Y},\infty} = \sqrt{L}E_{\mathcal{Y},\infty}$.

Having done this, we then use the bounds from Step 4 of the proof of Theorem 3.1, to get

$$\|F - \widehat{F}\|_{L^2_\mu(\mathcal{X};\mathcal{Y})} \lesssim E_{\mathsf{app},2} + E_{\mathcal{X},2} + E_{\mathcal{Y},2} + \|\boldsymbol{z}\|_2 + 2^{-m} + E_{\mathsf{samp},2}$$

$$\|F - \widehat{F}\|_{L^\infty_\mu(\mathcal{X};\mathcal{Y})} \lesssim E'_{\mathsf{app},\infty} + E'_{\mathcal{X},\infty} + E'_{\mathcal{Y},\infty} + \sqrt{r}\|\boldsymbol{z}\|_2 + 2^{-m} + E'_{\mathsf{samp},\infty},$$

where $E'_{\mathsf{app},\infty} = \sqrt{L}E_{\mathsf{app},\infty}$.

*Step 6: Existence of uncountably many minimizers.* Let $\boldsymbol{z} = (z_i)_{i=1}^r \in N(\boldsymbol{B})\backslash\{\boldsymbol{0}\}$ be any vector and consider $\boldsymbol{z}_1 = \theta_1 \boldsymbol{z}$ and $\boldsymbol{z}_2 = \theta_2 \boldsymbol{z}$ for $\theta_1, \theta_2 \in [-1, 1]$ with $\theta_1 \neq \theta_2$. Then these vectors define functions $\bar{p}_1$ and $\bar{p}_2$ as in (E.2) and DNNs $\widehat{N}_1$ and $\widehat{N}_2$ as in (E.3). Suppose that $\widehat{N}_1 = \widehat{N}_2$. Then, since $\mathcal{D}_\mathcal{Y}$ is linear, we have that $\bar{p}_1 = \bar{p}_2$. But then, by definition and the fact that $y \in \widetilde{\mathcal{Y}}\backslash\{0\}$, we must have

$$0 = \sum_{i=1}^r (\boldsymbol{z}_1 - \boldsymbol{z}_2)_i N_{\boldsymbol{e}_i} = (\theta_1 - \theta_2) \sum_{i=1}^r z_i N_{\boldsymbol{e}_i}.$$

Suppose that $\theta_1 > \theta_2$ without loss of generality and let $\boldsymbol{x} \in D$ be the vector $(\mathrm{sign}(z_i))_{i=1}^\infty$. Then, $\Psi_{\boldsymbol{e}_i}(\boldsymbol{x}) = \sqrt{3}\mathrm{sign}(z_i)$ and therefore

$$0 \geq (\theta_1 - \theta_2)\left(\sqrt{3}\|\boldsymbol{z}\|_1 - \delta r\right) \geq (\theta_1 - \theta_2)\left(\sqrt{3}\|\boldsymbol{z}\|_2 - \delta r\right) = (\theta_1 - \theta_2)\left(\sqrt{3}\|\boldsymbol{z}\|_2 - 2^{-m}/r\right),$$

since $\|N_{\boldsymbol{e}_i} - \Psi_{\boldsymbol{e}_i}\|_{L^\infty_\varrho(D)} \leq \delta$ and $\delta \leq 2^{-m}/r^2$ by definition. We now choose $\boldsymbol{z}$ with $\|\boldsymbol{z}\|_2 = 2^{-m}/r \leq 2^{-m}$. Note that this choice of $\boldsymbol{z}$ does not change the error bound, except for a constant. It also yields

$$0 \geq (\theta_1 - \theta_2)(\sqrt{3}2^{-m} - 2^{-m}) > 0$$

which is a contradiction. Hence $\widehat{N}_1 \neq \widehat{N}_2$. Thus, we have shown that any $\theta \in [-1, 1]$ leads to a distinct DNN minimizer that satisfies the desired bounds. We get the result.

*Step 7: Modifications when $\mathcal{Y}$ is a Hilbert space.* The modifications required when $\mathcal{Y}$ is a Hilbert space are identical to those needed in the proof of Theorem 3.1 (see §D.8). We omit the details. $\square$

*Proof of Theorem 3.2, Statement (B).* Let $N = N_{\boldsymbol{\theta}} : \mathbb{R}^{d_\mathcal{X}} \to \mathbb{R}^{d_\mathcal{Y}}$ be a tanh DNN, where $\boldsymbol{\theta} \in \mathbb{R}^D$ are the network parameters (weights and biases). Let $\|\cdot\|_{(d_\mathcal{X})}$, $\|\cdot\|_{(d_\mathcal{Y})}$ and $\|\cdot\|_{(D)}$ be arbitrary norms on $\mathbb{R}^{d_\mathcal{X}}$, $\mathbb{R}^{d_\mathcal{Y}}$ and $\mathbb{R}^D$, respectively. Then, since the activation function is a Lipschitz function, we have

$$\|N_{\boldsymbol{\theta}'}(\boldsymbol{x}) - N_{\boldsymbol{\theta}}(\boldsymbol{x})\|_{(d_\mathcal{Y})} \leq c_{\boldsymbol{\theta}}\|\boldsymbol{\theta}' - \boldsymbol{\theta}\|_{(D)}(\|\boldsymbol{x}\|_{(d_\mathcal{X})} + 1),$$

where $c_{\boldsymbol{\theta}} > 0$ is a constant depending on $\boldsymbol{\theta}$.

Now let $\widehat{N} = \widehat{N}_{\hat{\boldsymbol{\theta}}}$ and $\widehat{F} = \mathcal{D}_\mathcal{Y} \circ \widehat{N} \circ \mathcal{E}_\mathcal{X}$ and consider $F = \mathcal{D}_\mathcal{Y} \circ N \circ \mathcal{E}_\mathcal{X}$ for $N = N_{\boldsymbol{\theta}}$. Since $\mathcal{D}_\mathcal{Y} : \mathbb{R}^{d_\mathcal{Y}} \to \mathcal{Y}$ is linear and therefore bounded, we have

$$\|\widehat{F}(X) - F(X)\|_{\mathcal{Y}} \leq c_{\hat{\boldsymbol{\theta}}}\|\mathcal{D}_\mathcal{Y}\|_{(\mathbb{R}^{d_\mathcal{Y}}, \|\cdot\|_{(d_\mathcal{Y})}) \to \mathcal{Y}}\|\hat{\boldsymbol{\theta}} - \boldsymbol{\theta}\|_{(D)}\left(\|\mathcal{E}_\mathcal{X}(X)\|_{(d_\mathcal{X})} + 1\right).$$

We deduce that

$$\|\widehat{F} - F\|_{L^2_\mu(\mathcal{X};\mathcal{Y})} \leq \|\widehat{F} - F\|_{L^\infty_\mu(\mathcal{X};\mathcal{Y})}$$

$$\leq c_{\hat{\boldsymbol{\theta}}}\|\mathcal{D}_\mathcal{Y}\|_{(\mathbb{R}^{d_\mathcal{Y}}, \|\cdot\|_{(d_\mathcal{Y})}) \to \mathcal{Y}}\|\hat{\boldsymbol{\theta}} - \boldsymbol{\theta}\|_{(D)}\left(\|\mathcal{E}_\mathcal{X}\|_{L^\infty_\mu(\mathcal{X};(\mathbb{R}^{d_\mathcal{X}}, \|\cdot\|_{(d_\mathcal{X})}))} + 1\right).$$

Therefore, there exists a neighbourhood around $\hat{\boldsymbol{\theta}}$ for which

$$\|\widehat{F} - F\|_{L^2_\mu(\mathcal{X};\mathcal{Y})} \leq \|\widehat{F} - F\|_{L^\infty_\mu(\mathcal{X};\mathcal{Y})} \leq \tau_o$$

for all parameters $\boldsymbol{\theta}$ in the neighbourhood. The result now follows. $\square$

*Proof of Theorem 3.2, Statement (C).* By construction, the DNN $\widehat{N}$ defined in (E.4) contains subnetworks that compute the DNNs $N_{\boldsymbol{e}_i}$, $i = 1, \ldots, r$, which themselves are approximations to the Legendre polynomials $\Psi_{\boldsymbol{e}_i}$. The construction of these subnetworks was described in the proof of

Lemma D.9 as the composition of an affine map defined by the fundamental theorem of algebra and a tanh DNN that approximately multiplies $m(\Gamma)$ numbers. In this specific case, we have

$$\Psi_{\boldsymbol{e}_i}(\boldsymbol{x}) = x_i.$$

Therefore, the corresponding affine map (D.18) $\mathcal{A}_{\boldsymbol{e}_i} : \mathbb{R}^{d_{\mathcal{X}}} \to \mathbb{R}^{m(\Gamma)}$ has a bias vector that is all ones, except for a single entry that has value zero. We deduce that the bias vector $\boldsymbol{b}$ of the full DNN (E.4) contains a subblock of size $rm(\Gamma)$ that is all ones, except for $r$ zeroes. There are $\binom{rm(\Gamma)}{r}$ rearrangements of this subblock, each of which leading to a bias vector $\boldsymbol{b}'$ with $\|\boldsymbol{b} - \boldsymbol{b}'\| \gtrsim 1$. Moreover, $\boldsymbol{b}'$ leads to the same DNN, after permuting the various weight matrices in the corresponding way. Indeed, if $\boldsymbol{P}$ is a permutation matrix, then $\sigma(\boldsymbol{W}\boldsymbol{x} + \boldsymbol{b}) = \boldsymbol{P}^{-1}\sigma(\boldsymbol{P}\boldsymbol{W}\boldsymbol{x} + \boldsymbol{P}\boldsymbol{b})$, since $\sigma$ acts componentwise.

It remains to bound $\binom{rm(\Gamma)}{r}$ from below. We have

$$\binom{rm(\Gamma)}{r} \geq \frac{(rm(\Gamma))^r}{r^r} = m(\Gamma)^r.$$

By (E.13), we have that $r \geq 2m$. Now, by the definition (E.1) of $\Gamma$,

$$m(\Gamma) \geq \max_{S \in \mathcal{S}} m(S),$$

where $\mathcal{S}$ is as in (D.1) for $k$ as in (E.12) and $\Lambda = \Lambda_n^{\mathsf{HCl}}$ as in (D.26) with $n$ as in (D.29). Now consider the set $S = \{l\boldsymbol{e}_1\}$ for some $l \in \mathbb{N}$. Then

$$|S|_{\boldsymbol{v}} = v_{\boldsymbol{\nu}}^2 = (2l+1)^{5+\xi} = (2l+1)^{1/\delta}.$$

Therefore, $S \in \mathcal{S}$ provided $(2l+1)^{1/\delta} \leq k$ and $l \leq n$. Set $l = \lfloor (k^{\delta} - 1)/2 \rfloor$ and observe that $l \leq n$ since $k \leq n$ and $\delta \leq 1/5$ by assumption. Therefore $S \in \mathcal{S}$. Using the definition of $k$, we get that

$$m(\Gamma) \geq m(S) = l \geq (m/(c_4 L))^{\delta}$$

for all sufficiently large $m$. The result now follows. $\qquad\square$

# F   Proof of Theorem 4.1

The proof of Theorem 4.1 will follow as a consequence of the following result. For this, we require the following notation. Given $0 < p \leq \infty$, $s \in \mathbb{N}$ and a sequence $\boldsymbol{c} = (c_i)_{i=1}^{\infty}$, we let

$$\sigma_s(\boldsymbol{c})_p = \min\{\|\boldsymbol{c} - \boldsymbol{z}\|_p : \boldsymbol{z} \in \ell^2, |\mathrm{supp}(\boldsymbol{z})| \leq s\},$$

where $\mathrm{supp}(\boldsymbol{z}) = \{i \in \mathbb{N} : z_i \neq 0\}$ for $\boldsymbol{z} = (z_i)_{i \in \mathbb{N}} \in \mathbb{R}^{\mathbb{N}}$.

**Theorem F.1.** *For any $0 < p < 1$ then term $\theta_m(\boldsymbol{b})$ defined in (4.2) satisfies*

$$\theta_m(\boldsymbol{b}) \gtrsim \sigma_m(\boldsymbol{b})_2, \quad \forall \boldsymbol{b} \in \ell^1(\mathbb{N}), \ \boldsymbol{b} \geq \boldsymbol{0}, \|\boldsymbol{b}\|_1 \leq 1.$$

As noted, the proof of this theorem is based on [7, Thm. 4.4]. We recap the details as they will also be needed in the proof of the next result. First, we recall some basic definitions. See [82] or [29, Ch. 10] for more details. Let $\mathcal{K}$ be a subset of a normed space $(\mathcal{X}, \|\cdot\|_{\mathcal{X}})$. Then its *Gelfand m-width* is

$$d^m(\mathcal{K}, \mathcal{X}) = \inf\left\{\sup_{x \in \mathcal{K} \cap L^m} \|x\|_{\mathcal{X}}, \ L^m \text{ a subspace of } \mathcal{X} \text{ with } \mathrm{codim}(L^m) \leq m\right\}. \qquad \text{(F.1)}$$

An equivalent representation is

$$d^m(\mathcal{K}, \mathcal{X}) = \inf\left\{\sup_{x \in \mathcal{K} \cap \mathrm{Ker}(A)} \|x\|_{\mathcal{X}}, \ A : \mathcal{X} \to \mathbb{R}^m \text{ linear}\right\}.$$

The Gelfand width is related to the following quantity:

$$E_{\mathsf{ada}}^m(\mathcal{K}, \mathcal{X}) = \inf\left\{\sup_{x \in \mathcal{K}} \|x - \Delta(\Gamma(x))\|_{\mathcal{X}}, \ \Gamma : \mathcal{X} \to \mathbb{R}^m \text{ adaptive}, \ \Delta : \mathbb{R}^m \to \mathcal{X}\right\}, \qquad \text{(F.2)}$$

where $\Delta$ is an arbitrary (potentially nonlinear) reconstruction map and $\Gamma$ is an *adaptive* sampling map. By this, we mean that

$$\Gamma(x) = \begin{bmatrix} \Gamma_1(x) \\ \Gamma_2(x, \Gamma_1(x)) \\ \vdots \\ \Gamma_m(x, \Gamma_1(x), \ldots, \Gamma_{m-1}(x)) \end{bmatrix},$$

where $\Gamma_1 : \mathcal{X} \to \mathbb{R}$ is linear and, for $i = 2, \ldots, m$, $\Gamma_i : \mathcal{X} \times \mathbb{R}^{i-1} \to \mathbb{R}$ is linear in its first component.

*Proof of Theorem F.1.* We proceed in a series of steps.

*Step 1: Setup.* Define the functions

$$\phi_i(\boldsymbol{x}) = \sqrt{3}x_i, \quad \boldsymbol{x} \in D, \qquad i = 1, 2, \ldots.$$

Notice that these functions form an orthonormal system in $L_\varrho^2(D)$. (A.I) implies that these functions form a Riesz system in $L_\varsigma^2(D)$, with Riesz constants that are $\asymp 1$. Hence they have a (unique) biorthogonal Riesz system $\{\psi_i\}_{i=1}^\infty \subset L_\varsigma^2(D)$. Now define $\Phi_i = \phi_i \circ \iota$ and $\Psi_i = \psi_i \circ \iota$. Let $G \in L_\mu^2(\mathcal{X}; \mathbb{R})$ be arbitrary. Then

$$\|G\|_{L_\mu^2(\mathcal{X};\mathbb{R})} \geq \frac{\langle G, \sum_{i=1}^\infty \langle G, \Psi_i \rangle_{L_\mu^2(\mathcal{X};\mathbb{R})} \Psi_i \rangle_{L_\mu^2(\mathcal{X};\mathbb{R})}}{\| \sum_{i=1}^\infty \langle G, \Psi_i \rangle_{L^2(\mathcal{X};\mathbb{R})} \Psi_i \|_{L_\mu^2(\mathcal{X};\mathbb{R})}} = \frac{\sum_{i=1}^\infty |\langle G, \Psi_i \rangle_{L^2(\mathcal{X};\mathbb{R})}|^2}{\| \sum_{i=1}^\infty \langle G, \Psi_i \rangle_{L^2(\mathcal{X};\mathbb{R})} \Psi_i \|_{L_\mu^2(\mathcal{X};\mathbb{R})}}.$$

Consider the denominator. Using (A.II) and fact that the $\psi_i$ form a Riesz system, we see that

$$\left\| \sum_{i=1}^\infty \langle G, \Psi_i \rangle_{L^2(\mathcal{X};\mathbb{R})} \Psi_i \right\|_{L_\mu^2(\mathcal{X};\mathbb{R})} = \left\| \sum_{i=1}^\infty \langle G, \Psi_i \rangle_{L^2(\mathcal{X};\mathbb{R})} \psi_i \right\|_{L_\varsigma^2(D)} \lesssim \sqrt{\sum_{i=1}^\infty |\langle G, \Psi_i \rangle_{L^2(\mathcal{X};\mathbb{R})}|^2}.$$

We deduce that

$$\|G\|_{L^2(\mathcal{X};\mathbb{R})} \gtrsim \sqrt{\sum_{i=1}^\infty |\langle G, \Psi_i \rangle_{L^2(\mathcal{X};\mathbb{R})}|^2}, \quad \forall G \in L_\mu^2(\mathcal{X}; \mathbb{R}). \tag{F.3}$$

Now let $\boldsymbol{b} \geq \boldsymbol{0}$ with $\boldsymbol{b} \in \ell^1(\mathbb{N})$ and $I \subset \mathbb{N}$ with $|I| = N$. Using [7, Lem. 5.2] we see that the function

$$f = c \sum_{i \in I} c_i y \phi_i \in \mathcal{H}(\boldsymbol{b}), \tag{F.4}$$

for any $y \in \mathcal{Y}$, $\|y\|_{\mathcal{Y}} = 1$ and $\boldsymbol{c} = (c_i)_{i \in \mathbb{N}} \subset \mathbb{R}^{\mathbb{N}}$ with $|\boldsymbol{c}| \leq \boldsymbol{b}$ (i.e., $|c_i| \leq b_i, \forall i$), where $c > 0$ is a universal constant.

*Step 2: Reduction to a discrete problem.* Let $\mathcal{L}$ and $\mathcal{R}$ be arbitrary sampling and reconstruction maps as in (4.2). Following [7, Lem. 5.3], let $F = f \circ \iota$ and observe that

$$F(X) = cy \sum_{i \in I} c_i \Phi_i(X)$$

and therefore

$$\mathcal{L}(F) = y\Gamma(\boldsymbol{c}),$$

where $\Gamma : \mathbb{R}^{|I|} \to \mathbb{R}^m$ is given by

$$\Gamma(\boldsymbol{z}) = \begin{bmatrix} c \sum_{i \in I} z_i \Phi_i(X_1) \\ \vdots \\ c \sum_{i \in I} z_i \Phi_i(X_m) \end{bmatrix},$$

due to (4.1). Notice that $\Gamma$ is an adaptive sampling map of the form defined above. Now let $y^* \in B(\mathcal{Y}^*)$ be such that $|y^*(y)| = \|y\|_{\mathcal{Y}}$. Then, by (F.3),

$$\|F - \mathcal{R} \circ \mathcal{L}(F)\|_{L_\mu^2(\mathcal{X};\mathcal{Y})}^2 \geq \|y^*(F - \mathcal{R} \circ \mathcal{L}(F))\|_{L_\mu^2(\mathcal{X};\mathbb{R})}^2$$

$$\gtrsim \sum_{i \in I} |\langle y^*(F - \mathcal{R} \circ \mathcal{L}(F)), \Psi_i \rangle_{L_\mu^2(\mathcal{X};\mathbb{R})}|^2.$$

By biorthogonality, $\langle y^*(F), \Psi_i \rangle_{L^2_\mu(\mathcal{X};\mathbb{R})} = c\|y\|_{\mathcal{Y}} c_i = c \cdot c_i$, which implies that

$$\||F - \mathcal{R} \circ \mathcal{L}(F)|\|_{L^2_\mu(\mathcal{X};\mathcal{Y})} \gtrsim \|\boldsymbol{c} - \Delta \circ \Gamma(\boldsymbol{c})\|_2, \qquad (\text{F.5})$$

where

$$\Delta : \mathbb{R}^m \to \mathbb{R}^N, \quad \boldsymbol{y} \mapsto \Delta(\boldsymbol{y}) = \left( \langle y^*(\mathcal{R}(y \cdot \boldsymbol{y})), \Psi_i \rangle_{L^2_\mu(\mathcal{X};\mathbb{R})}/c \right)_{i \in I}.$$

Therefore,

$$\sup_{F \in \mathcal{H}(\boldsymbol{b},\iota)} \||F - \mathcal{R} \circ \mathcal{L}(F)|\|_{L^2_\mu(\mathcal{X};\mathcal{Y})} \gtrsim \inf_{\Gamma,\Delta} \sup_{\substack{\boldsymbol{c} \in \mathbb{R}^\mathbb{N} \\ \text{supp}(\boldsymbol{c}) \subseteq I \\ |\boldsymbol{c}| \leq \boldsymbol{b}}} \|\boldsymbol{c} - \Delta \circ \Gamma(\boldsymbol{c})\|_2,$$

where the infimum is taken over all adaptive sampling maps $\Gamma$ and reconstruction maps $\Delta$. Since $\mathcal{L}$ and $\mathcal{R}$ were arbitrary, we get

$$\theta_m(\boldsymbol{b}) \gtrsim E_m^{\text{ada}}(B(\boldsymbol{b},I), \ell_N^2),$$

where $B(\boldsymbol{b}, I) = \{\boldsymbol{z} \in \mathbb{R}^{|I|} : |z_i| \leq b_i, \ i \in I\}$ and $\ell_N^2 = (\mathbb{R}^N, \|\cdot\|_2)$.

*Step 3: Derivation of the lower bounds.* The next step is identical to the proof of Theorem 4.4 in [7]. This gives (F.1). □

*Proof of Theorem 4.1.* We use Theorem F.1. For (i), we let $\boldsymbol{b} = (b_i)_{i=1}^\infty$ by defined by

$$b_i = (2m)^{-1/p}, \ i = 1, \dots, 2m, \qquad b_i = 0, \ i > 2m.$$

This sequence $\boldsymbol{b} \in \ell_{\mathsf{M}}^p(\mathbb{N})$ with $\|\boldsymbol{b}\|_{p,\mathsf{M}} = \|\boldsymbol{b}\|_p = 1$. Moreover, we have

$$\sigma_m(\boldsymbol{b})_2 = 2^{-1/p} m^{1/2 - 1/p}.$$

For (ii) we let $\boldsymbol{b} = (b_i)_{i=1}^\infty$ be defined by $b_i = c_p (i \log^2(i))^{-1/p}$, where $c_p = \left( \sum_{i=1}^\infty 1/(i \log^2(i)) \right)^{-1/p}$. This sequence $\boldsymbol{b} \in \ell_{\mathsf{M}}^p(\mathbb{N})$ with $\|\boldsymbol{b}\|_{p,\mathsf{M}} = \|\boldsymbol{b}\|_p = 1$. Moreover,

$$\sigma_m(\boldsymbol{b})_2^2 \geq c_p^2 \sum_{i=m+1}^{2m} (i \log^2(i))^{-2/p} \geq c_p^2 \cdot m \cdot (2m \log^2(2m))^{-2/p},$$

as required. □

## G   Proof of Theorem 4.2

Much as in the previous section, the proof of Theorem 4.2 is a consequence of the following result.

**Theorem G.1.** *Suppose that the pushforward $\varsigma$ in (A.I) is a tensor-product of a univariate probability measure. Then the term $\tilde{\theta}_m(\boldsymbol{b})$ defined in (4.3) satisfies*

$$\tilde{\theta}_m(\boldsymbol{b}) \gtrsim \sigma_m(\boldsymbol{b})_1 / \log(m), \quad \forall \boldsymbol{b} \in \ell^1(\mathbb{N}), \ \boldsymbol{b} \geq \boldsymbol{0}, \|\boldsymbol{b}\|_1 \leq 1.$$

*Proof of Theorem G.1.* We proceed in a similar series of steps to those of the last proof.

*Step 1: Setup.* Let $\pi : \mathbb{N} \to \mathbb{N}$ be a bijection that gives a nonincreasing rearrangement of $\boldsymbol{b}$, i.e., $b_{\pi(1)} \geq b_{\pi(2)} \geq \cdots$. Now, let $r \in \mathbb{N}$ be arbitrary and consider the index set

$$I = I_1 \cup \cdots \cup I_r, \quad I_l = \{\pi((l-1)(m+1)+1), \dots, \pi(l(m+1))\}.$$

Notice that $|I| = r(m+1)$. Define the matrix

$$\boldsymbol{A} = \left( (\iota(X_i))_{\pi(j)} \right)_{i,j=1}^{m,r(m+1)}$$

and notice that we may write

$$\boldsymbol{A} = [\boldsymbol{A}_1 \quad \cdots \quad \boldsymbol{A}_r], \quad \text{where } \boldsymbol{A}_l = \left( (\iota(X_i))_{\pi((l-1)(m+1)+j)} \right)_{i,j=1}^{m,m+1}.$$

Let $\sigma$ be the one-dimensional probability measure associated with $\varsigma$. Then notice that each $\boldsymbol{A}_l$ is a random matrix whose entries are drawn i.i.d. from $\sigma$. Since $\sigma$ is supported in $[-1, 1]$, we deduce that

the $\boldsymbol{A}_l$ are independent subgaussian random matrices with the same distribution. Write $\gamma$ for this distribution. Let $t_1, \ldots, t_r > 0$ and $E_{l,t_l}$ be the event

$$E_{l,t_l} = \left\{ \exists \boldsymbol{u} \in N(\boldsymbol{A}_l) : \|\boldsymbol{u}\|_2 = 1, \ \|\boldsymbol{u}\|_\infty < \sqrt{t_l/(m+1)} \right\}, \quad l = 1, \ldots, r. \tag{G.1}$$

We will make a suitable choice of $t_1, \ldots, t_r$ later.

*Step 2: Reduction to a discrete problem.* Let

$$\mathcal{C} = \left\{ \boldsymbol{c} \in \mathbb{R}^{|I|} : |c_i| \le b_i, \ \forall i \in I, \ \boldsymbol{c} = \begin{bmatrix} \boldsymbol{c}_1 \\ \vdots \\ \boldsymbol{c}_r \end{bmatrix}, \boldsymbol{c}_l \in N(\boldsymbol{A}_l), \ l = 1, \ldots, r \right\}.$$

Notice that any $\boldsymbol{c} \in \mathcal{C}$ also satisfies $\boldsymbol{c} \in N(\boldsymbol{A})$. Now let $f = f_{\boldsymbol{c}}$ be as in (F.4) (we make the dependence on $\boldsymbol{c}$ explicit now for convenience). Let $\boldsymbol{x} \in D$. Then

$$f_{\boldsymbol{c}}(\boldsymbol{x}) = \sqrt{3} c y \sum_{i \in I} c_i x_i. \tag{G.2}$$

We deduce that

$$(F_{\boldsymbol{c}}(X_i))_{i=1}^m = \sqrt{3} c y \boldsymbol{A} \boldsymbol{c} = \boldsymbol{0},$$

where $F_{\boldsymbol{c}} = f_{\boldsymbol{c}} \circ \iota$. This implies that

$$\|F - \mathcal{R} \circ \mathcal{L}(F)\|_{L^\infty_\mu(\mathcal{X};\mathcal{Y})} = \|F - \mathcal{R}(\{X_i, 0\}_{i=1}^m)\|_{L^\infty_\mu(\mathcal{X};\mathcal{Y})},$$

where, for convenience, we let $\mathcal{L} : F \mapsto \{X_i, F(X_i)\}_{i=1}^m$. Therefore,

$$\sup_{F \in \mathcal{H}(\boldsymbol{b},\iota)} \|F - \mathcal{R} \circ \mathcal{L}(F)\|_{L^\infty_\mu(\mathcal{X};\mathcal{Y})} \ge \sup_{\boldsymbol{c} \in \mathcal{C}} \|F_{\boldsymbol{c}} - \mathcal{R}(\{X_i, 0\}_{i=1}^m)\|_{L^\infty_\mu(\mathcal{X};\mathcal{Y})}.$$

Now observe that $F_{\boldsymbol{0}} = 0$ and $\boldsymbol{0} \in \mathcal{C}$. Hence

$$\sup_{F \in \mathcal{H}(\boldsymbol{b},\iota)} \|F - \mathcal{R} \circ \mathcal{L}(F)\|_{L^\infty_\mu(\mathcal{X};\mathcal{Y})}$$

$$\ge \max \left\{ \|\mathcal{R}(\{X_i, 0\}_{i=1}^m)\|_{L^\infty_\mu(\mathcal{X};\mathcal{Y})}, \sup_{\boldsymbol{c} \in \mathcal{C}} \|F_{\boldsymbol{c}}\|_{L^\infty_\mu(\mathcal{X};\mathcal{Y})} - \|\mathcal{R}(\{X_i, 0\}_{i=1}^m)\|_{L^\infty_\mu(\mathcal{X};\mathcal{Y})} \right\}.$$

For $a > 0$, the function $x \mapsto \max\{x, a - x\}$ is minimized at $x = a/2$ and takes value $a/2$ there. We deduce that

$$\sup_{F \in \mathcal{H}(\boldsymbol{b},\iota)} \|F - \mathcal{R} \circ \mathcal{L}(F)\|_{L^\infty_\mu(\mathcal{X};\mathcal{Y})} \ge \frac{1}{2} \sup_{\boldsymbol{c} \in \mathcal{C}} \|F_{\boldsymbol{c}}\|_{L^\infty_\mu(\mathcal{X};\mathcal{Y})}.$$

Now, by (A.I),

$$\|F_{\boldsymbol{c}}\|_{L^\infty_\mu(\mathcal{X};\mathcal{Y})} = \|f_{\boldsymbol{c}}\|_{L^\infty_\varsigma(D;\mathcal{Y})} \gtrsim \|f_{\boldsymbol{c}}\|_{L^\infty_\varrho(D;\mathcal{Y})} = \sqrt{3} c \sup_{\|\boldsymbol{x}\|_\infty \le 1} \left| \sum_{i \in I} c_i x_i \right| = \sqrt{3} c \|\boldsymbol{c}\|_1.$$

With this in hand, we conclude that

$$\mathbb{E}_{X_1, \ldots, X_m \sim \mu} \sup_{F \in \mathcal{H}(\boldsymbol{b},\iota)} \|F - \mathcal{R}(\{X_i, F(X_i)\}_{i=1}^m)\|_{L^\infty_\mu(\mathcal{X};\mathcal{Y})} \gtrsim \mathbb{E}_{\boldsymbol{A}_1, \ldots, \boldsymbol{A}_r \sim \gamma} \sup_{\boldsymbol{c} \in \mathcal{C}} \|\boldsymbol{c}\|_1.$$

We now use the definition of $\mathcal{C}$ to write

$$\mathbb{E}_{X_1, \ldots, X_m \sim \mu} \sup_{F \in \mathcal{H}(\boldsymbol{b},\iota)} \|F - \mathcal{R}(\{X_i, F(X_i)\}_{i=1}^m)\|_{L^\infty_\mu(\mathcal{X};\mathcal{Y})}$$

$$\gtrsim \sum_{l=1}^r \mathbb{E}_{\boldsymbol{A}_l \sim \gamma} \sup_{\substack{\boldsymbol{c}_l \in N(\boldsymbol{A}_l) \\ |(\boldsymbol{c}_l)_i| \le b_i, \forall i \in I_l}} \|\boldsymbol{c}_l\|_1. \tag{G.3}$$

*Step 3: Bounding the expected error.* Fix $l = 1, \ldots, r$ and suppose the event $E_{l,t_l}$ defined (G.1) occurs. Let $\boldsymbol{u}_l$ be the corresponding vector and define

$$\boldsymbol{c}_l = \frac{b_{\pi(l(m+1))}}{\|\boldsymbol{u}_l\|_\infty} \boldsymbol{u}_l, \ l = 1, \ldots, r.$$

By construction, we have that $\boldsymbol{c}_l \in N(\boldsymbol{A}_l)$ and $|(\boldsymbol{c}_l)_i| \leq b_i, \forall i \in I_l$. We deduce that

$$\sup_{\substack{\boldsymbol{c}_l \in N(\boldsymbol{A}_l) \\ |(\boldsymbol{c}_l)_i| \leq b_i, \forall i \in I_l}} \|\boldsymbol{c}_l\|_1 \geq \frac{b_{\pi(l(m+1))}\|\boldsymbol{u}_l\|_1}{\|\boldsymbol{u}_l\|_\infty}.$$

Now observe that

$$1 = \|\boldsymbol{u}_l\|_2^2 \leq \|\boldsymbol{u}_l\|_1 \|\boldsymbol{u}_l\|_\infty.$$

Therefore, we get that

$$\sup_{\substack{\boldsymbol{c}_l \in N(\boldsymbol{A}_l) \\ |(\boldsymbol{c}_l)_i| \leq b_i, \forall i \in I_l}} \|\boldsymbol{c}_l\|_1 \geq \frac{b_{\pi(l(m+1))}}{\|\boldsymbol{u}_l\|_\infty^2} \geq \frac{b_{\pi(l(m+1))}(m+1)}{t_l}$$

whenever the event $E_{l,t_l}$ occurs. Using the law of total expectation, we deduce that

$$\mathbb{E}_{\boldsymbol{A}_l \sim \gamma} \sup_{\substack{\boldsymbol{c}_l \in N(\boldsymbol{A}_l) \\ |(\boldsymbol{c}_l)_i| \leq b_i, \forall i \in I_l}} \|\boldsymbol{c}_l\|_1 \geq \frac{b_{\pi(l(m+1))}(m+1)}{t_l} \mathbb{P}(E_{l,t_l})$$

for any fixed $t_l > 0$. We now appeal to [75, Thm. 1.4]. This shows that

$$\mathbb{P}(E_{l,t_l}^c) \leq c_2 m^2 \exp(-t_l/c_2), \quad \forall t_l \geq c_1 \log(m+1),$$

where $c_1, c_2 > 0$ are universal constants. We may without loss of generality assume that $c_2 \geq c_1 \geq 1$. Now set $t_l = c_2 \log(2c_2 m^2) \geq c_1 \log(m+1)$. Hence

$$\mathbb{P}(E_{l,t_l}^c) \leq 1/2.$$

We deduce that $\mathbb{P}(E_{t,t_l}) > 1/2$ and therefore

$$\mathbb{E}_{\boldsymbol{A}_l \sim \gamma} \sup_{\substack{\boldsymbol{c}_l \in N(\boldsymbol{A}_l) \\ |(\boldsymbol{c}_l)_i| \leq b_i, \forall i \in I_l}} \|\boldsymbol{c}_l\|_1 \geq \frac{b_{\pi(l(m+1))}(m+1)}{2c_2 \log(2c_2 m^2)}.$$

Now observe that

$$b_{\pi(l(m+1))}(m+1) \geq b_{\pi(l(m+1))} + \cdots + b_{\pi(l(m+1)+m)}$$

Substituting this into (G.3), we deduce that

$$\mathbb{E}_{X_1,\ldots,X_m \sim \mu} \sup_{F \in \mathcal{H}(\boldsymbol{b},\iota)} \|F - \mathcal{R}(\{X_i, F(X_i)\}_{i=1}^m)\|_{L_\mu^\infty(\mathcal{X};\mathcal{Y})} \gtrsim \frac{1}{\log(2m)} \sum_{i=m+1}^{r(m+1)+m} b_{\pi(i)}.$$

Since $r$ was arbitrary, we may take the limit $r \to \infty$. We now use the fact that

$$\sigma_m(\boldsymbol{b})_1 = b_{\pi(m+1)} + b_{\pi(m+2)} + \cdots.$$

to obtain the result. $\qquad \square$

*Proof of Theorem 4.2.* Using Theorem G.1, statements (i) and (ii) are derived in exactly the same way as in the proof of Theorem 4.1 $\qquad \square$

