# OpenReview forum: "Optimal deep learning of holomorphic operators between Banach spaces"
_NeurIPS.cc/2024/Conference — NeurIPS 2024 spotlight_

### Official Review · Reviewer_4yPo · 2024-07-08

**Soundness:** 4
**Presentation:** 3
**Contribution:** 4
**Rating:** 8
**Confidence:** 4

**Summary:**

This work examines learning holomorphic operators between Banach spaces. It shows that these learning tasks can be tackled using encoder-decoder approaches with deep neural networks (DNNs), achieving optimal approximation rates up to log factors. The architectures used are problem-agnostic and depend only on the amount of available training data. The study also delves into theoretical aspects of the associated training problem for standard feed-forward networks and presents numerical findings to support its claims.

**Strengths:**

* The paper is well-written and includes a thorough review of current research on learning holomorphic operators.
* The results are new and interesting, expanding on previous work by addressing holomorphic mappings between arbitrary Banach spaces and allowing standard training methods for the DNNs utilized.
* Although some assumptions and results are technical, I appreciate how the authors provide a clear overview of their meaning and interpretation. The proof techniques are explained well and summarized in the appendix, making them accessible without diving into deep details.
* The theoretical findings are also backed up by practical experiments.

**Weaknesses:**

Overall I do not have any serious concerns about this work and consider it to be a valuable contribution. Some questions are listed below.

**Questions:**

* Perhaps I missed it, but I could not locate a precise reference discussing the holomorphicity of the Navier-Stokes-Brinkman and Boussinesq equations, similar to Section B.3.2. for the Diffusion problem. It would be helpful if such a discussion could be included in the manuscript.

* Could you briefly summarize to what extent Theorems 4.1 and 4.2 require different techniques compared to those used in [7]? How does the use of a general Banach space $\mathcal{X}$ complicate the analysis?

* Regarding the experimental results, do you have any insights into why different activation functions perform differently across individual tasks? Can this behavior be explained using the results from Theorems 3.1 and 3.2?

* Do the authors think the results from Theorem 3.2 could be extended to include information on optimization procedures used for solving (2.5) (e.g., stochastic gradient descent and variants thereof)? Such an extension would enhance the practical relevance of the result. I am only seeking an educated guess or a high-level opinion, without expecting detailed analysis.

* I would appreciate it if the authors could briefly summarize the differences between the Banach space and Hilbert space cases for $\mathcal{Y}$. Specifically, could you indicate precisely where the inner product structure is utilized in the proofs to achieve better rates?  Also please double check formulas (3.3) and (3.5) in this context, currently it seems that they are the same.

**Limitations:**

Limitations and social impact have been discussed.

---

> ### Author Rebuttal · Authors · 2024-08-05
>
> Thank you for your careful review our paper and your positive assessment of it. Here we address the comments in your report.
>
> ## 'Perhaps I missed...'
>
> Good point. Right now, we do not have a proof that the NSB and Boussinesq problems satisfy the same holomorphy assumption as the diffusion equation. However, we strongly suspect this to be the case and intend to look into this in future work. **We will add a brief comment about this theoretical gap to the discussion of these equations in Section 5.**
>
> ## 'Could you briefly...'
>
> Good question. The proof of Theorem 4.1 is quite similar to that of [7]. The main difference is in the setup leading to the construction in (F.4). The fact that $\mathcal{X}$ is a Banach space does not substantially complicate the proof, as we make use of the map $\iota$ to push everything from $\mathcal{X}$ to $D$.
>
> The proof of Theorem 4.2 builds on the ideas from [7], but is quite different and more technical. The issue is that the approach of [7] cannot be easily extended to the $L^{\infty}$-norm. Like in the proof of Theorem 4.1, Theorem 4.2 considers holomorphic functions that are linear in the variables $x_i$, thereby reducing the problem to a discrete one (see Step 2). However, it now deviates in two key ways. First, it uses a more involved splitting -- see the set $I$ defined below line 1623. Second, it now assumes the coefficients in the expansion (G.2) come from the null spaces of certain matrices $\boldsymbol{A}\_l$. See the set $\mathcal{C}$ defined below line 1631. This is done to exploit the following properties: (i) $\boldsymbol{A}\_l$ is a subgaussian random matrix, and (ii) vectors in the null space of a subgaussian random matrix are 'flat'. In particular, for (ii) we rely on a result of [77]. We need this more technical argument in order to obtain the lower bound $m^{1-1/p}$. The nonsharp lower bound $m^{1/2-1/p}$ follows immediately from Theorem 4.1.
>
> **We will add some more comments in Section C.2 to explain these differences.** Also, Theorem 4.2 is also of independent interest, since it partially answers an open problem in [7] about proving results in the $L^{\infty}$-norm. **We will mention this in Section 4.**
>
>
> ## 'Regarding the experimental...'
>
> We generally observe that networks with smooth activation functions (e.g., tanh and ELU) perform better than ReLU networks when approximating smooth target functions such as the mappings considered in this work. In all experiments, we see the ReLU networks perform poorly relative to the best performing ELU ($4\times 40$ and $10\times 100$) and tanh ($4\times 40$) networks. Theorems 3.1 and 3.2 pertain only to tanh networks. However, as we discuss in Remark D.11, a key step of the proof involves emulating polynomials and in particular the multiplication operation in emulating the multivariate Legendre polynomials. Our main theorems can therefore be adapted to other activation functions without change, so long as this emulation is possible.
>
> As we discuss after Remark D.11, one can also show similar results for ReLU networks, albeit with worse bounds in the depth. This is due to the relation between the depth of such networks and the accuracy of approximate multiplication in the polynomial emulation step.  These worse theoretical bounds broadly agree with worse empirical results, although we do not claim they fully explain them.
>
> On the other hand, the poor performance of the larger $10\times 100$ tanh networks is not explained by our theoretical results. Any number of issues could result in poor practical performance. We choose standard optimizers and initializations in this work, however there may be better choices in particular for larger tanh networks. We leave investigation of the best parameterization for such networks to a future work.
>
> **We will add short discussions along these lines to Sections 5 and 6.**
>
> ## 'Do the authors...'
>
> We think this could potentially be done, but this is essentially an educated guess. It is something we intend to look into in the future.
>
> ## 'I would appreciate...'
>
> Excellent question. In general, the differences are: (i) when $\mathcal{Y}$ is a Banach space, we obtain $L^2$-error bounds in the Pettis norm, as opposed to the Bochner norm; and (ii) the approximation error rate is a factor of $(m/L)^{1/2}$ slower than in the Hilbert space case. See equations (3.3)-(3.6). We agree this discussion could have been clearer. **We will modify and expand lines 204-206 along these lines.**
>
> To explain further. When $\mathcal{Y}$ is a Banach space, the Bochner $L^2$-space does not have an inner product structure. However, by switching to the Pettis $L^2$-space, we can restore an inner-product-type structure and a Parseval's identity. To be more precise, if $g = \sum\_{\boldsymbol{\mathbf{\nu}} \in \Lambda} c\_{\boldsymbol{\mathbf{\nu}}} \Psi\_{\boldsymbol{\mathbf{\nu}}}$ with coefficients $c\_{\boldsymbol{\mathbf{\nu}}} \in \mathcal{Y}$ then in the Pettis norm one has the Parseval-type identity $|||g|||\_{L^2\_{\varrho}(D;\mathcal{Y})} = |||\boldsymbol{c}|||\_{2;\mathcal{Y}}$, whereas an analogous property does not hold in the Bochner norm. This property is crucial to all our analysis.
>  **We will add a short discussion of this in Section C.2.**
>
> You are correct that (3.3) and (3.5) are very similar. The difference is the norm on the LHS, which is the Pettis norm in (3.3) and the Bochner norm in (3.5). So the result for the Hilbert space case is stronger. We already discussed this in lines 204-206, but it could be clearer. **We will amend these lines.**

---

> > ### Comment · Reviewer_4yPo · 2024-08-07
> >
> > I am pleased with the explanations provided by the authors and am willing to increase my score to 8. I am also looking forward to follow-up work that may address my first and fourth points (as mentioned, I did not expect detailed responses here). Here are a few additional comments:
> >
> > * Regarding the tanh network: One possible reason for its behavior could be the vanishing gradient property of the tanh function, which may lead to suboptimal performance of the optimization algorithm in larger (deeper) networks. However, this is merely a rough guess, and further investigation is needed since there could be many other contributing factors, as the authors have noted.
> > * Regarding the Pettis norm: It might be helpful for the reader to introduce the associated notation ($|||.|||$) in the equation between 102 and 103. I believe this would address the source of my confusion.

---

> > > ### Author Response · Authors · 2024-08-08
> > >
> > > Thanks for your second positive assessment and further comments.
> > >
> > > Your comment about tanh networks is a good one. We agree this is a good starting point for further investigations and will take it on board in our future work.
> > >
> > > Thanks also for flagging the point about the Pettis norm. As another also referee pointed out, this was a typo in the equation between lines 102 and 103. We will add this notation to that equation in the revision.

---

### Official Review · Reviewer_XKAz · 2024-07-12

**Soundness:** 4
**Presentation:** 4
**Contribution:** 4
**Rating:** 8
**Confidence:** 3

**Summary:**

Authors provide theoretical results on sample efficiency for learning holomorphic neural operators $\mathcal{X}\rightarrow\mathcal{Y}$ (results are available for both Hilbert and Banach spaces) with a neural network of a particular structure. The structure of the network is encoder $\mathcal{X}\rightarrow\mathbb{R}^{d_1}$, decoder $\mathcal{Y}\rightarrow\mathbb{R}^{d_2}$ and DNN $\mathbb{R}^{d_1}\rightarrow \mathbb{R}^{d_2}$, with both encoder and decoder considered to be given and DNN trained on $l_2$ loss in a usual supervised manner.

Besides the upper bound on approximation error, authors also provide lower bound and conclude that active learning (deliberate collection of data used to train neural operator) is not justified, since non-adaptive training is already optimal up to log terms.

Authors illustrate their theoretical results (relative test error vs number of samples for selected DNN configurations) with several numerical experiments, including diffusion equation in $D=2$, Navier-Stokes-Brinkman equation in $D=2$ and Boussinesq equation in $D=3$.

**Strengths:**

Operator learning is a relatively recent DL setup, so novel theoretical results are of considerable interest. Authors managed to significantly improve previously known bounds and show that DNN can obtain near-optimal performance (in terms of the number of samples needed) in non-adaptive iid settings.

The contribution is primarily theoretical, but formal statements are well explained, supplemented with discussions and examples which significantly simplifies understanding of the results.

**Weaknesses:**

Overall, in my opinion, the research is of excellent quality, so I can not point to any major weaknesses.

The two main issues that can be relevant are: (i) the requirements for the encoder and decoder are not sufficiently discussed, and I find restricting the assumption that they are already available, and (ii) the significance of the numerical confirmation is not apparent.

**Questions:**

1. After line 102 there is a definition of Bochner and Pettis $L^p$ norms. Presumably, the later expression is a definition of $|||F|||$ which is not apparent.
2. I want to clarify several statements on the encoder and decoder.
   1. In (1.3) authors introduce encoder $\mathcal{E}$ and decoder $\mathcal{D}$, and later approximate encoder and decoder $\widetilde{\mathcal{E}}$, $\widetilde{\mathcal{D}}$ after line 114. What is the difference between those?
   2. In particular, in the discussion (lines 145, 146) of assumption A.IV (line 131) that requires both $\mathcal{E}$ and $\mathcal{D}$ to be linear and bounded, authors claim that $\widetilde{\mathcal{E}}$ is not required to be linear. It is not clear to me how this agrees with the fact that $\mathcal{E}_{\mathcal{X}}$ is defined in A.III as a composition of approximate encoder and decoder. Can the authors please clarify that?
   3. According to (1.3) encoder and decoder are defined for the approximate operator $\hat{F}$. This means, as I understand, they are not rigidly fixed by $F$ since presumably $F$ can be approximated with many different encoders and decoders. Approximate encoders and decoders are not formally defined, so it is not clear what they approximate. Does it all mean that authors consider not a continuous operator learning problem but suppose that discretization (of the space of parameters and $\mathcal{Y}$) is already performed ($\mathcal{E}$ and $\mathcal{D}$ are selected) and $\widetilde{\mathcal{E}}$ and $\widetilde{\mathcal{D}}$ approximate $\mathcal{E}$ and $\mathcal{D}$?
   4. Why there are $\widetilde{\mathcal{E}}\_{\mathcal{X}}$ and $\widetilde{\mathcal{D}}\_{\mathcal{X}}$ but not the approximate encoder and decoder for $\mathcal{Y}$?
3. Several questions on Theorem 3.1:
   1. Why $a_{\mathcal{Y}}$ is called approximation error? What is the meaning of this term (it looks like some kind of volume)? In the ideal setting $\mathcal{D}\_{\mathcal{Y}}\circ\mathcal{E}\_{\mathcal{Y}} = I$, so we have a perfect approximation that has nonzero approximation error.
   2. Since $\lim_{m\rightarrow\infty}m \big/ L(m) = \infty$ many terms in the upper bound will diverge unless the encoder and decoder are of sufficient quality. More specifically, the encoder and decoder should improve their accuracy as $\sim m^{-1/2}$ for u.b. to be useful in case of $2$ norm and as $\sim m^{-1}$ in case of Chebyshev norm. Can the authors please discuss this consequence of their result in more detail?
   3. Can the authors also explain the consequence of the optimization error term? It seems that $\tau$ that appears in it is to some extent beyond control and completely independent of $m$.
4. Questions about experiments:
   1. In experiments authors strive to confirm theoretical scaling $m^{-1}$. Since the encoder and decoder are perfect, and noise is absent, the only two sources of error are approximation error and optimization error. In my opinion, one can clearly see the signs of saturation for larger $m$. This is also clear from operator learning literature that relative error rarely fell below $10^{-3}$. Can the authors please comment on that? Is it because optimization error prevails for large $m$?
   2. Dimensions $d=4,8$ in Figures 1, 2, 3 and in the text (e.g., lines 296, 297: "PDE in $d=4$ and $d=8$ dimensions") are slightly misleading because they refer to the dimension of the parameter space and can be confused with the number of spatial dimensions.
   3. All results are given for perfect encoder and decoder. Can the authors share some thoughts on whether it is possible to test their upper bound with an imperfect decoder and encoder? What are those experiments might be? Is it possible to create a controllable setting to test terms in ub that contain encoder and decoder errors?

**Limitations:**

yes

---

> ### Author Rebuttal · Authors · 2024-08-05
>
> We appreciate your effort in reviewing our paper and are delighted by the positive assessment. Here we address the comments in your report.
>
> ## '1. After line 102...'
>
> Sorry. The term $||| F |||$ was missing. **We will correct the equation.**
>
> ## '2. I want to...'
>
> ## '1. In (1.3) authors...'
>
> Excellent point. We define $\mathcal{E}_{\mathcal{X}}$ in (A.III) in this way is so we can get error terms in our theorems that depend on $\mathcal{I}\_{\mathcal{X}}  - \widetilde{\mathcal{D}}\_{\mathcal{X}} \circ \widetilde{\mathcal{E}}\_{\mathcal{X}}$. This term is standard in operator learning, and says that the encoder-decoder pair approximates the identity. For PCA-Net or DeepONet, there are standard estimates for it (see lines 198-201). However, if we were to set $\mathcal{E}\_{\mathcal{X}} = \widetilde{\mathcal{E}}\_{\mathcal{X}}$, then the best we can prove is an error term involving $\iota\_{d\_{\mathcal{X}}} - \mathcal{E}\_{\mathcal{X}}$, which may not be small. For instance, in PCA-Net $\mathcal{I}\_{\mathcal{X}} - \widetilde{\mathcal{D}}\_{\mathcal{X}} \circ \widetilde{\mathcal{E}}\_{\mathcal{X}}$ is small because the empirical PCA eigenspace approximates the true PCA eigenspace. However, for $\iota\_{d\_{\mathcal{X}}} - \mathcal{E}\_{\mathcal{X}}$ to be small, one would need the empirical PCA eigenvectors to approximate the true PCA eigenvectors, which is a much more stringent condition.
>
> We agree this should be mentioned. **We will add a short discussion to lines 198-201**, where we discuss the encoding-decoding errors. **We will also mention this as a limitation of our analysis in our conclusion.**
>
> ## '2. In particular, in...'
>
> Apologies. What we meant to say is that in PCA-Net, the encoder $\mathcal{E}\_{\mathcal{X}}$ is linear. In our setup $\mathcal{E}\_{\mathcal{X}}$ may be nonlinear. We only require $\mathcal{D}\_{\mathcal{Y}}$ and $\mathcal{E}\_{\mathcal{Y}}$ to be linear, not the encoder and decoders on $\mathcal{X}$. **We will rephrase lines 145-146.**
>
> ## '3. According to (1.3)...'
>
> This is correct. We assume the approximate encoders/decoders have already been learned, and focus only on training the DNN $\widehat{N}$ in (1.3). As we mention in lines 96-97, this is standard for other recent works on generalization error analysis for operator learning. Crucially, our theorems allow for arbitrary encoders and decoders, subject to mild restrictions, and show error bounds based on how well the each such pair approximates the identity map. So, to answer the question, the encoder-decoder should give a good approximation of the identity map. Specifically, see the terms $E\_{\mathcal{X},q}$ and $E\_{\mathcal{Y},q}$ in (3.7). As noted above (see also lines 198-201), these are standard terms in operator learning analysis.
>
> We agree this could be clearer. **We will add a comment below line 116 where the encoders/decoders are first introduced.**
>
> ## '4. Why there are...'
>
> We think we understand the confusion. The tilde notation is just for convenience, as we later define the encoder $\mathcal{E}\_{\mathcal{X}}$ used in (2.5) in (A.III) in terms of $\widetilde{\mathcal{E}}\_{\mathcal{X}}$ and $\widetilde{\mathcal{D}}\_{\mathcal{X}}$. In particular, the encoders and decoders $\mathcal{E}\_{\mathcal{Y}}$ and $\mathcal{D}\_{\mathcal{Y}}$ on $\mathcal{Y}$ are also approximate.
>
> ## '3. Several questions...'
>
> ## '1. Why $a\_{\mathcal{Y}}$ is...'
>
> We use the term 'approximation error' to refer to the whole expression (3.6). The term $a\_{\mathcal{Y}}$ is just one component of this error. As you point out, $a\_{\mathcal{Y}}\approx 1$. But the essence of the approximation error is the factor involving $m/L$ raised to the negative power, which $\rightarrow 0$ as $m \rightarrow \infty$.
>
> ## '2. Since $\lim_{m\rightarrow\infty} m / L(m) = \infty$...'
>
> Good point. Similar factors also come up in other recent analyses of operator learning. But it is relevant as it means the encoder and decoder pairs must be learned to sufficient accuracy. **We will add a short comment on this to lines 199-201.**
>
> ## '3. Can the authors...'
>
> Yes, it is independent of $m$. Basically, this term accounts for inexact solution of the optimization problem. It says that if one minimizes the loss function to within $\tau^2$ of the optimum (regardless of the method used) then the overall error will be proportional to $\tau$. See line 181 and the definition of $E\_{\mathsf{opt}}$ in (3.8). The discrepancy between $\tau$ and $\tau^2$ is because we do not consider the squared error in our error bounds (3.3)-(3.5).
> **We will add a short remark about this to line 202-203.**
>
> ## '4. Questions about...'
>
> ## '1. In experiments...'
>
> Good point. We agree it is uncommon in operator learning to achieve more than 2-3 digits of accuracy. Some of our plots show this type of saturation. This may be related to the tradeoff between model class capacity and the complexity of the optimization procedure for larger values of $m$. Many previous works observed increases in testing error as models are given more training data up to an ``interpolation threshold,'' after which the testing error resumes decreasing, a.k.a. the *double-descent phenomenon*. It is difficult to say whether our results exhibit this phenomenon. Our training setup is not designed to explore this relationship, but rather to ensure the models are trained for sufficiently long (within budget limitations) and saving and restoring the best parameters to obtain as close as possible to a consistent learning procedure. However, we agree that optimization error for larger values of $m$ may be contributing to saturation. A more expansive numerical study would be necessary to fully investigate this issue. We plan to include such findings in future works.
>
> ## '2. Dimensions $d = 4,8$...'
>
> Good point. **We will clarify this.**
>
> ## '3. All results are...'
>
> Yes, we think this is possible and are looking into doing it. However, we feel a full study of this type is beyond the scope of this work.

---

### Official Review · Reviewer_ZpED · 2024-07-12

**Soundness:** 4
**Presentation:** 3
**Contribution:** 4
**Rating:** 7
**Confidence:** 4

**Summary:**

The paper addresses the challenge of learning operators between Banach spaces, focusing on holomorphic operators which have many applications. The authors employ standard (DNN) architectures with constant width greater than depth and $\ell_2$-loss minimization. They identify a family of DNNs that achieve optimal generalization bounds for such operators. For fully-connected architectures, they show there are infinitely many solutions that offer optimal performance. The results demonstrate that deep learning achieves the best possible generalization bounds, and no method can surpass these bounds, up to logarithmic terms.

**Strengths:**

This work presents powerful and well-formulated theorems. The techniques used on holomorphic operators are interesting, and thanks to them, a generalization bound can be demonstrated. The work is novel, highly original, and I believe it could have a fruitful impact in the context of deep learning.

**Weaknesses:**

I think some of the figures of the experiments results are too small, making it a bit difficult to recognize the results. It would be good if the authors could review this issue.

**Questions:**

No questions.

**Limitations:**

Theorem 3.2 only asserts that some minimizers are ‘good’, not all.

---

> ### Author Rebuttal · Authors · 2024-08-05
>
> We greatly appreciate the effort you have put into reviewing our paper and are delighted by your positive opinion of the work. Here we address the comment(s) in your report.
>
> ## `I think some of the figures...'
>
> Good point. We will use the additional page allowed in the camera-ready version to enlarge the figures in the final version of the paper.
>
> ## `Theorem 3.2 only asserts...'
>
> We very much agree. As we say in Section 6 (lines 340-344), we are actively looking into whether we can prove stronger results, either about all minimizers or by showing that `good' minimizers can be obtained through standard training.

---

> > ### Comment · Reviewer_ZpED · 2024-08-13
> >
> > Thanks for your response!

---

### Decision · Program_Chairs · 2024-09-25

**Decision:**

Accept (spotlight)

**Comment:**

The paper presents some results on learning holomorphic operators between Banach spaces using deep neural networks. Theoretical results about generalization bounds and optimality of these bounds are given along with some numerical experiments relating to several different parametric PDEs. Based on the reviews, it is agreed that the strengths of the paper are its powerful, well-formulated theoretical results, its rigorous presentation, and its originality. All reviews furthermore indicated no major weaknesses in the paper, indicating that the results are technically sound and interesting to the community.

Regarding the rebuttal, as all reviews were quite positive, there was little to discuss besides superficial changes (figure sizes, minor clarifications/rewordings). The authors responded to all questions and criticisms of the reviewers sufficiently and the reviewers participated in the discussion as necessary.

Since all reviewers were in agreement with respect to the quality of the submission, I did not see it necessary to have a post-rebuttal discussion.

I recommend accepting the paper.